# Distributed Optimization for Overparameterized Problems: Achieving Optimal Dimension Independent Communication Complexity

**Bingqing Song**
Department of ECE
University of Minnesota
email:song0409@umn.edu

**Ioannis Tsaknakis**
Department of ECE
University of Minnesota
email:tsakn001@umn.edu

**Chung-Yiu Yau**
Department of SEEM
Chinese University of Hong Kong
email:cyyau@se.cuhk.edu.hk

**Hoi-To Wai**
Department of SEEM
Chinese University of Hong Kong
email:htwai@cuhk.edu.hk

**Mingyi Hong**
Department of ECE
University of Minnesota
email:mhong@umn.edu

## Abstract

Decentralized optimization are playing an important role in applications such as training large machine learning models, among others. Despite its superior practical performance, there has been some lack of fundamental understanding about its theoretical properties. In this work, we address the following open research question: To train an overparameterized model over a set of distributed nodes, what is the *minimum* communication overhead (in terms of the bits got exchanged) that the system needs to sustain, while still achieving (near) zero training loss? We show that for a class of overparameterized models where the number of parameters $D$ is much larger than the total data samples $N$, the best possible communication complexity is $\Omega(N)$, which is independent of the problem dimension $D$. Further, for a few specific overparameterized models (i.e., the linear regression, and certain multi-layer neural network with one wide layer), we develop a set of algorithms which uses certain linear compression followed by adaptive quantization, and show that they achieve dimension independent, near-optimal communication complexity. To our knowledge, this is the first time that dimension independent communication complexity has been shown for distributed optimization.

## 1 Introduction

The research of decentralized/distributed optimization has recently gained tremendous momentum, partly due to the fact that a well-designed decentralized/distributed algorithm is capble of training large-scale machine learning models, by utilizing massively distributed computation and data resources. In a typical decentralized optimization setting, multiple local agents hold small to moderately sized datasets, and they collaborate by iteratively solving their local problems while sharing some information with other agents. The majority of the state-of-the-art distributed/decentralized optimization algorithms are deeply rooted in classical consensus-based approaches [Tsitsiklis, 1984, Tsitsiklis and Luo, 1987], where the agents repetitively share the local optimization variables, or the gradients of local functions, so to reach an optimal *consensual* solution.

36th Conference on Neural Information Processing Systems (NeurIPS 2022).

It is important to note that in those classical settings, the agents typically solve *small-scale* problems where the problem dimension (i.e., the dimension of the optimization variable, and the gradients) is relatively small (e.g., in the order of hundreds or thousands). Therefore, it is reasonable that the agents should iteratively share their local parameters/gradients to improve the quality of the models. However, many modern learning models are in the *overparameterized* regime with very high-dimensional parameters [He et al., 2016, Vaswani et al., 2017], such as deep and/or wide neural networks. This practice poses some significant challenges to the classical consensus-based approaches, because by directly sharing the model parameters and/or gradients, the communication burden increases as problem size increases.

A line of recent works have been developed, which adopt strategies such as gradient/model compression, lazy communication, etc, and they are able to significantly reduce the overall communication complexities [Xu et al., 2020, Sun et al., 2020, Li et al., 2019, Karimireddy et al., 2019]. A few works focus on designing compressors [Safaryan et al., 2020, Horváth et al., 2019, Beznosikov et al., 2020, Konečný et al., 2016, Safaryan et al., 2021]. Although the majority of these works are able to reduce *per round* communication burden, it is by no means clear if, to achieve certain solution quality, the *total* amount of communication can be saved. In fact, it has been shown in [Arjevani and Shamir, 2015], that the total communication achievable is dependent on the problem dimension, which could be huge. More importantly, there is a general lack of fundamental understanding about the performance limits for decentralized and distributed optimization, when the models are 'large'. Without such characterization, it is by no means clear if the existing algorithms are *optimal* in reducing the communication complexities.

In this work, we focus on distributed optimization for 'large' machine learning models (i.e., overparameterized problems, to be defined shortly), and we ask the following question:

> **(Q)** What is the *best possible* communication complexity (in terms of number of bits needed to obtain certain solution accuracy), achievable by *any* distributed algorithms for a class of overparameterized problems? Which class of algorithms achieves such complexity bounds?

**Related Works.** A number of recent works are devoted to reducing the communication cost in distributed/decentralized optimization by combining the optimization algorithms with *compression*, e.g., sparsification, quantization, etc. For the distributed setting with the server-worker architecture, relevant works include [Alistarh et al., 2017, Wen et al., 2017, Wang et al., 2018, Bernstein et al., 2018, Haddadpour et al., 2021, Mishchenko et al., 2019, Gorbunov et al., 2021]; see [Richtárik et al., 2021] for a recent survey. For the decentralized setting without a central server, recent works such as [Koloskova et al., 2019, Reisizadeh et al., 2019, Tang et al., 2018, Vogels et al., 2020, Magnússon et al., 2020] proposed algorithms that utilize error compensation, low-rank compression, etc. While they have demonstrated appealing performance in practice, the theoretical guarantees do not scale for overparameterized problems. For instance, [Koloskova et al., 2019] showed an $\mathcal{O}(1/T + 1/(\omega T)^2)$ iteration complexity upper bound for their algorithm. The quantity $1/\omega$ is the compressor parameter which is at least $\Omega(\sqrt{D})$, where $D$ is the problem dimension.

Lower communication complexity bounds for distributed optimization problems have been studied since the 1980s. There are two major categories of lower bound results in literature depending on how we count the "communication" required to reach a solution of given accuracy, i.e., 1) in terms of the number of communication rounds, or 2) in terms of the number of bits got communicated. The first category is not directly related to the current work; we refer to readers to Appendix A for discussions. In second category, the first lower bound is given in [Tsitsiklis and Luo, 1987] where the authors consider a distributed setting with two nodes and provide a $\Omega\left(D\log\left(D/\epsilon\right)\right)$ bound for the class of quadratic objectives, where $\epsilon$ is the solution accuracy, and $D$ is the variable dimension. In [Zhang et al., 2013] lower communication bounds are derived for certain statistical estimation tasks, such as linear and probit regression. Moreover, in [Vempala et al., 2020] the communication complexity of several distributed optimization tasks is studied, including $\ell_p$ regression, linear, semidefinite and convex optimization. In [Korhonen and Alistarh, 2021], an $\Omega\left(KD\log\left(D/\epsilon\right)\right)$ bound is derived for quadratic problems on a multi-node setting, for both deterministic and randomized protocols, where $K$ is the number of clients.

Recently, overparameterized problem has drawn great research interests. It has been showed that remarkable performance can be achieved with overparameterized networks [Jacot et al., 2018, Du et al., 2019, Allen-Zhu et al., 2019]. Some recent works study the overparameterization of decentralized problems, which guarantee the global convergence [Huang et al., 2021, Deng et al., 2022]. However,

the high communication cost caused by the large model is a key problem in decentralized algorithm. For example, the ReLU fully connected networks designed in the aforementioned two works are of widths $\Omega(N^4)$ and $\Omega(N^{18})$ respectively, where $N$ is the number of data samples available.

**Contributions.** This work addresses both questions in **(Q)**. The contributions are highlighted below:

**(i)** Regarding to the *best possible* communication complexity (i.e., the communication complexity lower bound), we first show that for a class of Polyak-Lojasiewicz (PL) problems, the communication lower bound is $\Omega(KD \log(D/\epsilon))$, which is similar to the known results for strongly convex problems. Further, we show that for a class of overparameterized models where the loss function is a composition of a quadratic problem and some nonlinear functions, and that the number of parameters $D$ is much larger than the total data samples $N$, the communication complexity lower bound becomes $\Omega(KN \log(K/\epsilon))$, which is *independent* of the problem dimension $D$.

**(ii)** We develop a distributed algorithm which can be used to optimize both classes of problems, in the setting where multiple clients are coordinated by a server. We show that the lower bound for the PL problem can be matched; while by using certain linear random compression/decompression scheme, followed by adaptive quantization, the lower bound for the overparameterized problem can be matched (up to a $\log(D)$ factor) for two of its important special cases, i.e., linear regression, and certain multi-layer neural network with one wide layer. Further, we show that the proposed algorithm can be extended to a fully decentralized setting without the coordination of a server.

To our knowledge, these are the first set of results that attempt to comprehensively address the issue of optimal communication complexities for overparameterized problems. We expect that results provided in this work will open doors for understanding more complex communication-efficient distributed algorithms, as well as larger classes of overparameterized problems.

## 2   The System Model

Consider a system having $K$ distributed agents, and a total of $N$ samples $\{(X_i, y_i)\}_{i=1}^N$. Suppose each agent has $N_k$ data samples, with $\sum_{k=1}^K N_k = N$; let set $\mathcal{X}_k$ (resp. $\mathcal{Y}_k$) collect agent $k$'s data samples (resp. labels). They cooperatively solve an optimization problem of the following form:

$$\min_{\theta \in \Theta} \ f(\theta; \mathcal{X}, \mathcal{Y}) := \sum_{k=1}^K f_k(\theta; \mathcal{X}_k, \mathcal{Y}_k) \tag{1}$$

where $f_k : \Theta \to \mathbb{R}$ is the local loss function for agent $k$; $\theta$ is the optimization variable, whose domain is $\Theta \subseteq \mathbb{R}^D$. For notational simplicity, we will ignore the reference for the data samples $\mathcal{X}, \mathcal{Y}$ and $\mathcal{X}_k, \mathcal{Y}_k$ whenever possible, i.e., we will write $f(\theta)$ and $f_k(\theta)$ as the global and local loss functions.

Throughout the paper, we will consider a few special cases of (1), where the local function $f_k$'s are *overparameterized*. The first of such problem class is referred to as the *distributed PL problems*. To introduce this class, let us provide a few assumptions.

**Assumption 1.** *The local objective functions satisfy:*

$$2\mu_k \cdot (f_k(\theta) - f_k(\theta_{(k)}^*)) \le \|\nabla f_k(\theta)\|^2, \quad \forall\, \theta, \quad \forall\, k, \tag{2}$$

*where $\theta_{(k)}^*$ is a global minimum of $f_k(\cdot)$; $\mu_k$'s some positive constants.*

**Assumption 2.** *There exists positive constants $L_k$'s and $L \ge 1$ such that:*

$$\|\nabla f_k(\theta) - \nabla f_k(\theta')\| \le L_k \|\theta - \theta'\|, \quad \|\nabla f(\theta) - \nabla f(\theta')\| \le L \|\theta - \theta'\|, \quad \forall\, \theta, \theta', \,\forall\, k.$$

Note that (2) is the so-called PL assumption, which was introduced in [Polyak, 1963]. It is weaker than strong convexity, while it can still be used to ensure the linear convergence of gradient descent. It is also less restrictive than several other related conditions under which linear global convergence can be derived, such as the restricted secant inequality [Karimi et al., 2016]. The second inequality in (2) has also been used in the literature; see, e.g., [Oymak and Soltanolkotabi, 2019], in which such a bound for overparameterized network has been showed. Assumption 2 is the standard Lipschitz gradient condition.

**Definition 1** (Distributed PL problems). *Consider problems of the form* (1). *Define:*

$$\mathcal{C}_{\mathrm{pl}} := \left\{ f(\theta) = \sum_{k=1}^K f_k(\theta) : f_k(\theta) \text{ satisfies Assumptions } 1-2, \,\forall\, k \in [K] \right\}.$$

The second class of problems is referred to as the *distributed overparameterized* problem, where the loss function is a composition of least square loss and a non-linear function. This form of loss function often appears in analyzing neural networks, see, e.g., [Oymak and Soltanolkotabi, 2019].

**Definition 2** (Distributed overparameterized problems). *Consider problems of the form* (1). *Define functions $G_k(\cdot) : \Theta \to \mathbb{R}^{N_k}$ and constants $b_k \in \mathbb{R}^{N_k}$, for all $k$. We call $G_k(\cdot)$ overparameterized if for every $b_k \in \mathbb{R}^{N_k}$, there exists $\theta \in \Theta$ such that $G_k(\theta) = b_k$. Define the following class of problems:*

$$\mathcal{C}_{\mathrm{op}} := \left\{ f(\theta) = \sum_{k=1}^{K} f_k(\theta), \ f_k(\theta) = \frac{1}{2}\|G_k(\theta) - b_k\|_2^2 : G_k(\cdot) : \Theta \to \mathbb{R}^{N_k} \text{ overparameterized.} \right\}$$

## 3 Communication Complexity Lower Bounds for Deterministic Algorithms

In this section we derive the communication complexity lower bounds for classes $\mathcal{C}_{\mathrm{op}}$ and $\mathcal{C}_{\mathrm{pl}}$. Throughout this section and the paper, our focus is given to *deterministic* algorithms where no randomness is involved. All the proofs for results presented in this section are given in Appendix B.

We start with a description of the *coordinator model* (also see [Braverman et al., 2013, Korhonen and Alistarh, 2021]), which will be used later as the basis of our derivation. Consider a set of $K$ distinct nodes/agents, and a single coordinator node. Each node and the coordinator have a two-way communication channel, over which binary messages are exchanged. Consider a *class of problems $\mathcal{C}$* of the form (1), that is $f(\theta) = \sum_{k=1}^{K} f_k(\theta)$, where each node $k$ has access only to objective $f_k(\theta)$. Moreover, let $\Pi$ be a set of protocols that are executed over the coordinator model. Then, given a problem class $\mathcal{C}$, a *protocol $\pi \in \Pi(\epsilon)$* takes as input a problem $c \in \mathcal{C}$, performs a number of message exchanges (over the coordinator model), and outputs an $\epsilon$-approximate solution $\tilde{\theta} \in \Theta$ of problem (1). That is, the output $\tilde{\theta}$ satisfies (where $\theta^*$ is a point where the exact minimum is attained):

$$f(\theta^*) \leq f(\tilde{\theta}) \leq f(\theta^*) + \epsilon, \tag{3}$$

Next, let us look at $\pi$ in more detail. We assume that for $\pi \in \Pi(\epsilon)$, it performs $T(\epsilon)$ rounds of communication and computation, where during each round the following steps are carried out:

- At round $t = 0$, each agent/node $k$ is given as input the local objective $f_k$ (e.g., the data samples $\mathcal{X}_k, \mathcal{Y}_k$ assigned to each node);
- At round $1 \leq t < T(\epsilon)$,
    - The coordinator sends a message $m_{c \to k}^{(t)} \in \{0, 1\}$ to some node $k$.
    - Node $k$ sends message $m_{k \to c}^{(t)} \in \{0, 1\}$ to the coordinator.
- At the final round $t = T(\epsilon)$ the coordinator outputs an $\epsilon$-approximate solution.

Next, we define the communication complexity of the distributed protocol. Let us denote the total number of bits exchanged under protocol $\pi$, for problem $c \in \mathcal{C}$, as $M(c, \pi, \epsilon)$; Use the following definition, which has been used extensively in related works such as [Tsitsiklis and Luo, 1987].

**Definition 3** (Communication complexity). *The communication complexity of protocol $\pi$ over problem class $\mathcal{C}$ is defined as the maximum number of bits exchanged across the problems in $\mathcal{C}$, i.e., $M(\mathcal{C}, \pi, \epsilon) := \max_{c \in \mathcal{C}} M(c, \pi, \epsilon)$. Also, the communication complexity of problem class $\mathcal{C}$ is the minimum communication complexity across all protocols that solve $\mathcal{C}$, i.e.,*

$$M(\mathcal{C}, \epsilon) := \min_{\pi \in \Pi(\epsilon)} M(\mathcal{C}, \pi, \epsilon) = \min_{\pi \in \Pi(\epsilon)} \max_{c \in \mathcal{C}} M(c, \pi, \epsilon).$$

Next, we derive communication complexity lower bounds for certain problem classes $\mathcal{C}$. Our construction strategies follow those from [Korhonen and Alistarh, 2021], that is, we will reduce a problem with known communication complexity, to hard problem instances within classes of our interest. The main difference from the aforementioned work (which only deals with quadratic problems) is that, we need to construct hard PL and overparameterized problems instances.

Below we derive communication complexity lower bounds for the class of deterministic distributed problems where the local loss functions belong to the class of PL functions defined in Def. 1.

**Theorem 3.1.** *Consider the problem class $\mathcal{C}_{\mathrm{pl}}$ with given parameters $K, D, \epsilon$, for which it holds that $\frac{D}{\epsilon} = \Omega(1)$. Then we have $M(\mathcal{C}_{\mathrm{pl}}, \epsilon) = \Omega(KD \log\left(\frac{D}{\epsilon}\right))$.*

Note that the condition $D/\epsilon = \Omega(1)$ is reasonable since we will mainly focus on the regime where $D$ is large. Next we consider $\mathcal{C}_{\mathrm{op}}$. We have the following communication complexity lower bound.

**Theorem 3.2.** *Consider the problem class $\mathcal{C}_{\mathrm{op}}$, with given parameters $K, N, D, \epsilon$, for which it holds that $\frac{N}{\epsilon} = \Omega(1)$. Then we have that $M(\mathcal{C}_{\mathrm{op}}, \epsilon) = \Omega(KN \log\left(\frac{N}{\epsilon}\right))$.*

Clearly, the lower bound for class $\mathcal{C}_{\mathrm{op}}$ does not explicitly depend on the problem dimension $D$, while that for the $\mathcal{C}_{\mathrm{pl}}$ does. This appears to be reasonable, partly because in the class of $\mathcal{C}_{\mathrm{pl}}$ the number of data samples $N$ has not been explicitly modeled. Nevertheless, at this point it is not clear if such a *dimension-independent* lower bound is tight or not, because we are yet to find an algorithm to match such a lower bound. To our knowledge, these are the first set of results that characterize the communication complexity for the classes $\mathcal{C}_{\mathrm{pl}}$ and $\mathcal{C}_{\mathrm{op}}$, and they will serve to guide our subsequent algorithm development and analysis.

## 4 Algorithms Achieving the Optimal Communication Complexities

In this section, we design algorithms and analyze the *achievable* communication complexities, for the PL and the overparameterized problems in Def. 1 and 2. We will begin with the simple setting where all the agents are connected to a coordinator, and then extend to the case where the agents are decentralized. Throughout this section, for simplicity of exposition, we will assume that $\Theta \equiv \mathbb{R}^D$.

Let us begin with a generic algorithm to deal with the server-agent setting. The algorithm is an adaptation of the classical quantize-then-communicate strategy [Tsitsiklis and Luo, 1987, Magnússon et al., 2020]. At each iteration, each agent first computes its gradient, followed by certain *adaptive* quantization scheme, before getting transmitted to the collaborating agents.

Specifically, at $t$-th iteration, each agent uses a function $F : \mathbb{R}^D \to \mathbb{R}^H$ to compress the local gradient; Then the output is quantized as $q_k^t \in \mathbb{R}^H$ (use the method in Def. 4). Each agent then shares $q_k^t$ to the server, and the aggregated $\mathbf{q}^t$ is broadcast to all agents. Each agent then uses a function $\tilde{F} : \mathbb{R}^H \to \mathbb{R}^D$ to process the received vector, and updates the local parameters. One key difference compared with [Tsitsiklis and Luo, 1987, Magnússon et al., 2020] is that, the algorithm uses a pair of *compression and decompression functions* $F(\cdot)$ and $\tilde{F}(\cdot)$ to process the gradients transmitted/received. We will see shortly that such a pair plays an important role in matching the lower bound. For the detailed steps, see Algorithm 1 below.

---

**Algorithm 1** Limited Communication Distributed Optimization Algorithm
___

1: Initialize: Fix $F(\cdot)$ and $\tilde{F}(\cdot)$, $\eta > 0$, $\gamma^t \in \mathbb{R}_+$, $b \in \mathbb{Z}_+$; set $\theta_k^0 = \theta^0$, choose $q_k^0$ which can be represented by $b_0$ bits for each entry, such that $\|q_k^0 - F(\nabla f_k(\theta^0))\|_\infty \leqslant \frac{\sqrt{f_k(\theta^0)}}{C \cdot D^3 \sqrt{K}}$, where $C$ is some constant independent of $D$.

2: **for** $t = 0, 1, \ldots, T-1$ **do**

3:    `Update:` Each agent updates $\theta_k^{t+1} = \theta^t - \eta \tilde{F}(\sum_{k=1}^{K} q_k^t)$; set $\theta^{t+1} = \theta_k^{t+1}$.

4:    `Compute:` The function of gradient of each agent $g_k^{t+1} = F(\nabla f_k(\theta^{t+1}))$;

5:    `Quantize:` Each agent quantizes $q_k^{t+1} = \mathrm{quant}(g_k^{t+1}, q_k^t, \gamma^t, b)$, defined in Def. 4;

6:    `Communicate:` Each agent sends $q_k^{t+1}$ to the server;

7:    `Aggregation:` The server collects $\mathbf{q}^{t+1} = (q_1^{t+1}, q_2^{t+1}, \cdots, q_K^{t+1})$;

8:    `Broadcast:` The server broadcasts $\mathbf{q}^{t+1} = (q_1^{t+1}, q_2^{t+1}, \cdots, q_K^{t+1})$ to each agent;

9: **end for**

10: Return: Parameters $\theta^T$.

---

In Algorithm 1, we note that as long as the initial solution $\theta^0$ is the same among all the agents, then $\theta_k^{t+1}$ computed in the 'Update' step will always be the same across the agents. This is the reason that we can directly set $\theta_k^{t+1}$ as $\theta^{t+1}$. Subsequently we will extend this algorithm to a *fully decentralized* setting, where precise consensus is no longer required. We choose the quantization method for the first iteration defined in Appendix C. In the subsequent iterations, we use the following adaptive quantization scheme; also see [Magnússon et al., 2020].

**Definition 4.** *Fix $r \in \mathbb{R}_+$ and let $b \in \mathbb{N}$ be an positive integer. Let* quant: $\mathbb{R}^H \times \mathbb{R}^H \times \mathbb{R}_+ \times \mathbb{N} \to \mathbb{R}^H$ *be the quantization function defined component-wise as follows, where $\delta(r,b) := r / \left(2^b - 1\right)$,*

$$[\text{quant}(c, p, r, b)]_j = \begin{cases} p_j - r & \text{if } p_j \leq c_j - r + \delta(r,b) \\ p_j + r & \text{if } p_j \geq c_j + r - \delta(r,b) \\ p_j - r + 2\delta(r,b) \left\lfloor \frac{c_j - p_j + r + \delta(r,b)}{2\delta(r,b)} \right\rfloor & \text{otherwise.} \end{cases}$$

Essentially, the quantization scheme searches for a closest point on the grid (which has a width $2r$ and is centered at $\mathbf{p} = (p_1, \ldots, p_H)$. The distance between the neighbor points on the grid is $\delta(r,b)$.

**Lemma 4.1.** *Fix $p \in \mathbb{R}^H$, then for all $c \in \mathbb{R}^H, Q \in \mathbb{R}^{D \times H}$, such that $\|c - p\|_\infty \leq r$ we have*

$$\|\text{quant}(c, p, r, b) - c\|_\infty \leq \frac{r}{2^b - 1}, \quad \|Q \cdot \text{quant}(c, p, r, b) - Q \cdot c\|_\infty \leq \frac{rH\|Q\|_\infty}{2^b - 1}.$$

### 4.1 Communication Complexity Upper Bounds for $\mathcal{C}_{\text{pl}}$

We will begin with analyzing the class of problem $\mathcal{C}_{\text{pl}}$, as defined in Def. 1. Before we proceed, let us make an additional assumption.

**Assumption 3.** *Assume that each $f_k(\cdot)$ is non-negative in its domain; Further, assume that the optimal objective value for problem (1) is zero, that is, there exists $\theta^* \in \mathbb{R}^D$ such that $f(\theta^*) = 0$.*

The above 'zero loss' assumption has been used by a number of recent works to characterize problems with a large number of parameters, e.g., in overparameterized neural network [Liu et al., 2022, Nguyen and Mondelli, 2020]; in optimization literature, this assumption is also used to improve the analysis of gradient-based algorithms; see, e.g. [Razaviyayn et al., 2019].

**Lemma 4.2.** *Suppose Assumption 2 – 3 are satisfied, then we have the following inequality:*

$$\|\nabla f(\theta)\|^2 \leqslant 2L \cdot f(\theta), \quad \forall \theta.$$

The above lemma shows that together with the PL condition, Assumptions 2, 3 imply:

$$2\mu \cdot f(\theta) \leq \|\nabla f(\theta)\|^2 \leq 2L \cdot f(\theta), \quad \forall \theta.$$

For convenience, throughout our analysis we set $\mu < 1, L > 1$.

We will refer to the class of problems satisfying Assumption 1 to 3 as $\widetilde{\mathcal{C}}_{\text{pl}}$. We note that $\widetilde{\mathcal{C}}_{\text{pl}}$ is a strict subset of the class $\mathcal{C}_{\text{pl}}$. Next, let us present our result for $\mathcal{C}_{\text{pl}}$. The proof can be found in Appendix C.

**Theorem 4.1.** *Suppose Assumptions 2 – 3 are satisfied. Consider using Alg. 1 to solve $\mathcal{C}_{\text{pl}}$; set*

$$\eta = (2L^{\frac{3}{2}})^{-1}, \tau = (\sqrt{C}D)^{-1}, b = \max\left(\log_2(\sqrt{C}D + 1), b_0\right), \tilde{F}(x) = F(x) = x,$$

*and $C := \max\left(\sqrt{16L/\mu^2 + L^{\frac{3}{2}}/\mu}, 100\right)$. Then the following holds true:*

$$f\left(\theta^{t+1}\right) \leq \left(1 - \mu/(8L^{\frac{3}{2}})\right) f\left(\theta^t\right), \quad \forall t = 0, 1, \cdots.$$

*Further, to compute an $\epsilon$-optimal solution satisfying (3), each agent is required to transmit:*

$$8b \cdot D \cdot \mu^{-1} L^{\frac{3}{2}} \log(f(\theta^0)/\epsilon) \quad \text{bits/agent}.$$

The quantization strategy at the first iteration (in Appendix C) requires $b_0$ to be $\mathcal{O}\left(\log(D)\right)$, so we can conclude $b = \mathcal{O}\left(\log(D)\right)$. Thus Theorem 4.1 gives the total communication complexity for each agent as $\mathcal{O}\left(D \log(D) \cdot \log(\frac{1}{\epsilon})\right)$, which matches the lower bound in Theorem 3.1, up to a gap $\log(D)$.

### 4.2 Communication Complexity Upper Bounds for $\mathcal{C}_{\text{op}}$

Next, we develop algorithms for the distributed overparameterized problem defined in Def. 2. Note that functions in the class $\mathcal{C}_{\text{op}}$ can be non-convex, therefore one may not be able to find an algorithm that can solve the entire class of problems in polynomial time. Our strategy is to identify special cases of $\mathcal{C}_{\text{op}}$ for which the analysis is possible. Again, although these problems are restrictions of $\mathcal{C}_{\text{op}}$, we do not expect the lower bound in Theorem 3.2 to be improved in either the dependency on $N$ or on $\epsilon$.

**The Case for Quadratic Problems.** Towards this end, let us consider an overparameterized quadratic problem. To proceed, assume that each agent $k$ has data tuple $(A_k, b_k)$ with $N_k$ samples, where $A_k \in \mathbb{R}^{N_k \times D}$ is the feature matrix, and $b_k \in \mathbb{R}^{N_k \times 1}$ is the label. Denote $A := [A_1; \cdots; A_K] \in \mathbb{R}^{N \times D}$, where $N = \sum_{k=1}^{K} N_k$. Suppose $N < D$. Then the generic problem (1) reduces to:

$$f(x) = \frac{1}{2}\|A\theta - b\|^2 = \frac{1}{2}\sum_{k=1}^{K}\|A_k\theta - b_k\|^2. \tag{4}$$

Assuming that each $A_k$ has full row rank, then the above problem is a special case of $\mathcal{C}_{\mathrm{op}}$.

Next, we specialize Alg. 1. Recall that the lower bound derived in Theorem 3.2 is *independent* of $D$, so we cannot directly apply the algorithm we used for the class $\mathcal{C}_{\mathrm{pl}}$, in which each agent transmits a dimension $D$ vector at every iteration. To proceed, let us consider the following *linear compression* scheme, in which we use a matrix $B$ of size $H \times D$ to compress (resp. decompress) the gradients that get transmitted (resp. received), where $H$ is a fixed positive constant satisfying $N \leqslant H \ll D$. In particular, let us specialize the two processing operators in Alg. 1 as: $F(x) = Bx$, $\tilde{F}(y) = B^\top y$. Further, define $\sigma_{\max}(\cdot)$ and $\sigma_{\min}(\cdot)$ (resp. $s_{\max}(\cdot)$ and $s_{\min}(\cdot)$) as the largest and smallest eigenvalues (resp. singular values) for a matrix. Then we have the following result; see Appendix D for proof.

**Theorem 4.2.** *Consider using Alg. 1 to solve the quadratic problem* (4), *with $A$ being full row rank. Choose $B$ such that* $\mathrm{rank}(BA^\top) = N$; *set* $\eta = \frac{\sigma_{\min}Z}{\sigma_{\max}^2(Z)}$; *set $b$ and $C$ according to* (30), *and choose $\tilde{F}(\cdot)$ and $F(\cdot)$ as $F(x) = Bx$, $\tilde{F}(y) = B^\top y$. Then the following holds true:*

$$f(\theta^{t+1}) \leq (1 - 1/2\kappa^4)f(\theta^t),$$

*where $\kappa$ is the condition number of the matrix $Z := AB^\top BA^\top$ given by $\kappa(Z) := \frac{\sigma_{\max}(Z)}{\sigma_{\min}(Z)}$. Further, to compute an $\epsilon$-optimal solution* (3), *each agent is required to transmit:*

$$2b \cdot \kappa^4 \cdot H \log(f(\theta^0)/\epsilon) \quad \textit{bits/agent}. \tag{5}$$

In Theorem 4.2, it is required that matrix $BA^\top$ is full rank, and this can be satisfied with probability 1 if we choose $B$ to be a random matrix. Further, the convergence rate depends on the condition number $\kappa(Z)$. Although we have showed the per iteration communication cost can be reduced to $H$ by linear compression, it is not clear yet whether the dimension $D$ can affect the value of $\kappa(Z)$. In the following, we show $\kappa(Z)$ is independent of $D$; The proof can be found in Appendix E.

**Proposition 1.** *Consider the matrices $A$ and $B$ in Theorem 4.2, and define $Z := AB^\top BA^\top$. Suppose each entry in $B$ satisfies $B_{hj} \overset{i.i.d}{\sim} \mathcal{N}(0,1), \forall\, h \in [H], j \in [D]$. Then the following holds:*
*(1) The condition number $\kappa(Z)$ is independent of $D$.*
*(2) Let $A^\top = V\Sigma W$ be the compact form of SVD, where $V \in \mathbb{R}^{D \times N}$ satisfies $V^TV = \mathbf{I}_N$; $\Sigma \in \mathbb{R}^{N \times N}$ is the diagonal matrix whose diagonal entries are singular values of $A$; and $W \in \mathbb{R}^{N \times N}$ is a unitary matrix. Then for any fixed $A$, we have*

$$\mathbb{P}\left(\kappa(Z) \leqslant \left(\frac{s_{\max}(\Sigma)}{s_{\min}(\Sigma)}\right)^2 \cdot \left(\frac{\sqrt{H} + \sqrt{N} + t}{\sqrt{H} - \sqrt{N} - t}\right)^2\right) \geqslant 1 - e^{-t^2/2}.$$

Proposition 1 implies that, as long as the compression matrix is standard normal, then $\kappa(Z)$ is independent of $D$. This means that the total bits required for each agent, as provided in (8), will only be dependent on $N, H$ and $\log(f(\theta^0)/\epsilon)$, but not on $D$. To further understand the dependency of $\kappa(Z)$ with $N$ and $H$, let us treat the ratio between the largest and smallest singular values as being fixed [1]. Then set $2e^{-t^2/2} = \delta > 0$, we have $t = \sqrt{2\log(\frac{2}{\delta})}$, so the upper bound for $\kappa$ becomes $\kappa(AA^\top) \cdot \frac{\sqrt{H}+\sqrt{N}+\sqrt{2\log(\frac{2}{\delta})}}{\sqrt{H}-\sqrt{N}-\sqrt{2\log(\frac{2}{\delta})}}$. Assuming that $N \geqslant 4t^2, H = 4N$, then we can conclude that $\mathbb{P}(\kappa(Z) \leq 49) \geq 1 - \delta$. Applying this result to (8), we can conclude that with probability $1 - \delta$, the total number of bits transmitted by each agent is $8b \cdot \kappa^4(A^TA) \cdot N\log(f(\theta^0)/\epsilon)$. Similarly as in Theorem 4.1, $b$ is in the order of $\mathcal{O}(\log(D))$ (see Appendix C.). Therefore the total communication complexity of each agent can be reduced to $\mathcal{O}\big(H\log(D)\log(\frac{1}{\epsilon})\big)$. When $H = \Theta(N)$, then the complexity bound matches the lower bound in Theorem 3.2, up to a gap $\mathcal{O}(\log(D))$. Additionally,

---

[1]This is reasonable, since even when analyzing strongly convex distributed problems, the dependency on the condition number is unavoidable; see [Magnússon et al., 2020, Sec. III-C].

we note that by fixing the seed of random number generator, the random matrix $B$ can be generated by each agent locally, without needing to transmit any information.

**Decentralized Algorithm.** We propose a *fully decentralized algorithm* for solving (4) with a communication complexity that approaches the lower bound in Theorem 3.2. Our idea is similar to Alg. 1 that adopts a dimension reduction step for compression, yet we shall encapsulate the `Quantize`, `Communicate`, `Aggregation`, `Broadcast` steps through a decentralized average consensus protocol with support for quantized communication.

In particular, we consider a setting where $K$ agents are represented as nodes on a connected graph $(V, E)$ with a doubly stochastic mixing matrix $W \in \mathbb{R}_+^{K \times K}$. Observe the following fact:

**Fact 1.** *Consider a set of vectors $v_1, \ldots, v_K \in \mathbb{R}^H$ held by $K$ agents on $(V, E)$. For any $\epsilon > 0$, the* `CHOCO-GOSSIP` *protocol [Koloskova et al., 2019] is described as the map:*

$$\bar{v}_1, \ldots, \bar{v}_K = \texttt{CHOCO-GOSSIP}(v_1, \ldots, v_K; \epsilon)$$

*such that $\bar{v}_k$ is the vector available at agent $k$ satisfying $\|\bar{v}_k - K^{-1} \sum_{j=1}^K v_j\|^2 \leq \epsilon$. Let $\xi = \sum_{k=1}^K \|v_k - K^{-1} \sum_{j=1}^K v_j\|^2$ be the initial consensus error, the protocol requires at most*

$$(82/\delta^2 \omega) \log(\xi/\epsilon) \ \text{ rounds of quantized communication,}$$

*where $\delta = 1 - |\lambda_2(W)|$ and $\omega \in (0, 1]$ is controlled by the number of bits in the quantizer. Details about the* `CHOCO-GOSSIP` *protocol can be found in Appendix F.1.*

We replace line 5–line 8 in Alg. 1 by `CHOCO-GOSSIP` to yield the decentralized algorithm:

---

**Algorithm 2** Decentralized Gradient Descent with Compressed Comm. via Linear Compression

---

1: **Input:** step size $\eta$; initial iterate $\theta^0 \in \mathbb{R}^D$.
2: `Initialize:` $\theta_i^0 = \bar{\theta}^0$, $\forall i \in [K]$
3: **for** $t$ **in** $0, \ldots, T - 1$ **do**
4:    `Compute:` Each agent compute the compressed local gradient $g_k^t = F(\nabla f_k(\theta_k^t))$.
5:    `Consensus:` Agents exchange gradients $(\bar{g}_1^t, \ldots, \bar{g}_K^t) = \texttt{CHOCO-GOSSIP}(g_1^t, \ldots, g_K^t; \frac{\bar{\epsilon}}{t+1})$.
6:    `Update:` Each agent updates $\theta_k^{t+1} = \theta_k^t - \eta \tilde{F}(\bar{g}_k^t)$.
7: **end for**
8: **Output:** the last iterate $\theta_i^T$

---

**Theorem 4.3.** *Consider using Alg. 2 to solve the quadratic problem (4), with $A$ being full row rank. Choose $B$ such that $\mathrm{rank}(BA^\top) = N$; set $\eta = \frac{\sigma_{\min}(Z)}{4\sigma_{\max}^2(Z)}$, and choose $\tilde{F}(\cdot)$ and $F(\cdot)$ as $F(x) = Bx$, $\tilde{F}(y) = B^\top y$. For any $T \geq 1$, the following holds true:*

$$f(\theta_k^T) \leq \left(1 - 1/4\kappa^2\right)^T f(\bar{\theta}^0) + \mathcal{O}(\bar{\epsilon}^2), \ k = 1, \ldots, K.$$

*where $\kappa$ was defined in Theorem 4.2. Moreover, suppose that $A, B$ are constructed as in Proposition 1, to compute an $\epsilon$-optimal solution (3), each agent is required to transmit:*

$$\mathcal{O}\left(\frac{\kappa^2 H \log H}{\delta^2} \Big[ \log \frac{1}{\epsilon} + \log \big(1 + \frac{\sqrt{D} + \sqrt{H} + \sqrt{\log 1/\zeta}}{n\sigma_{\max}^2(Z)}\big) \Big] \log \frac{1}{\epsilon}\right) \ bits,$$

*with probability at least $1 - \zeta$.*

Proposition 1 shows that the condition number $\kappa$ is $\mathcal{O}(1)$. To satisfy $\mathrm{rank}(BA^\top) = N$, we need $H \geq N$. Altogether, this yields the communication complexity $\mathcal{O}(N \log(N) \log(D) \log(1/\epsilon)^2)$ to match Theorem 3.2 up to a $\log(D)$ factor.

**The Case of Overparameterized Neural Network.** We now explore the possibility of extending the analysis of quadratic problems to a specific overparameterized neural network, still belonging to class $\mathcal{C}_{\mathrm{op}}$. We will customize Alg. 1 to partially compress the gradients of the loss function. Our idea is to design $F(\cdot)$ (resp. $\tilde{F}(\cdot)$) so that we compress (resp. decompress) the gradient with respect to the parameters in the widest layer to $\mathcal{O}(N)$ bits, while directly quantize the remaining gradients. Note that for a few popular network architectures such as Neural Tangent Kernel (NTK) [Huang

et al., 2021], the widest layer contains most of the parameters $\Omega(N^4)$. Therefore, compressing these parameters into $\mathcal{O}(N)$ bits already represents significant communication . We will leave the design and analysis of more aggressive schemes which compress the parameters of all layers for future work.

To proceed, let us first state the structure of the network. Let $X \in \mathbb{R}^{N \times n_0}$ denote the data sample; let $a(\cdot)$ be an activation function; let $O_l \in \mathbb{R}^{N \times n_l}$ as the output of each layer, which can be expressed as

$$
O_l = \begin{cases} X & l = 0 \\ a\left(O_{l-1}W_l\right) & l \in [L-1] \\ O_{L-1}W_L & l = L \end{cases},
$$

where $W_l \in \mathbb{R}^{n_{l-1} \times n_l}$ is the weight matrix in $l$-th layer. Further, denote $O_{L,k}$ as the output of the network for $k$-th agent. Let $\theta = (W_l)_{l=1}^L$ collect all the weights. Then objective function becomes

$$
f(\theta) = \sum_{k=1}^{K} f_k(\theta) = \frac{1}{2} \sum_{k=1}^{K} \|O_{L,k} - y_k\|^2. \tag{6}
$$

Further, we define the vectorized gradient of each layer as $u_l = \text{vec}(\nabla_{W_l} f(\theta))$ and $u_{l,k}$ for each agent. Now we describe the assumptions on the network.

**Assumption 4.** *(Pyramidal network topology) Let $n_1 \geq N$ and $n_2 \geq n_3 \geq \ldots \geq n_L = 1$.*

**Assumption 5.** *(Activation function) Fix $\nu \in (0,1)$ and $\rho > 0$. Let $a$ satisfy that: $(i) a'(x) \in [\nu, 1]$, $(ii) |a(x)| \leqslant |x|$ for every $x \in \mathbb{R}$, and $(iii) a'$ is $\rho$-Lipschitz.*

As showed in [Nguyen and Mondelli, 2020], with Assumption 4 and 5, objective (6) belongs to $\mathcal{C}_{\text{op}}$. With the above Pyramidal structure, layer $l = 2$ is the widest layer and contains the largest number of parameters, denoted as $u_2 \in \mathbb{R}^{n_1 \times n_2}$. As mentioned above, we will design $F$ and $\tilde{F}$ to compress and decompress $u_2$. To proceed, Let us choose $B = \mathbf{I}_{n_2} \otimes \tilde{B} \in \mathbb{R}^{Hn_2 \times n_1 n_2}$ where $\tilde{B} \in \mathbb{R}^{H \times n_1}$, such that $N \leqslant H \ll n_1$. Then $F$ and $\tilde{F}$ are defined as:

$$
F(u_l) = \begin{cases} 0 & l = 1, \\ Bu_l & l = 2, \\ u_l & l \geqslant 3, \end{cases} \quad \tilde{F}(F(u_l)) = \begin{cases} 0 & l = 1, \\ B^\top Bu_l & l = 2, \\ u_l & l \geqslant 3. \end{cases} \tag{7}
$$

This means that we compress the parameters in the second layer, and perform usual gradient descent for the layers $l \geq 3$; further, the parameters in the first layer is left frozen at its initialization. Note that we freeze the first layer for some technical reasons, but layer freezing itself is a useful technique to accelerate the training in practice, which can provide strong generalization ability [Yosinski et al., 2014, Advani et al., 2020].

Let us denote $O_1^0$ as the output of the first hidden layer at initialization. We have the following result. Its proof as well as the expressions for various constants, can be found in Appendix G.

**Theorem 4.4. (Informal)** *Consider using Alg. 1 to solve the quadratic problem (6), with $X$ being full row rank. Suppose $\theta^0$ is initialized properly (see Appendix G for details). Choose $\tilde{B}$ such as $\text{rank}(\tilde{B}O_1^{0\top}) = N$; choose $\tilde{F}(\cdot)$ and $F(\cdot)$ as in (7). Then there exists stepsize $\eta$, constant $\phi, C, b$ such that to compute an $\epsilon$-optimal solution (6), each agent is required to transmit:*

$$
\frac{4b}{\eta\phi} \cdot (Hn_2 + \sum_{l=2}^{L-1} n_l n_{l+1}) \log(f(\theta^0)/\epsilon) \quad \text{bits/agent}, \tag{8}
$$

*where $\eta, \phi$ are related to $X, \nu, \rho$ and singular values of the weight matrices at initialization.*

Theorem 4.4 shows that if we choose $H = \Theta(N)$, and when $n_1$ is very large, then the total communication complexity can be reduced by a factor of $n_1/H$.

**Preliminary Numerical Experiments.** We conclude by presenting a numerical experiment for the UCI Tom's Hardware[2] dataset using Alg. 2 where we applied $\lfloor \log(t+100) \rfloor$ rounds of communication at the $t$-th iteration for the `CHOCO-GOSSIP` subroutine; see Appendix F.1. We consider a ring network with $K = 5$ agents, each one has 500 or 1000 samples (thus making $N = 2500$, or $N = 5000$).

---

[2]Available: https://archive.ics.uci.edu/ml/datasets/Buzz+in+social+media+

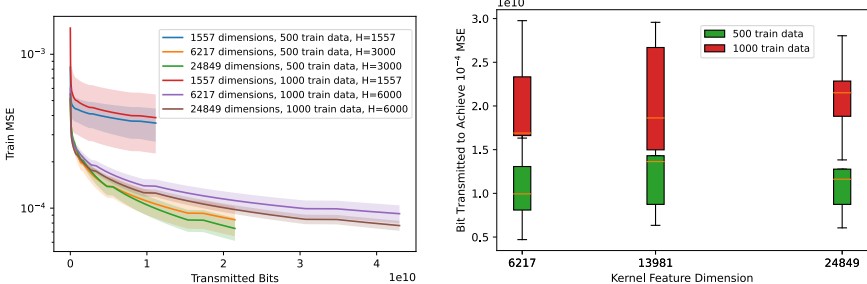

Figure 1: **Overparameterized Kernel Regression with Alg. 2.** Both figures show statistics of 5 random seeded runs. (Left) Training MSE against the number of bit transmitted on the whole network. (Right) Communication cost to achieve $10^{-4}$ MSE on training dataset, compared over different dimensions of kernel features.

We construct $D$-dimensional features from the dataset as NTK features [Bietti and Mairal, 2019]. In Fig. 1, we train a least square regression model in the overparameterized regime. We observe that the communication cost needed is proportional to the number of samples, instead of the feature dimension. Particularly, our results indicate that the communication cost required to reach a target train MSE (of $10^{-4}$) is proportional to the number of training samples.

**Acknowledgments and Disclosure of Funding** We sincerely thank the anonymous reviewers for the suggestions. M. Hong and B. Song are supported by NSF grants CIF-1910385, CMMI-1727757 and CNS-2003033.

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
