# A   Additional Related Literature and Comparison

In the first category, where the commmunication complexities are measured by *rounds* of communications (where in each round real-valued vectors get exchanged), recent works [Kovalev et al., 2021, 2022] provides lower bounds for a sum of smooth and strongly convex functions over time-varying networks, and for strongly monotone variational inequality problems in a stochastic (finite-sum) setting. Other related works include [Scaman et al., 2017, 2018, Arjevani and Shamir, 2015]. In addition, lower bounds for non-convex problems are considered in [Sun and Hong, 2019, Lu and De Sa, 2021]. Further, there are many works that derive rounds of communication upper bounds for decentralized and federated learning algorithms, see, e.g.,[Stich and Karimireddy, 2019, Stich et al., 2018, Patel and Dieuleveut, 2019, Yu et al., 2019, Wang and Joshi, 2018, Gorbunov et al., 2021].

A recent work [Gorbunov et al., 2021] has analyzed the communication efficient algorithm to solve functions that satisfy PL conditions. However, it has been focused on analyzing the number of communication rounds needed to achieve certain $\epsilon$ optimal solution, while the current paper is focused on finding the minimum bits to be communicated. Therefore, although the two works are both about developing communication efficient algorithms for PL functions, the bounds obtained in these works represent different physical quantities, thus cannot be directly compared.

# B   Communication Complexity Lower Bounds

Let us introduce the so-called equality problem Vempala et al. [2020], denoted as EQUAL.

**Definition 5** (Equality Problem). *Consider a set of $K$ agents, each is given an input $c_k \in \{0,1\}^m$; $m$ is the length of the binary input. Then, the* EQUAL *problem is defined as follows:*

$$\mathsf{EQUAL}_m(c_1, \ldots, c_K) = \begin{cases} 1, & \text{if } c_1 = \ldots = c_K \\ 0, & \text{otherwise} \end{cases}$$

*For any deterministic algorithms, the communication complexity lower bound of* $\mathsf{EQUAL}_m(c_1, \ldots, c_K)$ *is* $\Omega(Km)$ *[Vempala et al., 2020, Thm 3.5].*

Further, the intermediate steps required to derive the lower bounds involve packing arguments. Therefore, we provide below a lower bound for the maximum number of points that we can pack into a compact set $[0,1]^n$, such that the distance between each pair of points is at least $\delta$.

**Definition 6** (Packing Problem). *We define the following:*

- *For a given $\delta > 0$ we define the set $S(\delta) \subseteq [0,1]^n$ such that $\|x - y\| > \delta, \forall x, y \in S(\delta)$.*

- *Assuming that $|S(\delta)| \geq 2^m$, we define a function $h : \{0,1\}^m \to S(\delta)$. For $u, v \in \{0,1\}^m$ it holds that $u \neq v \Leftrightarrow h(u) \neq h(v)$.*

**Lemma B.1.** *([Korhonen and Alistarh, 2021, Lemma 2]) For a set $S(\delta) \subseteq [0,1]^n$ defined in Def. 6, it holds that $|S(\delta)| \geq \left(\frac{\sqrt{2n}}{\sqrt{\pi e}\delta}\right)^n$.*

Next, we repeat here for completeness the Assumptions that the local functions in the Distributed PL problem class satisfy (Def. 1).

**Assumption 6.** *The local objective functions satisfy:*

$$2\mu_k \cdot (f_k(\theta) - f_k(\theta^*_{(k)})) \leq \|\nabla f_k(\theta)\|^2, \quad \forall\, \theta, \quad \forall\, k,$$

*where $\theta^*_{(k)}$ is a global minimum of $f_k(\cdot)$; $\mu_k$'s some positive constants.*

**Assumption 7.** *There exists positive constants $L_k$'s and $L$ such that:*

$$\|\nabla f_k(\theta) - \nabla f_k(\theta')\| \leq L_k \|\theta - \theta'\|, \quad \|\nabla f(\theta) - \nabla f(\theta')\| \leq L \|\theta - \theta'\|, \quad \forall\, \theta, \theta', \,\forall\, k.$$

In the Lemma below we show that the function instance we are going to use in Theorem 3.1 satisfies the PL condition and both Assumptions 6 and 7.

**Lemma B.2.** *The function $f_k(\theta) = \frac{1}{2}\|\theta - c\|_2^2 + \sin^2(\theta_k - c_k), c \in \mathbb{R}^D$ (where with $c_k, \theta_k$ we denote the $k$th component of vectors $c, \theta$, respectively) satisfies the PL condition with $\mu = \frac{1}{8}$. That is, it holds that $f_k(\theta) - f_k(\theta^*) \leq \frac{1}{2\mu}\|\nabla f_k(\theta)\|^2, \forall \theta \in \mathbb{R}^D$, where $\theta^*$ is the global minimum of $f_k(\cdot)$.*

**Remark 1.** *The constructed function $f_k(\theta)$ and the respective sum across nodes $\sum_{k=1}^{K} f_k(\theta)$ satisfy Assumptions 6, 7. Specifically, Assumption 6 follows trivially from Lemma B.2. In addition, it can be shown that $f_k(\theta)$ and $\sum_{k=1}^{K} f_k(\theta)$ have bounded Hessians, which implies the Lipschitz gradient property of Assumption 7.*

*Proof of Lemma B.2.* To begin with we are going to show that the (one dimensional) function

$$\widetilde{f}_k(\theta_k) = \frac{1}{2}(\theta_k - c_k)^2 + \sin^2(\theta_k - c_k),$$

where $\theta_k, c_k \in \mathbb{R}$, satisfies the PL condition with $\mu = \frac{1}{8}$; notice that $\widetilde{f}_k(\theta_k)$ has a unique minimum with value 0 attained at $\theta_k = c_k$, and we have that $\widetilde{f}'_k(\theta_k) = \theta_k - c_k + \sin(2(\theta_k - c_k))$. Then, we are going to use this result to prove the PL condition for the function $f_k(\theta)$.

So, let us define the function

$$g(\theta_k) = 4\left[\theta_k - c_k + \sin(2(\theta_k - c_k))\right]^2 - \frac{1}{2}(\theta_k - c_k)^2 - \sin^2(\theta_k - c_k).$$

The gradient of the above function is given by

$$g'(\theta_k) = 8\left[\theta_k - c_k + \sin(2(\theta_k - c_k))\right]\left[1 + 2\cos(2(\theta_k - c_k))\right] - (\theta_k - c_k) - \sin(2(\theta_k - c_k))$$
$$= \left[\theta_k - c_k + \sin(2(\theta_k - c_k))\right]\left[7 + 16\cos(2(\theta_k - c_k))\right].$$

In order to show that $\widetilde{f}_k(\theta_k)$ satisfies the PL property it suffices to prove that $g(\theta_k) \geq 0, \forall \theta_k \in \mathbb{R}$. Then, it will hold that

$$4\left[\theta_k - c_k + \sin(2(\theta_k - c_k))\right]^2 \geq \frac{1}{2}(\theta_k - c_k)^2 - \sin^2(\theta_k - c_k), \forall \theta_k \in \mathbb{R},$$

that is the PL condition of $\widetilde{f}_k(\theta_k)$ will be satisfied.

First, notice that

$$g(\theta_k) = 4(\theta_k - c_k)^2 + 8(\theta_k - c_k)\sin(2(\theta_k - c_k)) + 4\sin^2(2(\theta_k - c_k)) - \frac{1}{2}(\theta_k - c_k)^2 - \sin^2(\theta_k - c_k)$$

$$\geq \frac{7}{2}(\theta_k - c_k)^2 + 8(\theta_k - c_k)\sin(2(\theta_k - c_k)) - 1.$$

It is clear from the above expression that the term $\frac{7}{2}(\theta_k - c_k)^2$ dominates the value of the objective for large values of $|\theta_k - c_k|$. Therefore, the objective does not become unbounded below.

Secondly, consider the stationary points of $g(\theta_k)$. Since the objective does not become unbounded below the only possible global minima of $g(\theta_k)$ are its stationary points. We are going to show that the values of the objective at those points is non-negative, effectively proving that $g(\theta_k) \geq 0, \forall \theta_k \in \mathbb{R}$. The stationary points of $g$ are defined by the following expressions:

- $\sin(2(\theta_k - c_k)) = -(\theta_k - c_k)$

  Notice that in the interval $\theta_k \in (c_k, \frac{\pi}{2} + c_k]$ it holds that $\sin[2(\theta_k - c_k)] \geq 0$ but $-(\theta_k - c_k) < 0$. Similarly, for $\theta_k \in [-\frac{\pi}{2} + c_k, c_k)$ it holds that $\sin[2(\theta_k - c_k)] \leq 0$ but $-(\theta_k - c_k) > 0$. Also, for $\theta_k \notin [-\frac{\pi}{2} + c_k, c_k) \cup (c_k, \frac{\pi}{2} + c_k] \cup \{c_k\}$ it holds that $|\theta_k - c_k| \geq \frac{\pi}{2} > 1$. Therefore, in all the above cases the equation of the stationary points does not have a solution (and thus there are no stationary points). Finally, note that the equation has a trivial solution at $\theta_k = c_k$, which corresponds to the only stationary point we can get from this equation. For that point it holds that $g(c_k) = 0$.

- $\cos(2(\theta_k - c_k)) = -\frac{7}{16}$

  From the above equations, and by using the proper trigonometric identities it follows that $\sin^2(2(\theta_k - c_k)) = \frac{207}{256}$, $\sin^2(\theta_k - c_k) = \frac{23}{32}$ and $\theta_k - c_k = \kappa\pi \pm \frac{1}{2}\arccos\left(-\frac{7}{16}\right) \approx \kappa\pi \pm \frac{2.02}{2}$. Then if we plug the above values into $g(\theta_k)$ we get

$$g(\theta_k) \approx \frac{7}{2}(\kappa\pi \pm 1.01)^2 + 8(\kappa\pi \pm 1.01)\sin[2\kappa\pi \pm 2.02] + 4\frac{207}{256} - \frac{23}{32}$$

$$= \frac{7}{2} (\kappa\pi \pm 1.01)^2 + 8 (\kappa\pi \pm 1.01) \sin [\pm 2.02] + \frac{644}{256}$$

$$= (\kappa\pi \pm 1.01) \left[ \frac{7}{2} (\kappa\pi \pm 1.01) + 8 \cdot (\pm 0.9) \right] + \frac{161}{64}.$$

It can be easily verified that $(\kappa\pi \pm 1.01) \left[ \frac{7}{2} (\kappa\pi \pm 1.01) + 8 \cdot (\pm 0.9) \right] + \frac{161}{64} > 0$ for all $\kappa \in \mathbb{Z}$.

In conclusion for all the stationary points $\hat{\theta}_k$ it holds that $g(\hat{\theta}_k) \geq 0$. As a result, we can claim that $g(\theta_k) \geq 0, \forall \theta_k \in \mathbb{R}$, and the function $\widetilde{f}(\theta_k)$ satisfies the PL property.

Next, consider the PL property (with $\mu = \frac{1}{8}$) for the function $f_k(\theta)$, that is

$$\frac{1}{2} \|\theta - c\|_2^2 + \sin^2 (\theta_k - c_k) \leq 4 \|\theta - c + \sin (2(\theta_k - c_k)) e_k\|^2, \tag{9}$$

where $e_k$ is a vector of all zeros except at index $k$.

We have shown above that

$$\frac{1}{2} (\theta_k - c_k)^2 + \sin^2 (\theta_k - c_k) \leq 4 \left[ \theta_k - c_k + \sin (2(\theta_k - c_k)) \right]^2. \tag{10}$$

Also, it trivially holds that

$$\frac{1}{2} \sum_{i=1, i \neq k}^{D} (\theta_i - c_i)^2 \leq 4 \sum_{i=1, i \neq k}^{D} (\theta_i - c_i)^2. \tag{11}$$

Adding inequalities (10) and (11) we obtain condition (9), which ensures the PL property for the objective $f_k(\theta)$, with $\mu = \frac{1}{8}$. This completes the proof.

$\square$

*Proof of Theorem 3.1.* To begin with, let $\pi \in \Pi(\epsilon)$ be an arbitrary protocol that solves the problems in $\mathcal{C}_{\mathrm{pl}}$. Then, for any input (i.e., for any specific problem within the class $\mathcal{C}_{\mathrm{pl}}$) the protocol returns an $\epsilon$-approximate minimum $\tilde{\theta}$ as given in Def. 3.

Moreover, consider the set $S(\delta) \subseteq [0, 1]^D$ introduced in Def. 6 (where in place of $n$ we have $D$) with $\delta = 2\sqrt{2\epsilon}$. Then, Lemma B.1 implies that

$$|S(\delta)| \geq \left( \frac{\sqrt{2D}}{2\sqrt{2\pi e \epsilon}} \right)^D = \left( \frac{1}{2\sqrt{\pi e}} \right)^D \left( \frac{D}{\epsilon} \right)^{D/2}.$$

Then, we set $m = \Theta (\log_2 |S(\delta)|) = \Theta \left( D \log \left( \frac{D}{\epsilon} \right) \right)$, where the exact value of $m$ is selected such that $|S(\delta)| > 2^m$ holds. As a result, under the assumption that $\frac{D}{\epsilon} = \Omega(1)$, it holds that $|S(\delta)| \geq 2$, and $m \geq 1$.

Next, we will show that protocol $\pi$ also solves the EQUAL problem. That is, we will reduce every instance of $\mathrm{EQUAL}_m(u_1, \ldots, u_K)$ to a problem in $\mathcal{C}_{\mathrm{pl}}$. Towards this end, for an arbitrary input $(u_1, \ldots, u_K) \in \{0, 1\}^{Km}$ we select the following function from $\mathcal{C}_{\mathrm{pl}}$,

$$f(\theta) = \sum_{k=1}^{K} f_k(\theta) \text{ with } f_k(\theta) = \frac{1}{2} \|\theta - h(u_k)\|_2^2 + \sin^2 (\theta_k - (h(u_k))_k), \tag{12}$$

where $h(\cdot)$ is introduced in Def. 6, and $h(u_k) \in S(\delta) \subseteq [0, 1]^D$; $(h(u_k))_k, \theta_k$ denote the $k$th component of $h(u_k)$ and $\theta$, respectively. It is shown in Lemma B.2 that the functions $f_k$'s that correspond to each node satisfy the PL condition, with $\mu_k = \frac{1}{8}$, for all $k$. We can also easily verify that the rest of the conditions in Assumption $6 - 7$ are satisfied.

Then, we have the following cases:

- Case 1 (equal inputs): It holds that $u_1 = \ldots = u_K := u$. Thus, we have that $h(u_1) = \ldots = h(u_K) = h(u)$ and the minimum of (12) is 0, attained at some point $\theta^*$ such that $\theta^* = h(u)$.

  Also, protocol $\pi$ returns an approximate minimum $f(\tilde{\theta})$ of (12), for which it holds that

  $$f(\tilde{\theta}) \overset{(a)}{\leq} f(\theta^*) + \epsilon = \frac{1}{2} \sum_{k=1}^{K} \|\theta^* - h(u)\|_2^2 + \sin^2\left(\theta_k^* - (h(u))_k\right) + \epsilon \overset{(b)}{=} \epsilon,$$

  where in (a) we used (3); and in (b) we exploited the fact that the minimum is 0.

- Case 2 (inputs not equal): There exists a pair of nodes $(i, j)$ such that $u_i \neq u_j$. As a result, $h(u_i) \neq h(u_j)$. Then, protocol $\pi$ returns an approximate minimum $f(\tilde{\theta})$ of (12) for which it holds that

  $$f(\tilde{\theta}) \overset{(a)}{\geq} f(\theta^*)$$
  $$= f_i(\theta^*) + f_j(\theta^*) + \sum_{k=1, k \neq i,j}^{K} f_k(\theta^*)$$
  $$\geq \frac{1}{2}\|\theta^* - h_i(u_i)\|_2^2 + \frac{1}{2}\|\theta^* - h_j(u_j)\|_2^2$$
  $$\overset{(b)}{>} \frac{1}{2}\left(\frac{\delta}{2}\right)^2 = \frac{\delta^2}{8} \overset{(c)}{=} \epsilon,$$

  where in (a) we used (3), in (b) we used the characteristic property (i.e., the minimum distance between two points is $\|x - y\| > \delta, \forall x, y \in S(\delta)$) of set $|S(\delta)|$ (from Def. 6), and in (c) we use the quantity $\delta = 2\sqrt{2\epsilon}$.

From the above analysis we see that if $f(\tilde{\theta}) \leq \epsilon$ then $\mathsf{EQUAL}_m(u_1, \ldots, u_K) = 1$. Otherwise, if we assume that $\mathsf{EQUAL}_m(u_1, \ldots, u_K) = 0$, then the analysis of case 2 implies that $f(\tilde{\theta}) > \epsilon$, a contradiction. Similarly, we can claim that if $f(\tilde{\theta}) > \epsilon$, then $\mathsf{EQUAL}_m(u_1, \ldots, u_K) = 0$. In the opposite case (i.e., if $f(\tilde{\theta}) > \epsilon \implies \mathsf{EQUAL}_m(u_1, \ldots, u_K) = 1$) we see from case 1 that $f(\tilde{\theta}) \leq \epsilon$, that is we reach a contradiction. In summary, we have that

$$\mathsf{EQUAL}_m(u_1, \ldots, u_K) = \begin{cases} 1, & \text{if } f(\tilde{\theta}) \leq \epsilon \\ 0, & \text{if } f(\tilde{\theta}) > \epsilon. \end{cases}$$

Finally, the fact that the communication complexity of $\mathsf{EQUAL}_m(u_1, \ldots, u_K)$ is $\Omega(Km)$, and the above reduction imply that $\Omega(Km) = \Omega(KD \log\left(\frac{D}{\epsilon}\right))$ is a lower bound for the communication complexity of $\mathcal{C}_{\mathrm{pl}}$. $\qquad \square$

*Proof of Theorem 3.2.* To begin with, let $\pi \in \Pi(\epsilon)$ be an arbitrary protocol that solves the problems in $\mathcal{C}_{\mathrm{op}}$. Then, for any input (i.e., for any specific problem within the class $\mathcal{C}_{\mathrm{op}}$) the protocol returns an $\epsilon$-approximate minimum $\tilde{\theta}$, as described in Def. 3.

Moreover, consider the set $S(\delta) \subseteq [0, 1]^N$ introduced in Def. 6 (where in place of $n$ we have $N$) with $\delta = 2\sqrt{2\epsilon}$. Then, Lemma B.1 implies that

$$|S(\delta)| \geq \left(\frac{\sqrt{2N}}{2\sqrt{2\pi e\epsilon}}\right)^N = \left(\frac{1}{2\sqrt{\pi e}}\right)^N \left(\frac{N}{\epsilon}\right)^{N/2}.$$

Then, we set $m = \Theta\left(\log_2 |S(\delta)|\right) = \Theta\left(N \log\left(\frac{N}{\epsilon}\right)\right)$, where the exact value of $m$ is selected such that $|S(\delta)| > 2^m$ holds. As a result, under the assumption that $\frac{N}{\epsilon} = \Omega(1)$, it holds that $|S(\delta)| \geq 2$, and $m \geq 1$.

Now, we are going to show that protocol $\pi$ also solves the $\mathsf{EQUAL}$ problem. That is, we are going to reduce every instance of $\mathsf{EQUAL}_m(u_1, \ldots, u_K)$ to a problem in $\mathcal{C}_{\mathrm{op}}$. To be more precise, for an

arbitrary input $(u_1, \ldots, u_K) \in \{0, 1\}^{Km}$ we select the following instance from $\mathcal{C}_{\text{op}}$,

$$f(\theta) = \sum_{k=1}^{K} f_k(\theta) \text{ with } f_k(\theta) = \frac{1}{2} \|G(\theta) - h(u_k)\|_2^2, \tag{13}$$

where $h(\cdot)$ is introduced in Def. 6; and with $h(u_k) \in S \subseteq [0, 1]^N$.

Then, we have the following cases:

- Case 1 (equal inputs): It holds that $u_1 = \ldots = u_K := u$. Thus, we have that $h(u_1) = \ldots = h(u_K) = h(u)$ and the minimum of (13) is 0, attained at some point $\theta^*$ such that $G(\theta^*) = h(u)$.

  Also, protocol $\pi$ returns an approximate minimum $f(\tilde{\theta})$ of (13) for which it holds that

  $$f(\tilde{\theta}) \overset{(a)}{\leq} f(\theta^*) + \epsilon = \frac{1}{2} \sum_{k=1}^{K} \|G(\theta^*) - h(u)\|_2^2 + \epsilon \overset{(b)}{=} \epsilon,$$

  where in (a) we used (3), and (b) follows from the fact that the minimum of (13) is 0 in this case (i.e., equal inputs).

- Case 2 (inputs not equal): There exists a pair of nodes $(i, j)$ such that $u_i \neq u_j$. As a result, $h(u_i) \neq h(u_j)$. Then, protocol $\pi$ returns an approximate minimum $f(\tilde{\theta})$ of (13) for which it holds that:

$$f(\tilde{\theta}) \overset{(a)}{\geq} f(\theta^*)$$

$$= \frac{1}{2} \|G(\theta^*) - h(u_i)\|_2^2 + \frac{1}{2} \|G(\theta^*) - h(u_j)\|_2^2 + \frac{1}{2} \sum_{k=1, k \neq i,j}^{K} \|G(\theta^*) - h(u_k)\|_2^2$$

$$\geq \frac{1}{2} \|G(\theta^*) - h(u_i)\|_2^2 + \frac{1}{2} \|G(\theta^*) - h(u_j)\|_2^2$$

$$\overset{(b)}{>} \frac{1}{2} \left( \frac{\delta}{2} \right)^2 = \frac{\delta^2}{8} \overset{(c)}{=} \epsilon,$$

where in (a) expression (3) is used, in (b) we used the characteristic property (i.e., the minimum distance between two points is $\|x - y\| > \delta, \forall x, y \in S(\delta)$) of set $|S(\delta)|$ (from Def. 6), and in (c) we use the quantity $\delta = 2\sqrt{2\epsilon}$.

From the above analysis we see that if $f(\tilde{\theta}) \leq \epsilon$ then $\text{EQUAL}_m(u_1, \ldots, u_K) = 1$. Otherwise, if $\text{EQUAL}_m(u_1, \ldots, u_K) = 0$, then the analysis of case 2 implies that $f(\tilde{\theta}) > \epsilon$, a contradiction. Similarly, we can claim that if $f(\tilde{\theta}) > \epsilon$, then $\text{EQUAL}_m(u_1, \ldots, u_K) = 0$. In the opposite case (i.e., if $f(\tilde{\theta}) > \epsilon \implies \text{EQUAL}_m(u_1, \ldots, u_K) = 1$) we see from case 1 that $f(\tilde{\theta}) \leq \epsilon$, that is we reach a contradiction. In summary, we have that

$$\text{EQUAL}_m(u_1, \ldots, u_K) = \begin{cases} 1, & \text{if } f(\tilde{\theta}) \leq \epsilon \\ 0, & \text{if } f(\tilde{\theta}) > \epsilon. \end{cases}$$

Finally, the fact that the communication complexity of $\text{EQUAL}_m(u_1, \ldots, u_K)$ is $\Omega(Km)$, and the above reduction imply that $\Omega(Km) = \Omega(KN \log \left( \frac{N}{\epsilon} \right))$ is a lower bound for the communication complexity of $\mathcal{C}_{\text{op}}$. □

## C   Proof for Theorem 4.1

### C.1   Proof of Lemma 4.2

Let us consider the function

$$\phi(\theta) = f(\theta) - \langle \nabla f(\theta^*), \theta \rangle.$$

Since $\nabla f(\theta^*) = 0$, it is easy to see $\phi(\theta) \geqslant 0$. It can be derived directly that $\phi(\theta^*) = 0$, which means $\phi(\cdot)$ can achieve the minimum at $\theta^*$. So we know $\theta^* \in \mathrm{argmin}\ \phi(\theta)$. Then we have the following inequality holds:

$$\phi(\theta^*) \leqslant \phi\left(\theta - \frac{1}{L}\nabla\phi(\theta)\right)$$

$$\overset{(i)}{=} f(\theta - \frac{1}{L}\nabla\phi(\theta)) \overset{(ii)}{=} f(\theta - \frac{1}{L}\nabla f(\theta))$$

$$\overset{(iii)}{\leqslant} f(\theta) - \frac{1}{2L}\|\nabla f(\theta)\|^2,$$

where (i) performed one step of gradient decent; (ii) uses the fact that $\nabla f(\theta^*) = 0$; (iii) is because the Lipshcitz gradient assumption. Then it follows directly

$$\|\nabla f(\theta)\|^2 \leqslant 2L \cdot f(\theta).$$

## C.2 Proof for Theorem 4.1

First, let us state the sketch of the proof. Denote $g^t := \sum_{k=1}^{K} g_k^t, q^t := \sum_{k=1}^{K} q_k^t$.

**Step1:** We show that the loss function decreases linearly if all the agents update the averaged gradient in one step. That is, the following holds true.

$$f(\theta^t - \eta g^t) \leq (1 - \eta\mu)f(\theta^t).$$

**Step2:** We show by induction that for $t = 1, \cdots$, the following inequalities hold true:

$$(1) f(\theta^t) \leqslant (\alpha)^t f(\theta^0), \text{ where } 0 < \alpha < 1, \tag{14}$$

$$(2) \|g^t - q^t\|_\infty \leqslant \frac{\gamma^{t-1}}{2^b - 1} = \tau\gamma^{t-1}, \text{ for some } \gamma^{t-1} > 0, \tag{15}$$

where $\{\gamma^t\}_{t=1}^{\infty}$ is a sequence of positive numbers.

The proof of **Step 1** is straightforward:

$$f(\theta^t - \eta g^t) = f(\theta^t - \eta \sum_{k=1}^{K} g_k^t)$$

$$\overset{(i)}{\leqslant} f(\theta^t) - \langle \nabla f(\theta^t), \eta \sum_{k=1}^{K} g_k^t \rangle + \frac{\eta^2 L}{2}\left\|\sum_{k=1}^{K} g_k^t\right\|^2$$

$$\overset{(ii)}{=} f(\theta^t) - \eta\|g^t\|^2 + \frac{\eta^2 L}{2}\|g^t\|^2$$

$$\overset{(iii)}{=} f(\theta^t) - \eta\|g^t\|^2 + \frac{\eta}{4L^2}\|g^t\|^2 \qquad\qquad (\eta = \frac{1}{2L^{\frac{3}{2}}})$$

$$\overset{(iv)}{\leqslant} f(\theta^t) - \frac{1}{2}\eta\|g^t\|^2$$

$$\overset{(v)}{\leqslant} f(\theta^t) - \eta\mu f(\theta^t)$$

$$= (1 - \eta\mu)f(\theta^t), \tag{16}$$

where $(i)$ is by Decent Lemma; (ii) uses the definition of $g^t$; (iii) uses the choice of $\eta$; (iv) is because $L > 1$; (v) uses Assumption 1.

Now we prove **Step 2**. To begin with, let us set $\eta = \frac{1}{2L^{\frac{3}{2}}}$, and set:

$$\alpha = 1 - \frac{\mu}{8L^{\frac{3}{2}}}, \ \gamma^t = \sqrt{(\alpha)^{t+1}f(\theta^0)}, \ \tau = \frac{1}{\sqrt{C}D}, \tag{17}$$

$$C = \max\left(\sqrt{\frac{16L}{\mu^2} + \frac{L^{\frac{3}{2}}}{\mu}}, 100\right), \ b = \max\left(\log(\frac{1}{\tau} + 1), b_0\right).$$

First, let us verify that (14) holds true when $t = 1$. We have the following series of inequalities:

$$f(\theta^1) = f(\theta^0 - \eta q^0) - f(\theta^0 - \eta g^0) + f(\theta^0 - \eta g^0)$$

$$\overset{(i)}{\leqslant} \eta\langle\nabla f(\theta^0 - \eta g^0), q^0 - g^0\rangle + \frac{\eta^2 L}{2}\|q^0 - g^0\|^2 + f(\theta^0 - \eta g^0)$$

$$\overset{(ii)}{\leqslant} \frac{\eta}{2\beta}\|\nabla f(\theta^0 - \eta g^0)\|^2 + 2\eta\beta\|q^0 - g^0\|^2 + \frac{\eta^2 L}{2}\|q^0 - g^0\|^2 + f(\theta^0 - \eta g^0)$$

$$\overset{(iii)}{\leqslant} \frac{\eta L}{\beta}f(\theta^0 - \eta g^0) + (2\eta\beta + \frac{\eta^2 L}{2})\|q^0 - g^0\|^2 + f(\theta^0 - \eta g^0)$$

$$\overset{(iv)}{=} (\frac{\eta L}{\beta} + 1)(1 - \eta\mu)f(\theta^0) + (2\eta\beta + \frac{\eta^2 L}{2})\|q^0 - g^0\|^2$$

$$\overset{(v)}{\leqslant} (\frac{\eta L}{\beta} + 1)(1 - \eta\mu)f(\theta^0) + (2\eta\beta + \frac{\eta^2 L}{2})D^2\|q^0 - g^0\|_\infty^2$$

$$\overset{(vi)}{\leqslant} (\frac{\eta L}{\beta} + 1)(1 - \eta\mu)f(\theta^0) + (2\eta\beta + \frac{\eta^2 L}{2})D^2(\sum_{k=1}^{K}\|q_k^0 - g_k^0\|_\infty)^2$$

$$\overset{(vii)}{\leqslant} (\frac{\eta L}{\beta} + 1)(1 - \eta\mu)f(\theta^0) + (2\eta\beta + \frac{\eta^2 L}{2})D^2\sum_{k=1}^{K}K\|q_k^0 - g_k^0\|_\infty^2$$

$$\overset{(viii)}{\leqslant} (\frac{\eta L}{\beta} + 1)(1 - \eta\mu)f(\theta^0) + (2\eta\beta + \frac{\eta^2 L}{2})\frac{f(\theta^0)}{C^2 D^4}$$

$$\overset{(ix)}{=} (1 - \frac{1}{2}\eta\mu)f(\theta^0) + (\frac{4\eta(1 - \eta\mu)L}{\mu} + \frac{\eta^2 L}{2})\frac{f(\theta^0)}{C^2 D^4} \qquad\qquad \beta = \frac{2(1 - \eta\mu)L}{\mu}$$

$$= (1 - \frac{1}{2}\eta\mu + (\frac{4\eta(1 - \eta\mu)L}{\mu} + \frac{\eta^2 L}{2})/C^2 D^4)f(\theta^0)$$

where (i) comes from the Decent Lemma; (ii) uses the Young's inequality with constant $\beta$; (iii) uses Assumption 1; (iv) is from (16); (v) uses the relationship between $\ell_2$ and $\ell_\infty$ norm; (vi) uses the triangle inequality; (vii) uses the Cauchy-Schwartz inequality; (viii) uses the initialization condition $\|q_k^0 - g_k^0\|_\infty \leqslant \frac{\sqrt{f_k(\theta^0)}}{CD^3\sqrt{K}}$; (ix) uses the choice of $\beta$.

Now let us define the quantization at initialization. For each entry in $q_k^0$, denoted as $(g_k^0)_j$, we consider the interval $\big[\lfloor(g_k^0)_j\rfloor, \lceil(g_k^0)_j\rceil\big]$, which is constructed by the closest integers. We use $b_0$ bits to make the grid, and quantize each element of the vector by the closest point on the grid. It is clear that the quantization error for each element is at most $\frac{1}{2^{b_0}-1}$. So we set

$$\frac{1}{2^{b_0} - 1} = \frac{\sqrt{f(\theta^0)}}{CD^3\sqrt{K}},$$

or equivalently, $b_0 = \log\big(\frac{CD^3\sqrt{K}}{\sqrt{f(\theta^0)}} + 1\big)$.

Now since we have chosen $C \geqslant \frac{16L}{\mu^2} + \frac{L^{\frac{3}{2}}}{\mu}$, we can bound the coefficient in front of $f(\theta^0)$ as:

$$1 - \frac{1}{2}\eta\mu + (\frac{4\eta(1 - \eta\mu)L}{\mu} + \frac{\eta^2 L}{2})/C^2 D^4$$

$$\overset{(i)}{\leqslant} 1 - \frac{1}{2}\eta\mu + (\frac{2(1 - \eta\mu)}{\sqrt{L}\mu} + \frac{1}{8L^2})/C^2$$

$$\overset{(ii)}{\leqslant} 1 - \frac{1}{2}\eta\mu + (\frac{2}{\sqrt{L}\mu} + \frac{1}{8})/C^2$$

$$\overset{(iii)}{\leqslant} 1 - \frac{1}{4}\eta\mu \qquad\qquad\qquad (18)$$

where in (i) we plugged in the choice of $\eta$ and $D \geqslant 1$; (ii) is because we have assumed that $L > 1$ and $D \geqslant 1$; (iii) uses the choice of $C$ in (17). It follows $f(\theta^1) \leqslant (1 - \frac{\mu}{8L^{\frac{3}{2}}})f(\theta^0)$. Thus, (14) holds for $t = 1$.

Second, let us analyze (15) for $t = 1$. Observe that:

$$\|g^1 - q^0\|_\infty \leqslant \|g^1 - g^0\| + \|g^0 - q^0\|_\infty \overset{(i)}{\leqslant} \eta L \|q^0\| + \|g^0 - q^0\|_\infty$$

$$\overset{(ii)}{\leqslant} \eta L \|g^0\| + \eta L \|q^0 - g^0\| + \|q^0 - g^0\|_\infty$$

$$\overset{(iii)}{\leqslant} \eta L \|g^0\| + (1 + \eta L D)\|q^0 - g^0\|_\infty$$

$$\overset{(iv)}{\leqslant} \eta L \|g^0\| + (1 + \eta L D)\sum_{k=1}^{K} \frac{\sqrt{f_k(\theta^0)}}{CD^3\sqrt{K}}$$

$$\overset{(v)}{\leqslant} \eta L \sqrt{2L \cdot f(\theta^0)} + (1 + \eta L D)\frac{\sqrt{f(\theta^0)}}{CD^3}$$

$$\overset{(vi)}{\leqslant} \frac{\sqrt{2}}{2}\sqrt{f(\theta^0)} + 2 \cdot \frac{\sqrt{f(\theta^0)}}{100} \overset{(vii)}{\leqslant} \sqrt{\alpha f(\theta^0)},$$

where (i) uses Assumption 2; (ii) is from triangle inequality; (iii) uses the relationship between $\ell_2$ and $\ell_\infty$ norm; (iv) uses the condition $\|q_k^0 - g_k^0\|_\infty \leqslant \frac{\sqrt{f_k(\theta^0)}}{CD^3\sqrt{K}}$; (v) uses Assumption 1 and the Cauchy-Schwartz inequality; (vi) plugs in the choice of stepsize and the fact that $C \geqslant 100$; (v) compares the left side with definition of $\alpha$ in (17). By Lemma 4.1, it follows that:

$$\|q^1 - g^1\|_\infty = \|\text{quant}(g^1, q^0, r, b) - g^1\|_\infty \leqslant \frac{\gamma^0}{2^b - 1}.$$

Next, let us use induction to prove (14) and (15). First, we analyze the decent of $f(\theta^t)$:

$$f(\theta^{t+1}) = f(\theta^t - \eta q^t) - f(\theta^t - \eta g^t) + f(\theta^t - \eta g^t)$$

$$\overset{(i)}{\leqslant} \eta\langle \nabla f(\theta^t - \eta g^t), q^t - g^t \rangle + \frac{\eta^2 L}{2}\|q^t - g^t\|^2 + f(\theta^t - \eta g^t)$$

$$\overset{(ii)}{\leqslant} \frac{\eta}{2\beta}\|\nabla f(\theta^t - \eta g^t)\|^2 + 2\eta\beta\|q^t - g^t\|^2 + \frac{\eta^2 L}{2}\|q^t - g^t\|^2 + f(\theta^t - \eta g^t)$$

$$\overset{(iii)}{\leqslant} \frac{\eta L}{\beta}f(\theta^t - \eta g^t) + (2\eta\beta + \frac{\eta^2 L}{2})\|q^t - g^t\|^2 + f(\theta^t - \eta g^t)$$

$$\overset{(iv)}{=} (\frac{\eta L}{\beta} + 1)(1 - \eta\mu)f(\theta^t) + (2\eta\beta + \frac{\eta^2 L}{2})\tau^2(\gamma^{t-1})^2$$

$$\overset{(v)}{\leqslant} (\frac{\eta L}{\beta} + 1)(1 - \eta\mu)(\alpha)^t f(\theta^0) + (2\eta\beta + \frac{\eta^2 L}{2})\tau^2(\gamma^{t-1})^2$$

$$\overset{(vi)}{\leqslant} (\frac{\eta L}{\beta} + 1)(1 - \eta\mu)(\alpha)^t f(\theta^0) + (2\eta\beta + \frac{\eta^2 L}{2})\tau^2(\alpha)^t f(\theta^0)$$

$$= \left(1 - \frac{1}{2}\eta\mu + \left(\frac{4\eta(1 - \eta\mu)L}{\mu} + \frac{\eta^2 L}{2}\right)\tau^2\right)(\alpha)^t f(\theta^0), \tag{19}$$

where (i) comes from the Decent Lemma; (ii) uses the Young's inequality with constant $\beta$; (iii) uses Assumption 1; (iv) is from (16) and induction assumption (15); (v) uses the induction assumption (14); (vi) uses the definition of $\gamma^{t-1}$. Finally, set $\beta = \frac{2(1 - \eta\mu)L}{\mu}$, we will get the last equality. Next, let us analyze the coefficient in the expression (19):

$$1 - \frac{1}{2}\eta\mu + \left(\frac{4\eta(1 - \eta\mu)L}{\mu} + \frac{\eta^2 L}{2}\right)\tau^2 \overset{(i)}{=} 1 - \frac{1}{2}\eta\mu + \left(\frac{2(1 - \eta\mu)}{\sqrt{L}\mu} + \frac{1}{8L^2}\right)\tau^2$$

$$\overset{(ii)}{\leqslant} 1 - \frac{1}{2}\eta\mu + \left(\frac{2}{\sqrt{L}\mu} + \frac{1}{8}\right)\tau^2$$

$$\overset{(iii)}{=} 1 - \frac{\mu}{4L^{\frac{3}{2}}} + \left(\frac{2}{\sqrt{L}\mu} + \frac{1}{8}\right)\tau^2$$

$$\overset{(iv)}{\leqslant} 1 - \frac{\mu}{8L^{\frac{3}{2}}} = \alpha, \tag{20}$$

where in (i) we plugged in the choice of $\eta$; (ii) is because we have assumed that $L > 1$ ; (iii) plugged in the choice of stepsize; (iv) uses the definition of $\tau$ in (17); and the last equality comes from the definition of $\alpha$ in (17). Plugging (20) to (19), we obtain $f(\theta^{t+1}) \leqslant (\alpha)^{t+1} f(\theta^0)$.

Second, we show that (15) holds for $t + 1$. We have:

$$
\begin{aligned}
\left\|g^{t+1} - q^t\right\|_\infty &\leqslant \left\|g^{t+1} - g^t\right\| + \left\|g^t - q^t\right\| \\
&\overset{(i)}{\leqslant} L\eta\|q^t\| + \tau\gamma^{t-1} \\
&\overset{(ii)}{\leqslant} L\eta(\|g^t\| + D\|g^t - q^t\|_\infty) + \tau\gamma^{t-1} \\
&\overset{(iii)}{\leqslant} L\eta\sqrt{2L}\sqrt{f(\theta^t)} + (1 + DL\eta)\tau\gamma^{t-1} \\
&\overset{(iv)}{\leqslant} L\eta\sqrt{L}\sqrt{(\alpha)^t f(\theta^0)} + (1 + DL\eta)\tau\gamma^{t-1} \\
&\overset{(v)}{=} (L\eta\sqrt{2L} + (1 + DL\eta)\tau)\gamma^{t-1} \\
&= (\frac{\sqrt{2}}{2} + (1 + \frac{D}{2})\tau)\gamma^{t-1} \\
&\overset{(vi)}{\leqslant} (\frac{\sqrt{2}}{2} + \frac{3}{2}D\tau)\gamma^{t-1} \overset{(vii)}{\leqslant} \frac{9}{10}\gamma^{t-1} \overset{(viii)}{\leqslant} \sqrt{\alpha}\gamma^t = \gamma^{t+1}
\end{aligned}
$$

where (i) uses Assumption 2 and the induction assumption; (ii) uses the triangle inequality to decompose $\|q^t\|$ and uses the relationship between $\ell_\infty$ and $\ell_2$ ; (iii) is from Assumption 1 and induction assumption (15); (iv) uses the induction assumption (14); (v) uses the definition of $\gamma^t$; (vi) is because $\mu < 1$; (vii) comes from the fact that $\tau < \frac{1}{10D}$ in (17); (viii) compares the choice of $\alpha$ in (17). Thus, we obtain $\left\|g^{t+1} - q^t\right\|_\infty \leqslant \sqrt{(\alpha)^{t+1} f(\theta^0)} = \gamma^t$. Then by Lemma 4.1, we have (15) holds for $t + 1$.

Now we have proved by induction that (14) and (15) hold. So for $t > 0$, there is

$$
f(\theta^t) \leqslant (\alpha)^t f(\theta^0), \text{ where } \alpha = 1 - \frac{\mu}{8L^{\frac{3}{2}}}.
$$

Thus, to compute an $\epsilon$-optimal solution, the total number of iterations required is $\log(f(\theta^0)/\epsilon)/\log(1 - \mu/8L^{\frac{3}{2}})$. Since in each iteration, each agent $k$ transmits a length-$D$ vector $q_k^t$, it follows that the total number of bits each agent needs to communicate is $D\log(f(\theta^0)/\epsilon)/\log(1/(1 - \mu/8L^{\frac{3}{2}}))$bits. Notice that $\log(1/(1 - \mu/8L^{\frac{3}{2}}) = -\log(1 - \mu/8L^{\frac{3}{2}}) \sim 8L^{\frac{3}{2}}/\mu$, so we can derive the simplified total number of bits as $bD \cdot \frac{8L^{\frac{3}{2}}}{\mu} \log \left(f\left(\theta^0\right)/\epsilon\right)$.

## D   The Proof of Theorem 4.2

First, let us provide the sketch of the proof. Denote

$$
g^t := \sum_{k=1}^K g_k^t, \quad q^t := \sum_{k=1}^K q_k^t, \quad \tilde{g}^t := B^\top g^t.
$$

Using the above notation, the agents' local update step (i.e., the 'Update' step in Alg. 1) can be expressed as:

$$
\theta_k^{t+1} = \theta_k^t - \eta B^\top q^t, \forall\, k. \tag{21}
$$

**Step 1:** We show that the loss function decreases linearly if all the agents update parameters using the direction $\tilde{g}^t$, as follows:

$$
f(\theta^t - \eta\tilde{g}^t) \leq \left(1 - \frac{1}{\kappa^2}\right) f(\theta^t).
$$

**Step 2:** Let $\tau = \frac{1}{2^b - 1}$, we show by induction that for $t = 1, \cdots$, the following inequalities hold true:

(1) $f(\theta^t) \leqslant (\alpha)^t f(\theta^0)$, for some $0 < \alpha < 1$. \tag{22}

(2) $\|g^t - q^t\|_\infty \leqslant \tau\gamma^{t-1}$, $\|\tilde{g}^t - B^\top q^t\|_\infty \leqslant \tau H\|B^\top\|_\infty \gamma^{t-1}$, for some $\gamma^{t-1} > 0$, $\qquad$ (23)

where $\{\gamma^t\}_{t=1}^\infty$ is a sequence of positive numbers.

We prove **Step 1** first. At $t$-the iteration, Let us first expand the objective function as:

$$f\left(\theta^t - \eta\tilde{g}^t\right) = \frac{1}{2}\|A(\theta^t - \eta\tilde{g}^t) - b\|^2 = f\left(\theta^t\right) - \langle A\theta^t - b, \eta A\tilde{g}^t\rangle + \frac{1}{2}\eta^2\left\|A\tilde{g}^t\right\|^2. \qquad (24)$$

To proceed, let us provide explicit expressions for $g^t$ and $\tilde{g}^t$:

$$g^t = B\nabla f(\theta^t) = BA^\top(A\theta^t - b) \qquad (25)$$

$$\tilde{g}^t = B^\top B\nabla f(\theta^t) = B^\top BA^\top(A\theta^t - b). \qquad (26)$$

By using the above, the inner product in (24) can be bounded as follows

$$-\langle A\theta^t - b, \eta A\tilde{g}^t\rangle = -\eta\langle A\theta^t - b, AB^\top BA^\top(A\theta^t - b)\rangle \leqslant -\eta\sigma_{\min}(Z)\left\|A\theta^t - b\right\|^2. \qquad (27)$$

Using the above relations, we can further bound the descent of the objective function as

$$f\left(\theta^t - \eta\tilde{g}^t\right) \overset{(i)}{\leqslant} f(\theta^t) - \eta\sigma_{\min}(Z)\left\|A\theta^t - b\right\|^2 + \frac{1}{2}\eta^2\left\|AB^\top BA^\top\left(A\theta^t - b\right)\right\|^2$$

$$\overset{(ii)}{\leqslant} f(\theta^t) - \eta\sigma_{\min}(Z)\left\|A\theta^t - b\right\|^2 + \frac{1}{2}\eta^2\sigma_{\max}^2(Z)\left\|A\theta^t - b\right\|^2$$

$$\overset{(iii)}{=} f(\theta^t) - 2\eta\sigma_{\min}(Z)f(\theta^t) + \eta^2\sigma_{\max}^2(Z)f(\theta^t)$$

$$= (1 - \eta\sigma_{\min}(Z))f(\theta^t)$$

$$= (1 - \frac{1}{\kappa^2})f(\theta^t), \qquad (28)$$

where (i) comes from plugging (27) and (26) into (24); (ii) extracts the largest eigenvalue of $Z$; (iii) is due to the definition of the objective function; the last two equalities hold due to the definition $\eta = \frac{\sigma_{\min}(Z)}{\sigma_{\max}^2(Z)}$.

Next, we prove **Step 2**. To begin with, let us define:

$$\alpha := 1 - \frac{1}{2\kappa^4}, \quad \lambda := 6\sqrt{2}\cdot\frac{\sigma_{\min}(Z)}{\sqrt{\sigma_{\max}(Z)}}, \quad \gamma^t := \lambda\sqrt{(\alpha)^{t+1}f(\theta^0)}. \qquad (29)$$

Further let us set

$$b = \max\left(\log_2\left(\frac{1}{\tau} + 1\right), b_0\right), \quad C = \max(\frac{1}{\lambda^2\tau^2 D^2}, \frac{12}{\lambda}) \quad \text{with} \quad \tau \quad \text{given by} \qquad (30)$$

$$\tau := \min\left(\frac{1}{\sqrt{2\kappa^4\lambda^2 D^2 H^2\|B^\top\|_\infty^2 s_{\max}^2(A)\left(\frac{\sigma_{\min}^2(Z)}{2\sigma_{\max}^4(Z)} + \frac{1}{\sigma_{\max}^2(Z)}\right)}}, \frac{1}{6(1 + \frac{H}{\kappa})}\right).$$

First, we analyze (22) for $t = 1$. We have the following relations:

$$f(\theta^1) = f(\theta^0 - \eta B^\top q^0) - f(\theta^0 - \eta\tilde{g}^0) + f(\theta^0 - \eta\tilde{g}^0)$$

$$\overset{(i)}{\leqslant} \frac{1}{2}\|\eta A(B^\top q^0 - \tilde{g}^0)\|^2 + \langle\eta A(B^\top q^0 - \tilde{g}^0), A(\theta^0 - \eta\tilde{g}^0) - b\rangle + (1 - \eta\sigma_{\min}(Z))f(\theta^0)$$

$$\overset{(ii)}{\leqslant} \frac{1}{2}\|\eta A(B^\top q^0 - \tilde{g}^0)\|^2 + \eta\left(\beta\|A(B^\top q^0 - \tilde{g}^0)\|^2 + \frac{1}{2\beta}\|A(\theta^0 - \eta\tilde{g}^0) - b\|^2\right) + (1 - \eta\sigma_{\min}(Z))f(\theta^0)$$

$$\overset{(iii)}{\leqslant} (\frac{1}{2}\eta^2 + \beta\eta)s_{\max}^2(A)\|B^\top q^0 - \tilde{g}^0\|^2 + \frac{\eta}{\beta}(1 - \eta\sigma_{\min}(Z))f(\theta^0) + (1 - \eta\sigma_{\min}(Z))f(\theta^0)$$

$$= (\frac{1}{2}\eta^2 + \beta\eta)s_{\max}^2(A)\|B^\top q^0 - \tilde{g}^0\|^2 + (1 + \frac{\eta}{\beta})(1 - \eta\sigma_{\min}(Z))f(\theta^0)$$

$$\overset{(iv)}{=} (\frac{1}{2}\eta^2 + \frac{\eta}{\sigma_{\min}(Z)})s_{\max}^2(A)\|B^\top q^t - \tilde{g}^0\|^2 + (1 - \eta^2\sigma_{\min}^2(Z))f(\theta^0)$$

$$\overset{(v)}{\leqslant} D^2(\frac{1}{2}\eta^2 + \frac{\eta}{\sigma_{\min}(Z)})s_{\max}^2(A)\|B^\top q^0 - \tilde{g}^0\|_\infty^2 + (1 - \eta^2\sigma_{\min}^2(Z))f(\theta^0)$$

$$\overset{(vi)}{\leqslant} D^2 H^2\|B^\top\|_\infty^2(\frac{1}{2}\eta^2 + \frac{\eta}{\sigma_{\min}(Z)})s_{\max}^2(A)\|q^0 - g^0\|_\infty^2 + (1 - \eta^2\sigma_{\min}^2(Z))f(\theta^0)$$

$$\overset{(vii)}{\leqslant} KD^2 H^2\|B^\top\|_\infty^2(\frac{1}{2}\eta^2 + \frac{\eta}{\sigma_{\min}(Z)})s_{\max}^2(A)\sum_{k=1}^{K}\|q_k^0 - g_k^0\|_\infty^2 + (1 - \eta^2\sigma_{\min}^2(Z))f(\theta^0)$$

$$\overset{(viii)}{\leqslant} H^2\|B^\top\|_\infty^2(\frac{1}{2}\eta^2 + \frac{\eta}{\sigma_{\min}(Z)})s_{\max}^2(A)\frac{f(\theta^0)}{C^2 D^4} + (1 - \eta^2\sigma_{\min}^2(Z))f(\theta^0)$$

$$= \left(1 - \eta^2\sigma_{\min}^2(Z) + H^2\|B^\top\|_\infty^2(\frac{1}{2}\eta^2 + \frac{\eta}{\sigma_{\min}(Z)})\frac{s_{\max}^2(A)}{C^2 D^4}\right)f(\theta^0)$$

$$\overset{(ix)}{=} \left(1 - \frac{1}{\kappa^4} + H^2\|B^\top\|_\infty^2\left(\frac{\sigma_{\min}^2(Z)}{2\sigma_{\max}^4(Z)} + \frac{1}{\sigma_{\max}^2(Z)}\right)s_{\max}^2(A)\lambda^2\tau^2/D^2\right)f(\theta^0)$$

$$\overset{(x)}{\leqslant} (1 - \frac{1}{2\kappa^4})f(\theta^0) = \alpha f(\theta^0),$$

where (i) explicitly expands the $f(\cdot)$ function, and uses (28); (ii) applies the Young inequality with constant $\beta$; (iii) extracts the largest singular value of $A$, and uses (28); (iv) set $\beta = \frac{1}{\sigma_{\min}(Z)}$; (v) uses the relationship between $\ell_2$ and $\ell_\infty$ norm; (vi) uses the fact that $\|B^\top q^0 - \tilde{g}^0\| \leqslant H^2\|\tilde{B}\|_\infty\|q^0 - g^0\|_\infty$; (vii) uses the Cauchy-Schwartz inequality; (viii) uses the condition $\|q_k^0 - g_k^0\|_\infty \leqslant \frac{\sqrt{f(\theta^0)}}{CD\sqrt{K}}$ in Algorithm 1; (vii) uses the induction assumption (22) and the definition of $\gamma^{t-1}$ in (29); (viii) plug in the choice of stepsize; (ix) comes from the choice of $\lambda$ and $\tau$. So we have showed (22) holds for $t = 1$; $(ix)$ plug in the choice of stepsize and constant $C \geqslant \frac{1}{\lambda^2\tau^2}$; (x) uses the choice of $\lambda$ and $\tau$.

Next, let us analyze (23) for $t = 1$. The idea is that, if we can show that $\|g^1 - q^0\|_\infty \leqslant \gamma^0$, then we will be able to use Lemma 4.1 to show (23). More specifically, we have:

$$\|q^1 - g^1\| \overset{(i)}{=} \|\text{quant}(g^1, q^0, \gamma^0, b) - g^1\|_\infty \overset{(ii)}{\leqslant} \tau\gamma^0,$$

$$\|B^\top q^1 - \tilde{g}^1\|_\infty \overset{(iii)}{=} \|B^\top \text{quant}(g^1, q^0, \gamma^0, b) - B^\top g^1\|_\infty \overset{(iv)}{\leqslant} \tau H\|B^\top\|_\infty\gamma^0,$$

where $(i)$ and $(iii)$ come from the 'Quantize' step in Algorithm 1; $(ii)$ and $(iv)$ are from the two inequalities in Lemma 4.1 (assuming that $\|g^1 - q^0\|_\infty \leqslant \gamma^0$ holds).

Next, we show $\|g^1 - q^0\|_\infty \leq \gamma^0$. We observe that:

$$\|g^1 - q^0\|_\infty \overset{(i)}{\leqslant} \|g^1 - g^0\|_\infty + \|g^0 - q^0\|_\infty$$

$$\overset{(ii)}{\leqslant} \eta\|BA^\top AB^\top q^0\| + \|g^0 - q^0\|_\infty$$

$$\overset{(iii)}{\leqslant} \eta\sigma_{\max}(Z)\|q^0\| + \|g^0 - q^0\|_\infty$$

$$\overset{(iv)}{\leqslant} \eta\sigma_{\max}(Z)(\|g^0\| + \|q^0 - g^0\|) + \|g^0 - q^0\|_\infty$$

$$\overset{(v)}{\leqslant} \eta\sigma_{\max}(Z)(\|BA^\top(A\theta^0 - b)\| + H\|q^0 - g^0\|_\infty) + \|g^0 - q^0\|_\infty$$

$$\overset{(vi)}{\leqslant} \eta\sigma_{\max}^{\frac{3}{2}}(Z)\|A\theta^0 - b\| + (1 + \eta H\sigma_{\max}(Z))\|g^0 - q^0\|_\infty$$

$$\overset{(vii)}{\leqslant} \eta\sigma_{\max}^{\frac{3}{2}}(Z)\|A\theta^0 - b\| + (1 + \eta H\sigma_{\max}(Z))\sum_{k=1}^{K}\|g_k^0 - q_k^0\|_\infty$$

$$\overset{(viii)}{\leqslant} \eta\sigma_{\max}^{\frac{3}{2}}(Z)\sqrt{2f(\theta^0)} + (1 + \eta H\sigma_{\max}(Z))\sum_{k=1}^{K}\frac{\sqrt{f_k(\theta^0)}}{CD^3\sqrt{K}}$$

$$\overset{(ix)}{\leqslant} \eta\sigma_{\max}^{\frac{3}{2}}(Z)\sqrt{2f(\theta^0)} + (1 + \eta H\sigma_{\max}(Z))\frac{\sqrt{f(\theta^0)}}{CD^3}$$

$$\overset{(x)}{=} \sqrt{f(\theta^0)}\left(\frac{1}{CD^3} + \frac{H}{CD^3\kappa} + \frac{\sqrt{2}\sigma_{\min}(Z)}{\sqrt{\sigma_{\max}(Z)}}\right)$$

$$\overset{(xi)}{\leqslant} \lambda\sqrt{f(\theta^0)}\left(\frac{2}{\lambda C} + \frac{1}{6}\right) \overset{(xii)}{\leqslant} \frac{1}{3}\lambda\sqrt{f(\theta^0)} \leqslant \lambda\sqrt{\alpha f(\theta^0)} = \gamma^0$$

where (i) is due to the triangle inequality; (ii) expands the expression of $g^1$ and $g^0$ in (25), uses the relation between $\ell_2$ norm and $\ell_\infty$ norm and uses the update rule (21); (iii) uses the fact that non-zero eigen values of $BA^\top AB^\top$ and $Z$ are the same and extracts the largest eigen value of $Z$; (iv) uses triangle inequality ; (v) uses the relationship between $\ell_2$ and $\ell_\infty$ norm; (vi) is because $s_{\max}^2(BA^\top) = \sigma_{\max}(Z)$ and extract the largest singular value of $BA^\top$; (vii) uses triangle inequality; (viii) uses the initial condition $\|q_k^0 - g_k^0\|_\infty \leqslant \frac{\sqrt{f_k(\theta^0)}}{C \cdot D\sqrt{K}}$ in Algorithm 1; (ix) uses Cauchy-Schwartz inequality; (x) plug in the choice of stepsize; (xi) is because $\kappa \geqslant 1, H \leqslant D, D \geqslant 1$ and the choice of $\lambda$; (xii) is from $C \geqslant \frac{12}{\lambda}$; the last inequality comes from $\sqrt{\alpha} \geqslant \frac{\sqrt{2}}{2} > \frac{1}{3}$ since $\kappa \geqslant 1$. So we can show $\|g^1 - q^0\|_\infty \leq \gamma^0$.

Next, we will show (22) holds for $t + 1$ by induction, based on the base assumption that (22) and (23) holds for $t$. We have the following series of relations:

$$f(\theta^{t+1}) = f(\theta^t - \eta B^\top q^t) - f(\theta^t - \eta\tilde{g}^t) + f(\theta^t - \eta\tilde{g}^t)$$

$$\overset{(i)}{\leqslant} \frac{1}{2}\|\eta A(B^\top q^t - \tilde{g}^t)\|^2 + \langle\eta A(B^\top q^t - \tilde{g}^t), A(\theta^t - \eta\tilde{g}^t) - b\rangle + (1 - \eta\sigma_{\min}(Z))f(\theta^t)$$

$$\overset{(ii)}{\leqslant} \frac{1}{2}\|\eta A(B^\top q^t - \tilde{g}^t)\|^2 + \eta\left(\beta\|A(B^\top q^t - \tilde{g}^t)\|^2 + \frac{1}{2\beta}\|A(\theta^t - \eta\tilde{g}^t) - b\|^2\right) + (1 - \eta\sigma_{\min}(Z))f(\theta^t)$$

$$\overset{(iii)}{\leqslant} (\frac{1}{2}\eta^2 + \beta\eta)s_{\max}^2(A)\|B^\top q^t - \tilde{g}^t\|^2 + \frac{\eta}{\beta}(1 - \eta\sigma_{\min}(Z))f(\theta^t) + (1 - \eta\sigma_{\min}(Z))f(\theta^t)$$

$$= (\frac{1}{2}\eta^2 + \beta\eta)s_{\max}^2(A)\|B^\top q^t - \tilde{g}^t\|^2 + (1 + \frac{\eta}{\beta})(1 - \eta\sigma_{\min}(Z))f(\theta^t)$$

$$\overset{(iv)}{=} (\frac{1}{2}\eta^2 + \frac{\eta}{\sigma_{\min}(Z)})s_{\max}^2(A)\|B^\top q^t - \tilde{g}^t\|^2 + (1 - \eta^2\sigma_{\min}^2(Z))f(\theta^t)$$

$$\overset{(v)}{\leqslant} D^2(\frac{1}{2}\eta^2 + \frac{\eta}{\sigma_{\min}(Z)})s_{\max}^2(A)\|B^\top q^t - \tilde{g}^t\|_\infty^2 + (1 - \eta^2\sigma_{\min}^2(Z))f(\theta^t)$$

$$\overset{(vi)}{\leqslant} D^2 H^2\|B^\top\|_\infty^2(\frac{1}{2}\eta^2 + \frac{\eta}{\sigma_{\min}(Z)})s_{\max}^2(A)(\tau\gamma^{t-1})^2 + (1 - \eta^2\sigma_{\min}^2(Z))f(\theta^t)$$

$$\overset{(vii)}{\leqslant} \left(1 - \eta^2\sigma_{\min}^2(Z) + D^2 H^2\|B^\top\|_\infty^2\left(\frac{1}{2}\eta^2 + \frac{\eta}{\sigma_{\min}(Z)}\right)s_{\max}^2(A)\lambda^2\tau^2\right)(\alpha)^t f(\theta^0)$$

$$\overset{(viii)}{=} \left(1 - \frac{1}{\kappa^4} + D^2 H^2\|B^\top\|_\infty^2\left(\frac{\sigma_{\min}^2(Z)}{2\sigma_{\max}^4(Z)} + \frac{1}{\sigma_{\max}^2(Z)}\right)s_{\max}^2(A)\lambda^2\tau^2\right)(\alpha)^t f(\theta^0)$$

$$\overset{(ix)}{\leqslant} (1 - \frac{1}{2\kappa^4})(\alpha)^t f(\theta^0) = (\alpha)^{t+1} f(\theta^0),$$

where (i) we have explicitly expands the $f(\cdot)$ function, and have used (28); (ii) applies the Young inequality with constant $\beta$; (iii) extracts the largest singular value of $A$, and uses (28); (iv) set $\beta = \frac{1}{\sigma_{\min}(Z)}$; (v) uses the relationship between $\ell_2$ and $\ell_\infty$ norm; (vi) uses the second inequality in induction assumption (23); (vii) uses the induction assumption (22) and the definition of $\gamma^{t-1}$ in (29); (viii) plug in the choice of stepsize; (ix) comes from the choice of $\lambda$ and $\tau$.

Finally, we will show (23) holds for $t + 1$ by induction, again base assumptions (22) holds for $t$ and (23) holds for $t$. We have:

$$\|g^{t+1} - q^t\|_\infty \overset{(i)}{\leqslant} \|g^{t+1} - g^t\|_\infty + \|g^t - q^t\|_\infty$$

$$\overset{(ii)}{\leqslant} \|BA^\top A(\theta^{t+1} - \theta^t)\| + \tau\gamma^{t-1}$$

$$\overset{(iii)}{=} \eta\|BA^\top AB^\top q^t\| + \tau\gamma^{t-1}$$

$$\overset{(iv)}{\leqslant} \eta\sigma_{\max}(Z)\|q^t\| + \tau\gamma^{t-1}$$

$$\overset{(v)}{\leqslant} \eta\sigma_{\max}(Z)(\|q^t - g^t\| + \|g^t\|) + \tau\gamma^{t-1}$$

$$\overset{(vi)}{\leqslant} \eta\sigma_{\max}(Z)(H\|q^t - g^t\|_\infty + \|g^t\|) + \tau\gamma^{t-1}$$

$$\overset{(vii)}{\leqslant} \tau\gamma^{t-1}(1 + \eta H\sigma_{\max}(Z)) + \eta\sigma_{\max}(Z)\|g^t\|$$

$$= \tau\gamma^{t-1}(1 + \eta H\sigma_{\max}(Z)) + \eta\sigma_{\max}(Z)\|BA^\top(A\theta^t - b)\|$$

$$\overset{(viii)}{\leqslant} \tau\gamma^{t-1}(1 + \eta H\sigma_{\max}(Z)) + \eta\sigma_{\max}^{\frac{3}{2}}(Z)\sqrt{2f(\theta^t)}$$

$$\overset{(ix)}{=} \tau\gamma^{t-1}(1 + \frac{H}{\kappa}) + \frac{\sqrt{2}\sigma_{\min}(Z)}{\sqrt{\sigma_{\max}(Z)}}\sqrt{f(\theta^t)}$$

$$\overset{(x)}{\leqslant} \left(\tau\lambda(1 + \frac{H}{\kappa}) + \frac{\sqrt{2}\sigma_{\min}(Z)}{\sqrt{\sigma_{\max}(Z)}}\right)\sqrt{(\alpha)^t f(\theta^0)}$$

$$= \lambda\left(\tau(1 + \frac{H}{\kappa}) + \frac{\sqrt{2}\sigma_{\min}(Z)}{\lambda\sqrt{\sigma_{\max}(Z)}})\right)\sqrt{(\alpha)^t f(\theta^0)}$$

$$\overset{(xi)}{\leqslant} \frac{1}{3}\lambda\sqrt{(\alpha)^t f(\theta^0)} \overset{(xii)}{\leqslant} \lambda\sqrt{(\alpha)^{t+1} f(\theta^0)} = \gamma^t,$$

where (i) uses the triangle inequality; (ii) expands the first term by (25), uses the relationship between $\ell_2$ and $\ell_\infty$ norm, and and uses the first inequality in induction assumption (23); (iii) plug in the update of parameter: $\theta^{t+1} = \theta^t - \eta B^\top q^t$; (iv) comes from the fact that non-zero singular values of $Z$ and $BA^\top AB^\top$ are the same and extracts the largest singular value of $Z$; (v) uses the triangle inequality; (vi) is due to the relationship between $\ell_2$ and $\ell_\infty$ norm; (vii) uses the first inequality in induction assumption (23); (viii) uses the fact that $s_{\max}^2(BA^\top) = \sigma_{\max}(Z)$ and extracts $s_{\max}(BA^\top)$; (ix) plug in the choice of stepsize; (x) uses the definition of $\gamma^{t-1}$ in (29) and induction assumption in (22); (xi) is because the choice of $\tau$ and $\lambda$; (xii) is because $\sqrt{\alpha} = \sqrt{1 - \frac{1}{2\kappa^2}} \geqslant \frac{\sqrt{2}}{2} > \frac{1}{3}$ since $\kappa \geqslant 1$. Thus, we obtain $\left\|g^{t+1} - q^t\right\|_\infty \leqslant \lambda\sqrt{(\alpha)^{t+1} f(\theta^0)} = \gamma^t$. Then by Lemma 4.1, with the correspondence that $c = g^{t+1}, p = q^t, r = \gamma^t$, we can obtain

$$\|q^{t+1} - g^{t+1}\| \overset{(i)}{=} \|\text{quant}(g^{t+1}, q^t, \gamma^t, b) - g^{t+1}\|_\infty \overset{(ii)}{\leqslant} \tau\gamma^t,$$

$$\|B^\top q^{t+1} - \tilde{g}^{t+1}\|_\infty \overset{(iii)}{=} \|B^\top\text{quant}(g^{t+1}, q^t, \gamma^0, b) - B^\top g^{t+1}\|_\infty \overset{(iv)}{\leqslant} \tau D\|B^\top\|_\infty\gamma^t,$$

where $(i)$ and $(iii)$ come from the 'Quantize' step in Algorithm 1; $(ii)$ and $(iv)$ are from the two relations in Lemma 4.1 (since we have proved that $\|g^{t+1} - q^t\|_\infty \leqslant \gamma^t$ holds).

Now we have proved that (14) and (15) hold. So for $t > 0$, there is

$$f(\theta^t) \leqslant (\alpha)^t f(\theta^0), \quad \text{where} \quad \alpha = 1 - \frac{1}{2\kappa^4}.$$

Thus, if we want the objective function to compute an $\epsilon$-optimal solution, the total number of iterations is $\log(f(\theta^0)/\epsilon)/\log(1/(1 - \frac{1}{2\kappa^4}))$. Since in each iteration, each agent $k$ transmits a length-$H$ vector $q_k^t$, so we conclude that the total number of bits each node needs to communicate is $\log(f(\theta^0)/\epsilon)/\log(1/(1 - \frac{1}{2\kappa^4}))$ bits. Notice that $\log(1/(1 - \frac{1}{2\kappa^4})) = -\log(1 - \frac{1}{2\kappa^4}) \sim 2\kappa^4$, so we can derive the simplified total number of bits as $2\kappa^4 \cdot \log(f(\theta^0)/\epsilon)$.

## E   Proof for Proposition 1

Now we prove Proposition 1. We first state two lemmas that will be used.

**Lemma E.1.** *Rudelson and Vershynin [2010] Let $X$ be a $H \times N$ matrix whose entries are independent standard normal random variables. Then*

$$\mathbb{P}\left(\sqrt{H} - \sqrt{N} - t \leq s_{\min}(X) \leq s_{\max}(X) \leq \sqrt{H} + \sqrt{N} + t\right) \geq 1 - 2e^{-t^2/2}, \quad t \geq 0.$$

**Lemma E.2.** *Suppose $A_i^\top \in \mathbb{R}^D$ follows $N(\mu, \Sigma)$. If $\Sigma$ is a diagonal matrix, then each element of $A_i$ is independent.*

We first write down the SVD decomposition of $A^\top$:

$$A^\top = V\Sigma W,$$

where $V \in \mathbb{R}^{D \times N}, \Sigma \in \mathbb{R}^{N \times N}, W \in \mathbb{R}^{N \times N}$. Since $A$ is full rank, denote $\hat{\kappa} := \frac{s_{\max}(A^\top)}{s_{\min}(A^\top)} > 0$. Consider each entry in $BA^\top = BV\Sigma W$. It is clear that each entry of $BV$ follows normal distribution because $(BV)_{hi}$ is linear combination of variables that are standard normal. Then we aim to show that each entry in $BV$ follows standard normal distribution. Notice that $\text{vec}(BV) = (B_1 V, B_2 V, \cdots, B_H V)^\top \in \mathbb{R}^{HN}$, where $B_h$ is each row of $B$. Denote $U_{ij} = \mathbf{Cov}(B_i V, B_j V) \in \mathbb{R}^{N \times N}$, we can obtain that

$$\mathbb{E}[\text{vec}(BV)] = (\mathbb{E}[B_1]V; \mathbb{E}[B_2]V; \cdots, \mathbb{E}[B_H]V)^\top = \mathbf{0},$$

$$\mathbf{Cov}(\text{vec}(BV)) = \begin{pmatrix} U_{11} & U_{12} & U_{13} & \cdots & U_{1N} \\ U_{21} & U_{22} & U_{23} & \cdots & U_{2N} \\ \vdots & \vdots & \vdots & \ddots & \vdots \\ U_{N1} & U_{N2} & U_{N3} & \cdots & U_{NN} \end{pmatrix}$$

Notice that $U_{ij} = \mathbf{Cov}(X^\top B_i^\top, X^\top B_j^\top) = X^\top \mathbf{Cov}(B_i^\top, B_j^\top)X$. Since $B_i^\top$ and $B_j^\top$ are independent, so $V_{ij} = \mathbf{0}, i \neq j$. Since $X^\top X = \mathbf{I}_N$, we have $U_{ij} = \mathbf{Cov}(B_i^\top, B_i^\top) = \mathbf{I}_N$ for $i = 1, 2, \cdots, N$. Then we obtain $\mathbf{Cov}(\text{vec}(BV)) = \mathbf{I}_{HN}$. From Lemma E.2, each entry in $BV$ are independent. By Lemma E.1, we know the condition number of $BV$, which is $\kappa(BV)$ is independent of $D$, and we have

$$\mathbb{P}\left( \kappa(BV) \leqslant \frac{\sqrt{H} + \sqrt{N} + t}{\sqrt{H} - \sqrt{N} - t} \right) \geqslant 1 - 2e^{-t^2/2}, \quad t \geqslant 0.$$

Now we consider $Z = AB^\top BA^\top$. Notice the non-zero eigen values of $Z$ are the same as $BA^\top AB^\top$, then we consider $BA^\top AB^\top$,

$$BA^\top AB^\top = BV\Sigma WW^\top \Sigma^\top V^\top B^\top = BV\Sigma^2 V^\top B^\top.$$

Notice the non-zero eigen values of $BV\Sigma^2 V^\top B^\top$ are the same as $\Sigma^2 BVV^\top B^\top$, we can derive

$$\kappa(Z) = \kappa(\Sigma^2 BVV^\top B^\top) \leqslant \hat{\kappa}^2 \cdot \kappa^2(BV),$$

where $\hat{\kappa}$ is condition number of $A$ and the inequality is because the property of square and invertible matrix. So we have with probability at least $1 - 2e^{-t^2/2}$,

$$\kappa(Z) \leqslant \hat{\kappa}^2 \left( \frac{\sqrt{H} + \sqrt{N} + t}{\sqrt{H} - \sqrt{N} - t} \right)^2.$$

# F The Proof of Theorem 4.3 and Details about Algorithm 2

## F.1 The CHOCO-GOSSIP Protocol

For completeness, we describe the CHOCO-GOSSIP protocol [Koloskova et al., 2019] for decentralized average consensus with compressed communication as follows. Notice the number of gossip rounds $T_g$ is dependent to the error $\epsilon$ as discussed in Fact 1. Note that the protocol uses the compressor $Q(\cdot)$ for compressed communication, for example, this can be the randomized quantizer, see [Koloskova et al., 2019] for other examples. The protocol is summarized below:

---
**Algorithm 3** CHOCO-GOSSIP
---
**Input:** step size $\gamma$; initial vectors $g_1^0, \ldots, g_K^0$; gossip rounds $T_g$; compressor $Q(.)$; mixing matrix $W$; neighbor sets $\mathcal{N}_1, \ldots, \mathcal{N}_K$.
`Initialize:` $\hat{g}_i^0 = \mathbf{0}$, $\forall i \in [K]$.
**for** $t$ **in** $0, \ldots, T_g - 1$ **do**
  `Compress:` Each agent $i$ compress the difference, $q_i^t = Q(g_i^t - \hat{g}_i^t)$
  `Communicate:` Each agent $i$ receives $q_j^t$ from neighbor $j \in \mathcal{N}_i$ and update $\hat{g}_j^{t+1} = \hat{g}_j^t + q_j^t$
  `Aggregation:` Each agent $i$ combines received vectors, i.e.,

$$g_i^{t+1} = g_i^t + \gamma \sum_{j \in \mathcal{N}_i} w_{ij}(\hat{g}_j^{t+1} - \hat{g}_i^{t+1})$$

**end for**
**Output:** $g_i^{T_g}$ at each agent $i$.

---

To derive the number of bits required in Theorem 4.3, we focus on using the random quantizer for $Q(.)$, i.e., for $x \in \mathbb{R}^d$, $s \in \mathbb{N}_+$ and $\tau = (1 + \min\{d/s^2, \sqrt{d}/s\})$, we have

$$Q(x) = \frac{\text{sign}(x) \cdot \|x\|}{s\tau} \cdot \left\lfloor s\frac{|x|}{\|x\|} + \xi \right\rfloor,$$

where $\xi \sim \text{Uniform}[0,1]^d$ and sending $Q(x)$ across the network requires $d\log(s+1) + d + 64$ bits of communication.

## F.2 The Proof of Theorem 4.3

Since $F(x) = Bx$, $\tilde{F}(y) = B^\top y$, the consensus error can be expressed as $E_i^t := \bar{g}_i^t - \frac{1}{n}\sum_{j=1}^n B\nabla f_j(\theta_j^t)$. We also denote the deviation of locally computed gradient as $\Delta G_i^t := \frac{1}{n}\sum_{j=1}^n B(\nabla f_j(\theta_j^t) - \nabla f_j(\theta_i^t))$. Note that $\|E_i^t\| \le \bar{\epsilon}/(t+1)$. We first observe the updated iterate,

$$\theta_i^{t+1} = \theta_i^t - \eta B^\top \bar{g}_i^t - \frac{1}{n}\sum_{j=1}^n B^\top B\nabla f_j(\theta_j^t) + \frac{1}{n}\sum_{j=1}^n B^\top B\nabla f_j(\theta_j^t)$$

$$= \theta_i^t - \eta\left[B^\top E_i^t + \frac{1}{n}B^\top B\sum_{j=1}^n\left(\nabla f_j(\theta_j^t) + \nabla f_j(\theta_i^t) - \nabla f_j(\theta_i^t)\right)\right]$$

$$= \theta_i^t - \eta\left[B^\top E_i^t + B^\top \Delta G_i^t + B^\top B\nabla f(\theta_i^t)\right]$$

Next, we observe that the objective function value evolves as,

$$f(\theta_i^{t+1}) = \frac{1}{2}\sum_{j=1}^n \|A_j\theta_i^{t+1} - b_j\|^2$$

$$= \frac{1}{2}\sum_{j=1}^n\left\{\|A_j\theta_i^t - b_j\|^2 - 2\eta\left\langle A_j^\top(A_j\theta_i^t - b_j)\,\big|\,B^\top E_i^t + B^\top \Delta G_i^t + B^\top B\nabla f(\theta_i^t)\right\rangle\right.$$

$$+ \eta^2 \|A_j \left[ B^\top E_i^t + B^\top \Delta G_i^t + B^\top B \nabla f(\theta_i^t) \right]\|^2 \Big\}$$

$$\leq f(\theta_i^t) - \eta \left\langle \nabla f(\theta_i^t) \mid B^\top B \nabla f(\theta_i^t) \right\rangle + \frac{1}{2} \sum_{j=1}^n 2\eta^2 \|A_j B^\top B \nabla f(\theta_i^t)\|^2$$

$$+ \frac{1}{2} \sum_{j=1}^n \Big\{ -2\eta \left\langle A_j \theta_i^t - \boldsymbol{b}_j \mid A_j B^\top (E_i^t + \Delta G_i^t) \right\rangle + 2\eta^2 \|A_j B^\top (E_i^t + \Delta G_i^t)\|^2 \Big\}$$

$$\leq (1 - 2\sigma_{\min}(Z)\eta + 2\sigma_{\max}^2(Z)\eta^2) f(\theta_i^t)$$

$$+ \sum_{j=1}^n \Big\{ -\eta \left\langle A_j \theta_i^t - \boldsymbol{b}_j \mid A_j B^\top (E_i^t + \Delta G_i^t) \right\rangle + \eta^2 \|A_j B^\top (E_i^t + \Delta G_i^t)\|^2 \Big\} \quad (31)$$

where the last inequality requires the observations

$$\eta^2 \sum_{j=1}^n \|A_j B^\top B \nabla f(\theta_i^t)\|^2 = \eta^2 (A\theta_i^t - \boldsymbol{b})^\top (AB^\top BA^\top)^\top AB^\top BA^\top (A\theta_i^t - \boldsymbol{b}) \leq 2\sigma_{\max}^2(Z)\eta^2 f(\theta_i^t),$$

$$\left\langle \nabla f(\theta_i^t) \mid B^\top B \nabla f(\theta_i^t) \right\rangle \geq \sigma_{\min}(Z) f(\theta_i^t).$$

It remains to deal with the error terms separately. Observe that

$$\sum_{j=1}^n \eta \left\langle A_j \theta_i^t - b_j \mid A_j B^\top (E_i^t + \Delta G_i^t) \right\rangle$$

$$\leq \eta \sum_{j=1}^n \left\{ \frac{\sigma_{\min}(Z)}{2} \|A_j \theta_i^t - b_j\|^2 + \frac{1}{2\sigma_{\min}(Z)} \|A_j B^\top (E_i^t + \Delta G_i^t)\|^2 \right\}$$

$$= \sigma_{\min}(Z)\eta f(\theta_i^t) + \frac{\eta}{2\sigma_{\min}(Z)} \sum_{j=1}^n \|A_j B^\top (E_i^t + \Delta G_i^t)\|^2,$$

and

$$\sum_{j=1}^n \|A_j B^\top (E_i^t + \Delta G_i^t)\|^2$$

$$= \|AB^\top (E_i^t + \Delta G_i^t)\|^2 = (E_i^t + \Delta G_i^t)^\top BA^\top AB^\top (E_i^t + \Delta G_i^t)$$

$$\leq \sigma_{\max}(Z) \|E_i^t + \Delta G_i^t\|^2 \leq 2\sigma_{\max}(Z)(\|E_i^t\|^2 + \|\Delta G_i^t\|^2)$$

Putting together into (31) and setting $\eta = \frac{\sigma_{\min}(Z)}{4\sigma_{\max}^2(Z)}$ yields

$$f(\theta_i^{t+1}) \leq \left( 1 - \frac{\sigma_{\min}(Z)}{8\sigma_{\max}(Z)} \right) f(\theta_i^t) + (\eta^2 + \frac{\eta}{2\sigma_{\min}(Z)}) \cdot 2\sigma_{\max}(Z)(\|E_i^t\|^2 + \|\Delta G_i^t\|^2). \quad (32)$$

To bound $\|\Delta G_i^t\|^2$, we observe that by the algorithm and the initial condition $\theta_i^0 = \theta_j^0$, for any $t \geq 0$,

$$\theta_j^{t+1} - \theta_i^{t+1} = \theta_j^t - \theta_i^t - \eta(E_j^t - E_i^t) = -\eta \sum_{s=0}^t (E_j^s - E_i^s)$$

$$\|\theta_j^{t+1} - \theta_i^{t+1}\|^2 \leq 2\eta^2 \sum_{s=0}^t [\|E_j^s\|^2 + \|E_i^s\|^2] \leq 4\eta^2 \sum_{s=0}^t (\frac{\bar{\epsilon}}{s+1})^2 \leq \frac{2\eta^2 \pi^2 \bar{\epsilon}^2}{3}, \quad (33)$$

where the last inequality applied $\sum_{s=1}^\infty s^{-2} = \pi^2/6$, together with Fact 1 and the assumption $\epsilon_t = \bar{\epsilon}/(t+1)$. Then, the error $\Delta G_i^t$ can be bounded by

$$\|\Delta G_i^t\|^2 = \frac{1}{n^2} \|\sum_{j=1}^n BA_j^\top A_j (\theta_j^t - \theta_i^t)\|^2$$

$$\leq \frac{1}{n^2} \sum_{j=1}^n \|BA_j^\top A_j\|^2 \|\theta_j^t - \theta_i^t\|^2 \overset{(33)}{\leq} \frac{2\eta^2 \pi^2 \bar{\epsilon}^2}{3n^2} \sum_{j=1}^n \|BA_j^\top A_j\|^2. \quad (34)$$

With the notation $\sigma_{BA^\top A} := \frac{1}{n}\sum_{j=1}^{n}\|BA_j^\top A_j\|^2$, (32) gives us

$$f(\theta_i^{t+1}) \overset{(34)}{\le} (1-\frac{1}{8\kappa^2})f(\theta_i^t) + \left(\eta^2 + \frac{\eta}{2\sigma_{\min}(Z)}\right)\cdot 2\sigma_{\max}^2(Z)\left(\epsilon_t^2 + \frac{2\eta^2\pi^2\bar{\epsilon}^2}{3n}\sigma_{BA^\top A}\right)$$

$$\le (1-\frac{1}{8\kappa^2})^{t+1}f(\theta_i^0) + \sum_{s=0}^{t}\left(1-\frac{\sigma_{\min}(Z)\eta}{2}\right)^{t-s}\left(\eta^2 + \frac{\eta}{2\sigma_{\min}(Z)}\right)\cdot 2\sigma_{\max}^2(Z)\left(\epsilon_s^2 + \frac{2\eta^2\pi^2\bar{\epsilon}^2}{3n}\sigma_{BA^\top A}\right)$$

$$\le (1-\frac{1}{8\kappa^2})^{t+1}f(\theta_i^0) + \frac{1}{6\sigma_{\min}^2(Z)}\left(1+\frac{\sigma_{\min}^2(Z)}{4n\sigma_{\max}^4(Z)}\sigma_{BA^\top A}\right)(\sigma_{\min}^2(Z)+2\sigma_{\max}^2(Z))\pi^2\bar{\epsilon}^2,$$

where we have simplified notations by setting $\kappa = \frac{\sigma_{\max}(Z)}{\sigma_{\min}(Z)}$. By adjusting $\bar{\epsilon}$, we can achieve dimension-independent linear convergence.

**Communication Complexity**  Let $\xi_t = \sum_{i=1}^{n}\|g_i^t - \frac{1}{n}\sum_{j=1}^{n}g_j^t\|^2$. We recall from Fact 1 that achieving a consensus error of $\bar{\epsilon}/(t+1)$ at iteration $t$ via CHOCO-GOSSIP requires

$$\frac{82}{\delta^2\omega}\log\frac{(t+1)\xi_t}{\bar{\epsilon}}\quad\text{rounds of communication.}$$

Applying $b$-bits random quantization to our projected $H$-dimensional vector, we have

$$\frac{1}{\omega} = 1 + \min\{H/(2^b-1)^2, \sqrt{H}/(2^b-1)\}.$$

Fix $\epsilon > 0$, to find an $\epsilon$-optimal solution, we set

$$\bar{\epsilon} := \sqrt{\epsilon\frac{3}{\pi^2(1+\kappa^2)}\left(1+\frac{1}{4n\kappa^2\sigma_{\max}^2(Z)}\sigma_{BA^\top A}\right)^{-1}}$$

and $t \ge T_\epsilon := 8\kappa^2\log(2f(\theta_i^0)/\epsilon)$. Altogether, the number of communication rounds for achieving an $\epsilon$ optimal solution is bounded by:

$$\frac{82}{\delta^2\omega}\sum_{t=1}^{T_\epsilon}\log\frac{(t+1)\xi_t}{\bar{\epsilon}} \le \frac{82T_\epsilon}{\delta^2\omega}\left(\max_{t\in[T_\epsilon]}\log(\xi_t) + \log(T_\epsilon+1) + \log(1/\bar{\epsilon})\right)$$

$$= \mathcal{O}\left(\frac{\kappa^2}{\delta^2\omega}\left(\log\frac{1}{\epsilon} + \log(1+\frac{1}{n\sigma_{\max}^2(Z)}\sigma_{BA^\top A})\right)\log\frac{1}{\epsilon}\right)$$

where we have assumed that $\max_{t\in[T_\epsilon]}\log(\xi_t)$ is dominated by $\max\{\log(1/\bar{\epsilon}), \log(T_\epsilon)\}$.

Each communication round requires to send $\mathcal{O}((b+1)H)$ bits per agent. We can optimize the choice of $b$ by

$$b^\star = \arg\min_b\ bH\omega^{-1} = \arg\min_b\left\{\min\left\{bH + bH^2/(2^b-1)^2, bH + bH^{3/2}/(2^b-1)\right\}\right\}$$

$$\Rightarrow b^\star H\omega^{-1} = \mathcal{O}(H\log H),\quad\text{with } b^\star = \log(H^{1/2}+1)$$

Under the properties of $A, B$ described in Proposition 1, by Lemma E.1, simplifying $\|BA_j^\top A_j\|$ shows that with probability $1-\zeta$,

$$\|BA_j^\top A_j\| \le \|A_j\|^2\|B\| \le \left(\sqrt{H} + \sqrt{D} + \sqrt{2\log\frac{2}{\zeta}}\right)^3$$

As such, we have $\sigma_{BA^\top A} \le \left(\sqrt{H} + \sqrt{D} + \sqrt{2\log\frac{2}{\zeta}}\right)^6$. Putting together gives the communication complexity upper bound of $\mathcal{C}_{\mathrm{op}}$ and the proof is completed.

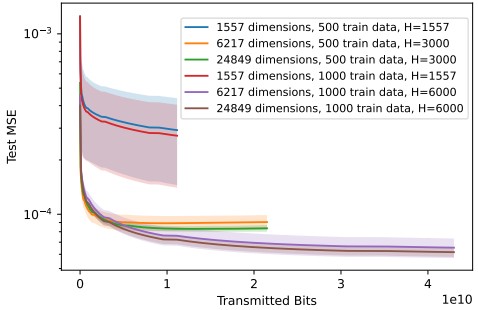

| $D$ | $H$ | **Compression Ratio** |
|---|---|---|
| 1557 | 1557 | 1.099 |
| 6217 | 3000 | 0.529 |
| 6217 | 6000 | 1.057 |
| 24849 | 3000 | 0.132 |
| 24849 | 6000 | 0.264 |

Figure 2: **Overparameterized Kernel Regression with Alg. 2.** Testing MSE against the number of bit transmitted on the whole network, averaged over 5 random seeded runs.

Table 1: **Communication Compression Ratio of Alg. 2** against vanilla DGD with double precision. 8-bits quantization is applied in `CHOCO-GOSSIP` and the average rounds of gossip $T_g$ is 7.78.

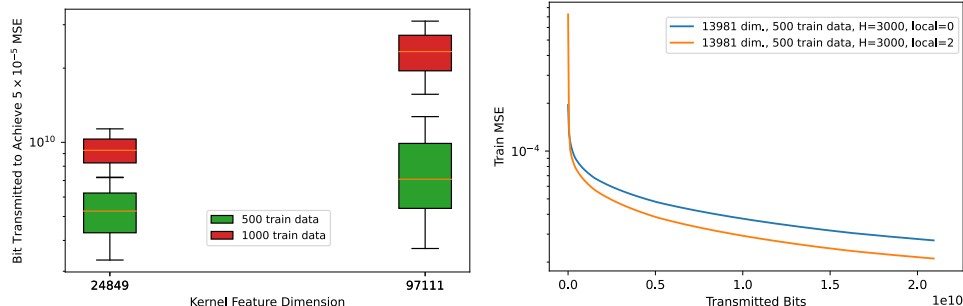

Figure 3: **Overparameterized Kernel Regression with Alg. 2.** (Left) Additional simulation results for large scale overparameterization (19×). (Right) Additional simulation results of Alg. 2 combining with local updates, compared to when no local updates are applied.

### F.3 Additional Numerical Result

Fig. 2 shows the test MSE against communication cost for the cases of $N = 2500$ and $N = 5000$. The models are evaluated on the testing dataset of 23175 samples. Our result indicates that increasing the dimension of kernel features decreases the testing MSE. It suggests that overparameterization improves generalization on unseen data. The communication budget limits the number of training samples, thus bottlenecks the generalization accuracy. In the scenario where both computation and communication budgets are limited, Fig. 2 shows that the generalization power of Alg. 2 benefits more from spending into communication budget (thus more training samples) than from overparametrizing, under the situation where $H \ll D$ and more training samples are available with no cost. Table 1 shows the compression ratio for the schemes used in our experiments.

Lastly, Fig. 3 (left) provides additional results for the large-scale overparameterized scenario, where $\frac{D}{N}$ is as large as 19 which is close to the degree of over-parameterization in practice. Comparing the two cases with $D = 24,849$, $D = 97,111$ where the latter case uses 4 times more parameters, we observe that the bit complexity to reach an MSE of $10^{-5}$ only increases by 1.4 times. Fig. 3 (right) examines empirically the effects of running $L_{\mathsf{local}} = 2$ local steps on the train MSE performance against the number of transmitted bits.

# G  The Proof of Theorem 4.4

## G.1  Additional Notations

Let us further define some notations before we go into the details of the proof. Denote the parameter in $t$-th iteration as $\theta^t = (W_l^t)_{l=1}^L$; $P_l = \text{vec}(O_l)$ as the vectorized output from layer $l$; the pre-activation output from layer $l$ as $Q_l$. Similarly, we can define $P_{l,k}, Q_{l,k}$ for each agent. Specifically, $O_L = P_L = Q_L$. Further, let us define some notations related to the singular values of the weight matrices.

$$\bar{\lambda}_l = \begin{cases} \frac{2}{3}\left(1 + s_{\max}(W_l^0)\right), & \text{for } l \in \{1,2\} \\ s_{\max}(W_l^0), & \text{for } l \in \{3,\dots,L\} \end{cases}, \ \underline{\lambda}_l = s_{\min}\left(W_l^0\right), \ \underline{\lambda}_{i \to j} = \prod_{l=i}^{j} \underline{\lambda}_l, \ \bar{\lambda}_{i \to j} = \prod_{l=i}^{j} \bar{\lambda}_l. \tag{35}$$

Specifically, $\bar{\lambda}_l$ represents the quantity related to the largest singular value of weight matrices for different layers; $\underline{\lambda}_l$ represents smallest singular value of weight matrices for different layers.

Further, denote $\lambda_O = s_{\min}\left(\tilde{B} \cdot a\left(XW_1^0\right)\right)$ as the smallest singular value of the output from first hidden layer at initialization multiplied by $\tilde{B}$. Let us define some notations related to gradient. Recall we have defined $u_l = \text{vec}(\nabla_{W_l} f(\theta))$. Let us define the per iteration gradient as: $g_k^t = (Bu_{2,k}^t, u_{3,k}^t; \cdots ; u_{L,k}^t)$ that collects all the (compressed) gradient of each layer. Further, let us denote $\tilde{u}_{2,k}^t = B^\top Bu_{2,k}^t$, $\tilde{g}_k^t = (\tilde{u}_{2,k}^t; u_{3,k}^t; \cdots ; u_{L,k}^t)$, $q_k^t = (z_{2,k}^t; z_{3,k}^t \cdots ; z_{L,k}^t)$, $\tilde{q}_k^t = (B^\top z_{2,k}^t; z_{3,k}^t \cdots ; z_{L,k}^t)$, where $z_{2,k}^t$ quantizes $Bu_{2,k}^t$ and $z_{l,k}^t$ quantizes $u_{l,k}^t$ for $l \geqslant 3$; denote $\tilde{z}_{2,k}^t = B^\top z_{2,k}^t$. Note, that in the above notation, both quantities with $\tilde{\ }$ has the original dimension $D$. Further, $\tilde{q}_k^t$ is the actual vector that gets used at $k$-th agent, while $\tilde{g}_k^t$ represents a "virtual" vector which has not been quantized.

Now let us define:

$$u_l^t := \sum_{k=1}^{K} u_{l,k}^t, \quad u^t = (u_2^t, u_3^t, \cdots, u_L^t),$$

$$g^t := \sum_{k=1}^{K} g_k^t = (Bu_2^t; u_3^t; \cdots ; u_L^t), \ \tilde{g}^t := \sum_{k=1}^{K} \tilde{g}_k^t = (\tilde{u}_2^t; u_3^t; \cdots ; u_L^t),$$

$$q^t := \sum_{k=1}^{K} q_k^t = (z_2^t; z_3^t; \cdots ; z_L^t), \ \tilde{q}^t := \sum_{k=1}^{K} \tilde{q}_k^t = (\tilde{z}_2^t; z_3^t; \cdots, z_L^t).$$

Further, we define $\Delta W_l^t, \tilde{\Delta} W_l^t$, which are the unvectorized $\tilde{u}_2^t$ and $\tilde{q}_2^t$ for $l = 2$ and unvectorized $\tilde{u}_l^t$ and $\tilde{q}_l^t$ for $l \geqslant 3$:

$$\text{vec}(\Delta W_l^t) = \begin{cases} \tilde{u}_2^t & l = 2 \\ u_l^t & l \geqslant 3, \end{cases} \qquad \text{vec}(\tilde{\Delta} W_l^t) = \begin{cases} \tilde{z}_2^t & l = 2 \\ z_l^t & l \geqslant 3. \end{cases} \tag{36}$$

That is, $\Delta W_l^t, \tilde{\Delta} W_l^t \in \mathbb{R}^{n_{l-1} \times n_l}$.

Using the above notation, the 'Update' step in Algorithm 1, it can be expressed as

$$\theta^{t+1} = \theta^t - \eta \tilde{q}^t. \tag{37}$$

Denote $\Sigma_l^t = \text{diag}\left[\text{vec}\left(a'\left(Q_l^t\right)\right)\right] \in \mathbb{R}^{Nn_l \times Nn_l}$, which is a diagonal matrix, whose diagonal entries are the vectorized gradient of the activation function in each layer. Recall that we have defined $B = \mathbf{I}_{n_2} \otimes \tilde{B}$ in Section 4.2, where $\tilde{B}$ is a Gaussian random matrix of size $H \times n_1$. For convenience, let us assume $s_{\max}(B) = s_{\max}(\tilde{B}) \geqslant 1$.

## G.2  Useful Lemma

Now we first state a collection of results from [Nguyen and Mondelli, 2020] that will be used in our proof. Notice that in the aforementioned work, the same pyramidal neural network structure and the $l_2$ loss function are used, so the loss function is the same as (6). It follows that all the properties of the loss function can be reused.

**Lemma G.1.** *Suppose Assumption 4 and 5 hold, for each $\theta^t$, we have the following relations:*

1. $u_l^t = \left(\mathbf{I}_{n_l} \otimes O_{l-1}^{t\top}\right) \prod_{p=l+1}^{L} \Sigma_{p-1}^t \left(W_p^t \otimes \mathbf{I}_N\right) \left(O_L^t - y\right),$ (38)

2. $\dfrac{\partial O_L^t}{\partial \operatorname{vec}(W_l)} = \prod_{p=0}^{L-l-1} \left(W_{L-p}^{t\top} \otimes \mathbf{I}_N\right) \Sigma_{L-p-1}^t \left(\mathbf{I}_{n_l} \otimes F_{l-1}^t\right),$ (39)

3. $\left\|u_2^t\right\| \geqslant s_{\min}\left(O_1^{t\top}\right) \prod_{p=3}^{L} s_{\min}\left(\Sigma_{p-1}^t\right) s_{\min}\left(W_p^t\right) \left\|O_L^t - y\right\|,$ (40)

4. $\left\|u_l^t\right\| \leqslant \|X\|_F \prod_{\substack{p=1 \\ p \neq l}}^{L} s_{\max}(W_p^t) \left\|O_L^t - y\right\|,$ (41)

5. $\|g^t\| \leqslant s_{\max}(\tilde{B}) L \|X\|_F \dfrac{\prod_{l=1}^{L} s_{\max}(W_l^t)}{\min_{l \in [L]} s_{\max}(W_l^t)} \|O_L^t - y\|.$ (42)

*Furthermore, given with $\theta^a$ and $\theta^b$, if $\bar{\Lambda}_l \geq \max\left(s_{\max}\left(W_l^a\right), s_{\max}\left(W_l^b\right)\right)$ for some scalars $\bar{\Lambda}_l$. Let $\tilde{R} := \prod_{p=1}^{L} \max\left(1, \bar{\Lambda}_p\right)$. Then, for $l \in [L]$,*

6. $\left\|O_L^a - O_L^b\right\|_F \leqslant \sqrt{L} \|X\|_F \dfrac{\prod_{l=1}^{L} \bar{\Lambda}_l}{\min_{l \in [L]} \bar{\Lambda}_l} \left\|\theta^a - \theta^b\right\|,$ (43)

7. $\left\|\dfrac{\partial \operatorname{vec}\left(O_L(\theta^a)\right)}{\partial \operatorname{vec}\left(W_l^a\right)} - \dfrac{\partial \operatorname{vec}(O_L\left(\theta^b\right))}{\partial \operatorname{vec}\left(W_l^b\right)}\right\|_2 \leq \sqrt{L}\|X\|_F \tilde{R}\left(1 + L\rho\|X\|_F \tilde{R}\right) \left\|\theta^a - \theta^b\right\|.$ (44)

8. $\left\|\operatorname{vec}\left(\nabla f(\theta^a)\right) - \operatorname{vec}\left(\nabla f(\theta^b)\right)\right\|$

$\leqslant \left(L\sqrt{L}\|X\|_F^2 \dfrac{\prod_{l=1}^{L} \bar{\Lambda}_l^2}{\min_{l \in [L]} \bar{\Lambda}_l^2} + L\sqrt{L}\|X\|_F \tilde{R}(1 + L\rho\|X\|_F \tilde{R})\right) \|\theta^a - \theta^b\|,$ (45)

Now let us discuss the properties above one by one. The relations (38) and (39) show how the vectorized gradients of each layer, as well as the vectorized gradients of the output over each layer are computed, which are true regardless of the algorithm; see the first and the second equalities in [Nguyen and Mondelli, 2020, Lemma 4.1], respectively; (40) is the lower bound of the norm of the vectorized gradient over $W_2$, which is true as long as the network has the pyramidal structure (and regardless of the algorithm); see the third relation in [Nguyen and Mondelli, 2020, Lemma 4.1]; (41) is the upper bound of the norm of vectorized gradient over $W_l$, which is true regardless of the algorithm; see the third relation in [Nguyen and Mondelli, 2020, Lemma 4.2]; (42) can be derived by summing over $l$ in (41). To be specific, we have

$$\|g^t\| \leqslant \|Bu_2^t\| + \sum_{l=3}^{L} \|u_l^t\| \leqslant s_{\max}(\tilde{B})\|u_2^t\| + \sum_{l=3}^{L} \|u_l^t\|$$

$$\leqslant s_{\max}(\tilde{B}) \sum_{l=2}^{L} \|u_l^t\|$$

$$\leqslant s_{\max}(\tilde{B}) \sum_{l=2}^{L} \|X\|_F \prod_{\substack{p=1 \\ p \neq l}}^{L} s_{\max}(W_p) \|O_L - y\|$$

$$\leqslant s_{\max}(\tilde{B})L\|X\|_F \frac{\prod_{l=1}^{L} s_{\max}(W_l^t)}{\min_{l\in[L]} s_{\max}(W_l^t)} \|O_L^t - y\|.$$

The relation in (43) gives the upper bound of the gap between output layer with two sets of parameters $\theta^a$ and $\theta^b$, and it is regardless of the algorithm; see [Nguyen and Mondelli, 2020, Eq. (19)]; Similarly, (44) states the gap between the vectorized Jacobian matrix over $W_l$; see [Nguyen and Mondelli, 2020, Eq. (20)]; Finally, (45) computes the Lipschitz constants for the gradient, it is independent of the algorithm, and comes from plugging in (42), (43) and (44) into [Nguyen and Mondelli, 2020, Eq. (22)].

### G.3   Initialization Strategy

As stated in Theorem 4.4, special initialization strategy is required. Now let us describe the initialization in detail. Recall the definition of $\bar{\lambda}_l$ and $\underline{\lambda}_l$ in (35) and $\lambda_O = s_{\min}(\tilde{B} \cdot a(XW_1^0))$. Initialize $\theta^0$ such that the following holds:

$$\lambda_O^2 \geqslant \frac{48 \cdot 6^{L-2} \|X\|_F \bar{\lambda}_{1\to L} s_{\max}^2(\tilde{B}) \|O_L^0 - y\|}{(\nu^2)^{L-2} \underline{\lambda}_{3\to L}^2} \max\left( \frac{1}{\bar{\lambda}_2}, \max_{l\geqslant 3} \frac{2}{\bar{\lambda}_l \underline{\lambda}_l} \right). \tag{46}$$

To satisfy the above relation, it requires that $\|O_L^0 - y\|$ cannot be large, which means the $\theta^0$ should not be far from the optimal solution so that the initial loss is small. Further, it requires the gap between $\bar{\lambda}_l$ and $\underline{\lambda}_l$ is small. From (45), we know that small $\bar{\lambda}_l$ induces small Lipschitz constant for gradient, so the condition requires that the Lipschitz constant is not very large, which further implies that the optimization landscape is smooth. On the other hand, from (40), we know the lower bound of the norm of $u_2^t$ is related to $\underline{\lambda}_l$. So the initialization condition guarantees the lower bound of $\|u_2^t\|$ is not too small, which avoids vanished gradient.

The initialization can be realized by using the procedure suggested [Nguyen and Mondelli, 2020, Sec. 3.1]. The idea is that we can scale up $W_1^0$ to make $\lambda_O$ not too small, and then randomly choose $W_2^0$ with small entries. Then for all $l \geqslant 3$, set $W_l^0$ as scaled identity matrices (top block as scaled identities) with large entries.

### G.4   Formal Statement and Proof of Theorem 4.4

First, let us state the formal Theorem 4.4:

**Theorem 4.4** *Consider using Alg. 1 to solve the problem (6), with $X$ being full row rank. Suppose $\theta^0$ is initialized as (46). Choose $\tilde{B}$ such as $\operatorname{rank}(\tilde{B}O_1^{0\top}) = N$; choose $\tilde{F}(\cdot)$ and $F(\cdot)$ as in (7). Set stepsize $\eta$ and bits number $b$ as following:*

$$\eta = \frac{\phi}{Q_0 \Lambda^2}, \ b = \max\left( \log(\frac{1}{\tau} + 1), b_0 \right),$$

*where $\phi$ is defined in (47), $\tau$ is defined in (50), $\Lambda$ is defined in (48), $Q_0$ is defined in (49). The following hold true:*

$$f(\theta^{t+1}) \leqslant (1 - \frac{1}{4}\eta \cdot \phi)f(\theta^t).$$

*To compute an $\epsilon$-optimal solution (6), each agent is required to transmit:*

$$\frac{4b}{\eta\phi} \cdot \left( Hn_2 + \sum_{l=2}^{L-1} n_l n_{l+1} \right) \log(f(\theta^0)/\epsilon) \quad \text{bits/agent.}$$

*Proof.* To begin with, let us set the following constants:

$$\alpha = 1 - \frac{1}{4}\eta\phi, \quad \gamma^t = \lambda\sqrt{(\alpha)^{t+1}f(\theta^0)}, \quad \eta = \frac{\phi}{Q_0 \Lambda^2}, \quad \lambda = 6\sqrt{2}s_{\max}(\tilde{B})\eta Q_0 \Lambda, \tag{47}$$

$$C_1 := \frac{\phi}{16\sqrt{2}Hn_1n_2^2\|\tilde{B}\|_\infty\lambda\|O_L^0 - y\|}, \quad C_2 := \frac{\phi\min_{l\geqslant 3}\underline{\lambda}_l}{32\sqrt{2}\lambda n_2 n_3\|O_L^0 - y\|},$$

$$C_3 := \frac{1}{\sqrt{2}\eta Hn_1n_2^2\|\tilde{B}\|_\infty\lambda\|O_L^0 - y\|}, \quad C_4 := \frac{\min_{l\geqslant 3}\underline{\lambda}_l}{2\sqrt{2}\eta\lambda n_2 n_3\|O_L^0 - y\|}$$

$$C_5 := \sqrt{\frac{\phi}{4\eta\left(\frac{1}{2} + \frac{1}{\beta}\right)\left(Hn_1n_2^2\|\tilde{B}\|_\infty^2 + \sum\limits_{l=3}^{L} n_{l-1}^2 n_l^2\right)\Lambda^2\lambda^2}},$$

$$C_6 := \frac{\sqrt{1 - \frac{1}{4}\eta\phi} - \frac{1}{6}}{1 + \eta s_{\max}(\tilde{B})Q_0\left(Hn_2 + \sum\limits_{l=3}^{L} n_{l-1}n_l\right)},$$

where

$$\phi = \left(\frac{1}{2}\nu\right)^{L-2}\lambda_O^2\underline{\lambda}_{3\to L}^2, \quad \Lambda = \left(\frac{3}{2}\right)^{L-1}\cdot Ls_{\max}^2(\tilde{B})\|X\|_F\frac{\bar{\lambda}_{1\to L}}{\min\limits_{l\geqslant 2}\bar{\lambda}_l} \tag{48}$$

$$Q_0 = L\sqrt{L}\left(\frac{3}{2}\right)^{2(L-1)}\|X\|_F^2\frac{\prod_{l=1}^{L}\bar{\lambda}_l^2}{\min\limits_{l\in[L]}\bar{\lambda}_l^2} + L\sqrt{L}\|X\|_F R\left(1 + L\rho\|X\|_F R\right), \tag{49}$$

$$R = s_{\max}^2(\tilde{B})\prod\limits_{p=1}^{L}\max\left(1, \frac{3}{2}\bar{\lambda}_p\right).$$

Define

$$\tau := \min(C_1, C_2, C_3, C_4, C_5, C_6). \tag{50}$$

Further, let us define

$$\Lambda_l := \left(\frac{3}{2}\right)^{L-1}\|X\|_F\frac{\bar{\lambda}_{1\to L}}{\bar{\lambda}_l}. \tag{51}$$

For convenience, let us assume $Q_0 > 1$. Let us explain the above constants: $\alpha$ is the constants based on which the objective function contracts; $\eta$ is the stepsize; $\phi$ is related to the lower bound of $\|u_2^t\|$; $\Lambda$ is related to the upper bound of $\|\tilde{g}^t\|$; $Q_0$ is the Lipschitz constant for the full gradient; $\tau$ is the parameter related to quantization; $\Lambda_l$ is related to the upper bound for $\|u_l^t\|$.

The majority of the proof consists of showing the following relations by induction:

$$\begin{cases} s_{\max}\left(W_l^t\right) \overset{(i)}{\leqslant} \frac{3}{2}\bar{\lambda}_l, l = \{2, 3, \cdots, L\}, \\ s_{\min}\left(W_l^t\right) \overset{(ii)}{\geqslant} \frac{1}{2}\underline{\lambda}_l, \ l \in \{3, \ldots, L\}, \\ s_{\max}(W_2^t + \eta\tilde{\Delta}W_2^{t-1} - \eta\Delta W_2^{t-1}) \overset{(iii)}{\leqslant} \frac{3}{2}\bar{\lambda}_l, l = \{2, 3, \cdots, L\}, \\ s_{\min}(W_2^t + \eta\tilde{\Delta}W_2^{t-1} - \eta\Delta W_2^{t-1}) \overset{(iv)}{\geqslant} \frac{1}{2}\underline{\lambda}_l, l = \{3, \cdots, L\}, \\ f(\theta^{t-1} - \eta\tilde{g}^{t-1}) \overset{(v)}{\leqslant} (1 - \eta\phi)f(\theta^{t-1}), \\ f(\theta^t) \overset{(vi)}{\leqslant} (\alpha)^t f(\theta^0), \\ \|Bu_2^t - z_2^t\|_\infty \overset{(vii)}{\leqslant} \tau\gamma^{t-1} = \tau\lambda\sqrt{(\alpha)^t f(\theta^0)}, \\ \|u_l^t - z_l^t\|_\infty \overset{(viii)}{\leqslant} \tau\gamma^{t-1} = \tau\lambda\sqrt{(\alpha)^t f(\theta^0)}. \end{cases} \tag{52}$$

Let us explain the meanings of the above relations. Relations (i) and (ii) provide the upper and lower bounds of the singular values of weight matrices in each iteration; (iii) and (iv) give the upper and lower bound of the singular values of weight matrices after one step update without quantization; (v) shows the decrease of loss function after one step of update without quantization; (vi) shows the

linear decrease of loss function in each iteration; (vii) and (viii) provide the error bound of gradient after quantization.

Compared to the induction proof in [Nguyen and Mondelli, 2020], the key challenges are: (1) Our analysis includes the quantization of gradient; (2) The update of parameter is not the simple gradient but a function of gradient.

Our proof consists of two steps:

**Step 1:** We show the above relations hold for $t = 1$.

**Step 2:** We show that if the above relations hold for $t$, then they hold for $t + 1$.

We show Step 1 first. For simplicity, we set $g^0 = q^0$, it is easy to verify the relations hold if we choose $q^0$ in Algorithm 1 with large enough C. Step 1 will be shown in the following five substeps: (a) Show (i) and (ii) in (52); (b) Show (iii) and (iv) in (52); (c) Show (v) in (52); (d) Show (vi) in (52); (e) Show (vii) and (viii) in (52).

**(Step 1.a)** We will show

$$\begin{cases} s_{\max}\left(W_l^1\right) \leqslant \frac{3}{2}\bar{\lambda}_l \;\; l \in [L],\; l \in \{2,3,\ldots,L\} \\ s_{\min}\left(W_l^1\right) \geqslant \frac{1}{2}\underline{\lambda}_l,\; l \in \{3,\ldots,L\}. \end{cases}$$

We will use the fact that, according to the update rule in (37), we have $q^0 = g^0$.

For $l = 2$, we have:

$$\|W_2^1 - W_2^0\|_F \overset{(i)}{=} \eta\|\tilde{z}_2^0\| = \eta\|B^\top B u_2^0\| \overset{(ii)}{\leqslant} \eta s_{\max}^2(\tilde{B})\|u_2^0\|$$
$$\overset{(iii)}{\leqslant} \eta s_{\max}^2(\tilde{B})\Lambda_2\|O_L^0 - y\|$$
$$\overset{(iv)}{\leqslant} \sum_{t=0}^{\infty} \eta s_{\max}^2(\tilde{B})\Lambda_2\|O_L^0 - y\| \cdot \sqrt{\alpha}^t$$
$$\leqslant \eta s_{\max}^2(\tilde{B})\Lambda_2\|O_L^0 - y\|\frac{1}{1-\alpha}(1 + \alpha^{\frac{1}{2}})$$
$$\overset{(v)}{\leqslant} \frac{8 s_{\max}^2(\tilde{B})\Lambda_2}{\phi}\|O_L^0 - y\|$$
$$\overset{(vi)}{\leqslant} \frac{1}{2},$$

where (i) is from the update rule in (37); (ii) extracts the largest singular value of $B$; (iii) uses the upper bound of the gradient norm in (41) and the definition of $\Lambda_2$ in (51) ; (iv) uses the fact that $\sum_{t=0}^{\infty} \sqrt{\alpha} > 1$; (v) plugs in the definition of $\alpha$ and $1 + \alpha^{\frac{1}{2}} < 2$; (vi) is because the initialization strategy in (46).

Next, we show the case where $l \geqslant 3$. We have

$$\|W_l^1 - W_l^0\|_F \overset{(i)}{=} \eta\|u_l^0\| \overset{(ii)}{\leqslant} \eta\Lambda_l\|u_l^0\|$$
$$\overset{(iii)}{\leqslant} \sum_{t=0}^{\infty} \eta\Lambda_l\|O_L^0 - y\| \cdot \sqrt{\alpha}^t$$
$$= \eta\Lambda_l\|O_L^0 - y\|\frac{1}{1-\alpha^{\frac{1}{2}}}$$
$$\leqslant \eta\Lambda_l\|O_L^0 - y\|\frac{1}{1-\alpha}(1 + \alpha^{\frac{1}{2}})$$
$$\overset{(iv)}{\leqslant} \frac{8\Lambda_l}{\phi}\|O_L^0 - y\|$$
$$\overset{(v)}{\leqslant} \frac{1}{4}\underline{\lambda}_l < \frac{1}{2}\underline{\lambda}_l,$$

where (i) is because the update of parameter in (37); (ii) uses the upper bound of the gradient norm in (41) and definition of $\Lambda_l$ in (51); (iii) uses the fact that $\sum_{t=0}^{\infty} \sqrt{\alpha} > 1$; (iv) plugs in the choice of $\alpha$ and $1 + \alpha^{\frac{1}{2}} < 2$; (v) comes from the initialization strategy in (46).

Applying Weyl' inequality to the matrices $W_l^0$ and $(W_l^1 - W_l^0)$, we have

$$
\begin{cases}
s_{\max}(W_l^1) \leqslant \bar{\lambda}_l + \frac{1}{2}\bar{\lambda}_l = \frac{3}{2}\bar{\lambda}_l, \ l \in \{3, \cdots, L\}, \\
s_{\max}(W_l^1) \leqslant \bar{\lambda}_l + 1 = \frac{3}{2}\bar{\lambda}_l, \ l = 2, \\
s_{\min}(W_l^1) \geqslant \underline{\lambda}_l - \frac{1}{2}\bar{\lambda}_l = \frac{1}{2}\underline{\lambda}_l, \ l \in \{3, \ldots, L\}.
\end{cases}
$$

This concludes the proof of this substep.

**(Step 1.b)** We will show

$$
\begin{cases}
s_{\max}(W_l^1 + \eta\tilde{\Delta}W_l^0 - \eta\Delta W_l^0) \leqslant \frac{3}{2}\bar{\lambda}_l, l = \{2, 3, \cdots, L\}, \\
s_{\min}(W_l^1 + \eta\tilde{\Delta}W_l^0 - \eta\Delta W_l^0) \geqslant \frac{1}{2}\underline{\lambda}_l, l = \{3, \cdots, L\}.
\end{cases}
$$

Since we have $\tilde{q}^0 = \tilde{g}^0$, it is easy to derive that

$$
\|\eta\tilde{\Delta}W_l^0 - \eta\Delta W_l^0\|_F = \eta\|\tilde{\Delta}W_l^0 - \eta\Delta W_l^0\| = \eta\|\tilde{q}^0 - \tilde{g}^0\| = 0.
$$

Thus we can conclude

$$
\begin{cases}
s_{\max}(W_l^1 + \eta\tilde{\Delta}W_l^0 - \eta\Delta W_l^0) = s_{\max}(W_l^1) \leqslant \frac{3}{2}\bar{\lambda}_l, l = \{2, 3, \cdots, L\} \\
s_{\min}(W_l^1 + \eta\tilde{\Delta}W_l^0 - \eta\Delta W_l^0) = s_{\max}(W_l^1) \geqslant \frac{1}{2}\underline{\lambda}_l, l = \{3, \cdots, L\}
\end{cases}
$$

This concludes the proof for this substep.

**(Step 1.c)** We will show

$$
f(\theta^0 - \eta\tilde{g}^0) \leqslant (1 - \eta\alpha_0)f(\theta^0).
$$

From (Step 1.a) and (Step 1.b) we know

$$
\max\left(s_{\max}(W_l^0), s_{\max}(W_l^1)\right) \leqslant \frac{3}{2}\bar{\lambda}_l.
$$

Using the above relation, we can upper bound the differences of the gradients by using (45). More specifically, using the the definition of $Q_0$ in (49), we have

$$
\|\text{vec}(\nabla f(\theta^1)) - \text{vec}(\nabla f(\theta^0))\| \leqslant Q_0\|\theta^1 - \theta^0\|, \tag{53}
$$

where we repeat the definition of $Q_0$ below (where $R$ is defined in (49)):

$$
Q_0 = L\sqrt{L}\left(\frac{3}{2}\right)^{2(L-1)} \|X\|_F^2 \frac{\prod_{l=1}^{L} \bar{\lambda}_l^2}{\min_{l \in [L]} \bar{\lambda}_l^2} + L\sqrt{L}\|X\|_F R\left(1 + L\rho\|X\|_F R\right),
$$

Further, it is easy to verify that for any $\hat{\theta}^0$ between $\theta^0$ and $\theta^1$, we still have $s_{\max}(W_l(\hat{\theta}^0)) \leqslant \frac{3}{2}\bar{\lambda}_l$. So we can apply the same argument leading to (53), and obtain:

$$
\|\text{vec}(\nabla f(\hat{\theta}^0)) - \text{vec}(\nabla f(\theta^0))\| \leqslant Q_0\|\hat{\theta}^0 - \theta^0\|, \forall\hat{\theta}^0 = \theta^0 + \delta(\theta^1 - \theta^0), \delta \in [0, 1]. \tag{54}
$$

As long as for any $\hat{\theta}^0$ (54) holds, we can apply Decent Lemma. More specifically, we have

$$
f(\theta^0 - \eta\tilde{g}^0) \overset{(i)}{\leqslant} f(\theta^0) - \eta\langle u^0, \tilde{g}^0\rangle + \frac{Q_0}{2}\eta^2\|\tilde{g}^0\|^2
$$

$$
\overset{(ii)}{=} f(\theta^0) - \eta\langle u_2^0, \tilde{u}_2^0\rangle - \eta\sum_{l=3}^{L}\|u_l^0\|^2 + \frac{Q_0}{2}\eta^2\|\tilde{g}^0\|^2
$$

$$
\leqslant f(\theta^0) - \eta\langle u_2^0, \tilde{u}_2^0\rangle + \frac{Q_0}{2}\eta^2\|\tilde{g}^0\|^2, \tag{55}
$$

where (i) uses Decent Lemma, (ii) uses the fact that $u^0$ is the stacked version of $\{u_i^0\}_{i\geq 2}$, and $\tilde{g}^0$ are stacked versions of $\{u_i^0\}_{i\geq 3}$ and $\tilde{u}_2^0$.

Next, we will bound the inner product on the right hand side of the above relation:

$$
\langle u_2^0, \tilde{u}_2^0 \rangle \overset{(i)}{=} \left\langle \left( \mathbf{I}_{n_2} \otimes O_1^{0\top} \right) \prod_{l=3}^{L} \Sigma_{l-1}^0 \left( W_l^0 \otimes \mathbf{I}_N \right) \left( O_L^{0\top} - y \right), B^\top B \left( \mathbf{I}_{n_2} \otimes O_1^{0\top} \right) \prod_{l=3}^{L} \Sigma_{l-1}^0 \left( W_l^0 \otimes \mathbf{I}_N \right) \left( O_L^{0\top} - y \right) \right\rangle
$$

$$
\overset{(ii)}{=} \left\langle \left( \mathbf{I}_{n_2} \otimes O_1^{0\top} \right) \prod_{l=3}^{L} \Sigma_{l-1}^0 \left( W_l^0 \otimes \mathbf{I}_N \right) \left( O_L^{0\top} - y \right), \left( \mathbf{I}_{n_2} \otimes \tilde{B}^\top \tilde{B} O_1^{0\top} \right) \prod_{l=3}^{L} \Sigma_{l-1}^0 \left( W_l^0 \otimes \mathbf{I}_N \right) \left( O_L^{0\top} - y \right) \right\rangle
$$

$$
\overset{(iii)}{=} \left\| \left( \mathbf{I}_{n_2} \otimes \tilde{B} O_1^{0\top} \right) \prod_{l=3}^{L} \Sigma_{l-1}^0 \left( W_l^0 \otimes \mathbf{I}_N \right) \left( O_L^0 - y \right) \right\|^2
$$

$$
\geqslant s_{\min}^2(\tilde{B} O_1^{0\top}) \prod_{l=3}^{L} s_{\min}^2 \left( \Sigma_{l-1}^0 \right) s_{\min}^2 \left( W_l^0 \right) \left\| O_L^0 - y \right\|^2
$$

$$
\overset{(iv)}{\geqslant} (\tfrac{1}{2}\nu)^{2(L-2)} \lambda_O^2 \underline{\lambda}_{3 \to L}^2 \| O_L^0 - y \|^2, \tag{56}
$$

where (i) uses the expression of $u_2^0$ by (38) and $\tilde{u}_2^0 = B^\top B u_2^0$; (ii) and (iii) are from the property of Kronecker product; (iv) is from Assumption 5 and fact that $s_{\max}(W_l^0) \geqslant \frac{1}{2}\lambda_l$.

Next, we upper bound the term $\|\tilde{g}^0\|^2$ in (55). We have

$$
\|\tilde{g}^0\|^2 \overset{(i)}{=} \|B^\top B u_2^0\|^2 + \sum_{l=3}^{L} \|u_l^0\|^2 \overset{(ii)}{\leqslant} s_{\max}^4(\tilde{B}) \|u_2^0\|^2 + \sum_{l=3}^{L} \|u_l^0\|^2
$$

$$
\overset{(iii)}{\leqslant} s_{\max}^2(\tilde{B}) \left( \|B u_2^0\|^2 + \sum_{l=3}^{L} \|u_l^0\|^2 \right) = s_{\max}^2(\tilde{B}) \|g^0\|^2
$$

$$
\overset{(iv)}{\leqslant} s_{\max}^2(\tilde{B}) \left( L s_{\max}\left( \tilde{B} \right) \|X\|_F \frac{\prod_{l=1}^{L} s_{\max}\left( W_l^0 \right)}{\min_l s_{\max}\left( W_l^0 \right)} \right)^2 \left\| O_L^0 - y \right\|_2^2
$$

$$
\overset{(v)}{\leqslant} (\tfrac{3}{2})^{2(L-1)} L^2 s_{\max}^4\left( \tilde{B} \right) \|X\|_F^2 \frac{\bar{\lambda}_{1 \to L}^2}{\min_l \bar{\lambda}_l^2} \| O_L^0 - y \|^2,
$$

$$
\overset{(vi)}{=} \Lambda^2 \| O_L^0 - y \|^2 \tag{57}
$$

where (i) uses the definition of $\tilde{g}^0$; (ii) extracts the largest singular value of $B$; (iii) is because we assume $s_{\max}(\tilde{B}) \geqslant 1$; (iv) uses the upper bound of gradient in (42), (v) comes from the fact that $s_{\max}(W_l^0) \geqslant \frac{1}{2}\lambda_l$; (iii) is because the definition of $\Lambda$ in (48).

Recall that we have defined

$$
\phi := (\tfrac{1}{2}\nu)^{2(L-2)} \lambda_O^2 \underline{\lambda}_{3 \to L}^2.
$$

If we choose $\eta = \frac{\phi}{Q_0 \Lambda^2}$, then plug (56) and (57) into (55), we obtain

$$
f(\theta^0 - \eta \tilde{g}^0) = f(\theta^0) - 2\eta \phi f(\theta^0) + \eta \phi f(\theta^0) = (1 - \eta\phi) f(\theta^0). \tag{58}
$$

**(Step 1.d)** We will show

$$
f(\theta^1) \leqslant \alpha f(\theta^0).
$$

Since $\tilde{g}^0 = \tilde{q}^0$, we have $f(\theta^1) = f(\theta^0 - \eta \tilde{g}^0) \overset{(i)}{\leqslant} (1 - \eta\phi) f(\theta^0) < \alpha f(\theta^0)$, where (i) is because (58). This completes the proof of this step.

**(Step 1.e)** We will show that:

$$
\begin{cases} \|B u_l^1 - v_l^1\|_\infty \leqslant \tau \gamma^0, \ l = 2 \\ \|u_l^1 - v_l^1\|_\infty \leqslant \tau \gamma^0, \ l = \{3, \cdots, L\}. \end{cases}
$$

Notice that for $l = 2$, we have

$$\|Bu_2^1 - z_2^0\|_\infty \overset{(i)}{\leqslant} \|Bu_2^1 - Bu_2^0\| + \|Bu_2^0 - z_2^0\|_\infty \overset{(ii)}{\leqslant} s_{\max}(\tilde{B})\|u_2^1 - u_2^0\|$$

$$\overset{(iii)}{\leqslant} s_{\max}(\tilde{B})\|\text{vec}(\nabla f(\theta^1)) - \text{vec}(\nabla f(\theta^0))\|$$

$$\overset{(iv)}{\leqslant} s_{\max}(\tilde{B})Q_0\|\theta^1 - \theta^0\| \overset{(v)}{=} \eta s_{\max}(\tilde{B})Q_0\|\tilde{q}^0\|$$

$$\overset{(vi)}{=} \eta s_{\max}(\tilde{B})Q_0\|\tilde{g}^0\| \overset{(vii)}{\leqslant} \eta s_{\max}(\tilde{B})Q_0\Lambda\|O_L^0 - y\|$$

$$\overset{(viii)}{\leqslant} \lambda\sqrt{\alpha f(\theta^0)} = \gamma^0,$$

where (i) uses triangle inequality; (ii) extracts the largest singular value of $B$ and uses $g^0 = q^0$; (iii) is because $\|u_2^1 - u_2^0\| \leqslant \|u^1 - u^0\|$; (iv) is from (53), which uses (45) and definition of $Q_0$ in (49); (v) uses the update rule in (37); (vi) is because $g^0 = q^0$; (vii) uses the definition of $\Lambda$ in (48); (viii) uses the definition of $\lambda$ in (47) and the fact $\alpha = 1 - \frac{\phi^2}{4Q_0\Lambda^2} > 1 - \frac{1}{4} = \frac{3}{4}$.

Then by Lemma 4.1, with the correspondence that $c = Bu_2^1, p = z_2^0, r = \gamma^0$, we can obtain

$$\|z_2^1 - Bu_2^1\| \overset{(i)}{=} \|\text{quant}(Bu_2^1, z_2^0, \gamma^0, b) - Bu_2^1\|_\infty \overset{(ii)}{\leqslant} \tau\gamma^0,$$

where $(i)$ comes from the 'Quantize' step in Algorithm 1; $(ii)$ is from the first relation in Lemma 4.1. For $l \geqslant 3$, we have

$$\|u_l^1 - v_l^0\|_\infty \overset{(i)}{\leqslant} \|u_l^1 - u_l^0\| + \|u_l^0 - z_l^0\|_\infty \overset{(ii)}{\leqslant} \|\text{vec}(\nabla f(\theta^1)) - \text{vec}(\nabla f(\theta^0))\|$$

$$\overset{(iii)}{\leqslant} Q_0\|\theta^1 - \theta^0\| \overset{(iv)}{=} \eta Q_0\|\tilde{q}^0\| \overset{(v)}{=} \eta Q_0\|\tilde{g}^0\| \overset{(vi)}{\leqslant} \eta Q_0\Lambda\|O_L^0 - y\|$$

$$\overset{(vii)}{\leqslant} \lambda\sqrt{\alpha f(\theta^0)} = \gamma_0,$$

where (i) uses triangle inequality; (ii) is because $g^0 = q^0$ and uses the fact $\|u_l^0 - z_l^0\| \leqslant \|g^0 - q^0\|$; (iii) is from (53); (iv) uses the update rule in (37); (v) is because $\tilde{g}^0 = \tilde{q}^0$; (vi) uses the upper bound of $\|\tilde{g}^0\|$ in (42) and the definition of $\Lambda$ in (48); (vii) comes from the choice of $\lambda$ in (47) and the fact $\alpha = 1 - \frac{\phi^2}{4Q_0\Lambda^2} > 1 - \frac{1}{4} = \frac{3}{4}$.

Then by Lemma 4.1, with the correspondence that $c = u_l^1, p = z_l^0, r = \gamma^0$, we can obtain

$$\|z_l^{t+1} - u_l^{t+1}\| \overset{(i)}{=} \|\text{quant}(u_l^1, z_l^0, \gamma^0, b) - u_l^1\|_\infty \overset{(ii)}{\leqslant} \tau\gamma^0.$$

where $(i)$ comes from the 'Quantize' step in Algorithm 1; $(ii)$ is from the first relation in Lemma 4.1. Now we have showed all the induction assumptions hold for $t = 1$.

Next, we show Step 2. Given these inequalities in (52) hold for iteration $t$, we aim to show that they hold for $t + 1$. We prove Step 2 in five substeps similarly as in the proof of Step 1.

**(Step 2.a)** We will show that

$$\begin{cases} s_{\max}\left(W_l^{t+1}\right) \leq \frac{3}{2}\bar{\lambda}_l \ \ l \in [L], \ l \in \{2, 3, \dots, L\} \\ s_{\min}\left(W_l^{t+1}\right) \geqslant \frac{1}{2}\underline{\lambda}_l, \ l \in \{3, \dots, L\}. \end{cases}$$

For $l = 2$, we have:

$$\|W_2^{t+1} - W_2^0\|_F \overset{(i)}{\leqslant} \sum_{r=0}^t \|W_2^{r+1} - W_2^r\|_F \overset{(ii)}{\leqslant} \sum_{r=0}^t \|\tilde{z}_2^r\|$$

$$\overset{(iii)}{\leqslant} \eta \sum_{r=0}^t (\|\tilde{u}_2^r\| + \|\tilde{u}_2^r - \tilde{z}_2^r\|)$$

$$\overset{(iv)}{\leqslant} \eta \sum_{r=0}^t (s_{\max}^2(\tilde{B})\Lambda_2\|O_L^r - y\| + n_1 n_2\|B^\top(Bu_2^r - z_2^r)\|_\infty)$$

$$\overset{(v)}{\leqslant} \eta \sum_{r=0}^{t} (s_{\max}^2(\tilde{B})\Lambda_2 \|O_L^r - y\| + H n_1 n_2^2 \|\tilde{B}^\top\|_\infty \|B u_2^r - z_2^r\|_\infty)$$

$$\overset{(vi)}{\leqslant} \eta \sum_{r=0}^{t} \left(s_{\max}^2(\tilde{B})\Lambda_2 \|O_L^r - y\| + H n_1 n_2^2 \|\tilde{B}^\top\|_\infty \tau \lambda \sqrt{(\alpha)^r f(\theta^0)}\right)$$

$$\overset{(vii)}{=} (\eta s_{\max}^2(\tilde{B})\Lambda_2 + \frac{\sqrt{2}}{2} \eta H n_1 n_2^2 \|\tilde{B}^\top\|_\infty \tau \lambda) \|O_L^0 - y\| \sum_{r=0}^{t} \alpha^{\frac{r}{2}}$$

$$= (\eta s_{\max}^2(\tilde{B})\Lambda_2 + \frac{\sqrt{2}}{2} \eta H n_1 n_2^2 \|\tilde{B}^\top\|_\infty \tau \lambda) \|O_L^0 - y\| \frac{1 - \alpha^{\frac{r+1}{2}}}{1 - \alpha^{\frac{1}{2}}}$$

$$\overset{(viii)}{\leqslant} (\eta s_{\max}^2(\tilde{B})\Lambda_2 + \frac{\sqrt{2}}{2} \eta H n_1 n_2^2 \|\tilde{B}^\top\|_\infty \tau \lambda) \|O_L^0 - y\| \frac{1}{1 - \alpha}(1 + \alpha^{\frac{1}{2}})$$

$$\overset{(ix)}{\leqslant} \frac{8(s_{\max}^2(\tilde{B})\Lambda_2 + \frac{\sqrt{2}}{2} H n_1 n_2^2 \|\tilde{B}^\top\|_\infty \tau \lambda)}{\phi} \|O_L^0 - y\|$$

$$\overset{(x)}{\leqslant} \frac{1}{4} + \frac{1}{4} = \frac{1}{2} < 1, \tag{59}$$

where (i) uses triangle inequality; (ii) uses the update rule in (37); (iii) uses triangle inequality; (iv) extracts the largest singular value of $B^\top B$, the upper bound of $\|u_2^t\|$ in (41), definition of $\Lambda_2$ in (51) and the relationship between $l_2$ and $l_\infty$ norm; (v) uses the fact that $\|B^\top(B u_2^r - z_2^r)\|_\infty \leqslant H n_2 \|\tilde{B}\|_\infty \|B u_2^r - z_2^r\|_\infty$; (vi) uses the induction assumption $\|B u_2^r - z_2^r\|_\infty \leqslant \tau \gamma^{r-1}$ and definition of $\gamma^{r-1}$; (vii) reorganizes the terms; (viii) is because $0 < \alpha < 1$; (ix) uses $1 + \alpha^{\frac{1}{2}} < 2$ and plugs in the choice of $\alpha$; (x) is because the initialization strategy in (46) and choice of $\tau$ in (50).

So Applying Weyl' inequality to the matrices $W_2^t$ and $(W_2^{t+1} - W_2^t)$ and (59), we have

$$s_{\max}(W_2^{t+1}) \leqslant \bar{\lambda}_2 + 1 = \frac{3}{2} \bar{\lambda}_2.$$

For $l \geqslant 3$, we have the following relation:

$$\|W_l^{t+1} - W_l^0\|_F \overset{(i)}{\leqslant} \sum_{r=0}^{t} \|W_l^{r+1} - W_l^r\|_F \overset{(ii)}{\leqslant} \sum_{r=0}^{t} \|z_l^r\|$$

$$\overset{(iii)}{\leqslant} \eta \sum_{r=0}^{t} (\|u_l^r\| + \|u_l^r - z_l^r\|)$$

$$\overset{(iv)}{\leqslant} \eta \sum_{r=0}^{t} (\Lambda_l \|O_L^r - y\| + n_{l-1} n_l \|u_l^r - z_l^r\|_\infty)$$

$$\overset{(v)}{\leqslant} \eta \sum_{r=0}^{t} \left(\Lambda_l \|O_L^r - y\| + n_{l-1} n_l \tau \lambda \sqrt{(\alpha)^r f(\theta^0)}\right)$$

$$\overset{(vi)}{\leqslant} (\eta \Lambda_l + \frac{\sqrt{2}}{2} \eta \tau \lambda n_{l-1} n_l) \|O_L^0 - y\| \sum_{r=0}^{t} \alpha^{\frac{r}{2}}$$

$$\overset{(vii)}{\leqslant} (\eta \Lambda_l + \frac{\sqrt{2}}{2} \eta \tau \lambda n_{l-1} n_l) \|O_L^0 - y\| \frac{1}{1 - \alpha}$$

$$= (\eta \Lambda_l + \frac{\sqrt{2}}{2} \eta \tau \lambda n_{l-1} n_l) \|O_L^0 - y\| \frac{1}{1 - \alpha}(1 + \alpha^{\frac{1}{2}})$$

$$\overset{(viii)}{\leqslant} \frac{8(\Lambda_l + \frac{\sqrt{2}}{2} \tau \lambda n_{l-1} n_l)}{\phi} \|O_L^0 - y\|$$

$$\overset{(ix)}{\leqslant} \frac{1}{8} \lambda_l + \frac{1}{8} \lambda_l = \frac{1}{4} \lambda_l < \frac{1}{2} \bar{\lambda}_l, \tag{60}$$

where (i) uses triangle inequality; (ii) uses the update rule in (37); (iii) uses triangle inequality; (iv) uses the upper bound of $\|u_l^t\|$ in (41), the definition of $\Lambda_l$ in (51) and the relationship between $l_2$ and

$l_\infty$ norm; (v) uses the induction assumption $\|u_l^r - z_l^r\| \leqslant \tau\gamma^{r-1}$; (vi) uses reorganizes the terms; (vii) is because $0 < \alpha < 1$; (viii) uses the fact $1 + \alpha^{\frac{1}{2}} < 2$ and plugs in the choice of $\alpha$; (ix) is from the initialization strategy in (46) and the choice of $\tau$ in (50).

Applying Weyl' inequality to the matrices $W_l^{t+1}$ and $(W_l^{t+1} - W_l^t)$, we have

$$s_{\max}(W_l^{t+1}) \leqslant \bar\lambda_l + \frac{1}{4}\bar\lambda_l = \frac{5}{4}\bar\lambda_l < \frac{3}{2}\bar\lambda_l, \; s_{\min}(W_l^{t+1}) \geqslant \bar\lambda_l - \frac{1}{2}\underline\lambda_l = \frac{1}{2}\bar\lambda_l. \tag{61}$$

**(Step 2.b)** We will show that

$$\begin{cases} s_{\max}(W_l^{t+1} + \eta\tilde\Delta W_l^t - \eta\Delta W_l^t) \leqslant \frac{3}{2}\bar\lambda_l, l = \{2,3,\cdots,L\} \\ s_{\min}(W_l^{t+1} + \eta\tilde\Delta W_l^t - \eta\Delta W_l^t) \geqslant \frac{1}{2}\underline\lambda_l, l = \{3,\cdots,L\}. \end{cases}$$

For $l = 2$, we have

$$s_{\max}(W_2^{t+1} + \eta\tilde\Delta W_2^t - \eta\Delta W_2^t) \overset{(i)}{\leqslant} s_{\max}(W_2^{t+1}) + \eta\|\tilde\Delta W_2^t - \Delta W_2^t\|_F$$

$$\overset{(ii)}{=} s_{\max}(W_2^{t+1}) + \eta\|\tilde z_2^t - \tilde u_2^t\|$$

$$\overset{(iii)}{\leqslant} s_{\max}(W_2^{t+1}) + \eta n_1 n_2\|B^\top(Bu_2^r - z_2^r)\|_\infty$$

$$\overset{(iv)}{\leqslant} s_{\max}(W_2^{t+1}) + Hn_1 n_2^2\|\tilde B^\top\|_\infty\|Bu_2^r - z_2^r\|_\infty$$

$$\overset{(v)}{\leqslant} s_{\max}(W_2^{t+1}) + Hn_1 n_2^2\|\tilde B^\top\|_\infty\eta\tau\lambda\sqrt{(\alpha)^r f(\theta^0)}$$

$$\overset{(vi)}{\leqslant} s_{\max}(W_2^{t+1}) + Hn_1 n_2^2\|\tilde B^\top\|_\infty\eta\tau\lambda\sqrt{f(\theta^0)} \overset{(v)}{\leqslant} \bar\lambda_l + \frac{1}{2} + \frac{1}{2} = \frac{3}{2}\bar\lambda_2,$$

where (i) uses Weyl's inequality and the relationship between the Frobenius norm and $l_2$ norm of matrix; (ii) uses the definition in (36); (iii) uses the relationship between $l_2$ and $l_\infty$ norm and the definition of $\tilde u_2^t$ and $\tilde z_2^t$; (iv) uses the fact that $\|B^\top(Bu_2^r - z_2^r)\|_\infty \leqslant Hn_2\|\tilde B\|_\infty\|Bu_2^r - z_2^r\|_\infty$ and the definition of $\gamma^{r-1}$; (v) is from the induction assumption $\|Bu_2^r - z_2^r\|_\infty \leqslant \tau\gamma^{r-1}$; (vi) is from the fact that $\alpha \leqslant 1$; (v) is from the (59) and choice of $\tau$ and $\lambda$ in (50) and (47).

For $l \geqslant 3$, by Weyl's inequality, we have

$$s_{\max}(W_l^{t+1} + \eta\tilde\Delta W_l^t - \eta\Delta W_l^t) \overset{(i)}{\leqslant} s_{\max}(W_l^{t+1}) + \eta\|\tilde\Delta W_l^t - \Delta W_l^t\|_F$$

$$\overset{(ii)}{=} s_{\max}(W_l^{t+1}) + \eta\|z_l^t - u_l^t\| \overset{(iii)}{\leqslant} s_{\max}(W_l^{t+1}) + \eta n_{l-1} n_l\tau\lambda\sqrt{(\alpha)^t f(\theta^0)}$$

$$\overset{(iv)}{\leqslant} s_{\max}(W_l^{t+1}) + \eta n_{l-1} n_l\tau\lambda\sqrt{f(\theta^0)} \overset{(v)}{\leqslant} \bar\lambda_l + \frac{1}{4}\underline\lambda_l + \frac{1}{4}\underline\lambda_l \leqslant \frac{3}{2}\bar\lambda_l, \tag{62}$$

where (i) uses the Weyl's inequality on $W_l^{t+1}$ and $\eta(\tilde\Delta W_l^t - \Delta W_l^t)$; (ii) uses the update rule in (37); (iii) uses the relationship between $l_2$ and $l_\infty$ norm and the induction assumption; (iv) is because $\alpha < 1$; (v) uses (60) and choice of $\tau$ and $\lambda$ in (50) and (47).

Similarly,

$$s_{\min}(W_l^{t+1} + \eta\tilde\Delta W_l^t - \eta\Delta W_l^t) \overset{(i)}{\geqslant} s_{\min}(W_l^{t+1}) - \eta\|\tilde\Delta W_l^t - \Delta W_l^t\|_F$$

$$\overset{(ii)}{=} s_{\min}(W_l^{t+1}) - \eta\|z_l^t - u_l^t\| \overset{(iii)}{\geqslant} s_{\min}(W_l^{t+1}) - \eta n_{l-1} n_l\tau\lambda\sqrt{(\alpha)^t f(\theta^0)}$$

$$\overset{(iv)}{\geqslant} s_{\min}(W_l^{t+1}) - \eta n_{l-1} n_l\tau\lambda\sqrt{f(\theta^0)} \overset{(v)}{\geqslant} \frac{3}{4}\underline\lambda_l - \frac{1}{4}\underline\lambda_l = \frac{1}{2}\underline\lambda_l,$$

where (i) uses Weyl's inequality on $W_l^{t+1}$ and $\eta(\tilde\Delta W_l^t - \Delta W_l^t)$; (ii) plugs in the update rule in (37); (iii) uses the relationship between $l_2$ and $l_\infty$ norm and the induction assumption; (iv) is because $\alpha < 1$; (v) comes from (60) and choice of $\tau$ and $\lambda$ in (50) and (47).

**(Step 2.c)** We will show that:

$$f(\theta^t - \eta\tilde g^t) \leqslant (1 - \eta\phi)f(\theta^t).$$

From (1) and (2) we know

$$\max\left(s_{\max}(W_l^t), s_{\max}(W_l^t)\right) \leqslant \frac{3}{2}\bar{\lambda}_l.$$

Using the above relation, we can upper bound the differences of the gradients by using (45). More specifically, using the the definition of $Q_0$ in (49), we have

$$\|\mathrm{vec}(\nabla f(\theta^{t+1})) - \mathrm{vec}(\nabla f(\theta^t))\| \leqslant Q_0\|\theta^{t+1} - \theta^t\|,$$

where $Q_0$ is defined in (49).

Further, it is easy to verify that for any $\hat{\theta}^t$ between $\theta^t$ and $\theta^{t+1}$, we still have $s_{\max}(W_l(\hat{\theta}^t)) \leqslant \frac{3}{2}\bar{\lambda}_l$. So we can apply the same argument leading to (53), and obtain:

$$\|\mathrm{vec}(\nabla f(\hat{\theta}^t)) - \mathrm{vec}(\nabla f(\theta^t))\| \leqslant Q_0\|\hat{\theta}^t - \theta^t\|, \forall \hat{\theta}^t = \theta^t + \delta(\theta^{t+1} - \theta^t), \delta \in [0,1].$$

So we have

$$f(\theta^t - \eta\tilde{g}^t) \overset{(i)}{\leqslant} f(\theta^t) - \eta\langle u^t, \tilde{g}^t\rangle + \frac{Q_0}{2}\eta^2\|\tilde{g}^t\|^2$$

$$\overset{(ii)}{=} f(\theta^t) - \eta\langle u_2^t, \tilde{u}_2^t\rangle - \eta\sum_{l=3}^{L}\|u_l^t\|^2 + \frac{Q_0}{2}\eta^2\|\tilde{g}^t\|^2$$

$$\leqslant f(\theta^t) - \eta\langle u_2^t, \tilde{u}_2^t\rangle + \frac{Q_0}{2}\eta^2\|\tilde{g}^t\|^2,$$

where (i) uses Decent Lemma; (ii) expands $\langle u^t, \tilde{g}^t\rangle$ by utilizing the stacked structure. Similar to (56), with the induction assumption $s_{\min}(W_l^t) \geqslant \frac{1}{2}\underline{\lambda}_l, l \geqslant 3$, we have the following relation:

$$\langle u_2^t, \tilde{u}_2^t\rangle \geqslant \phi\|O_L^t - y\|^2.$$

Similar to (57), with the induction assumption $s_{\max}(W_l^t) \leqslant \frac{3}{2}\bar{\lambda}_l, l \geqslant 2$, we have

$$\|\tilde{g}^t\|^2 \leqslant \Lambda^2\|O_L^t - y\|^2.$$

If we choose $\eta \leqslant \frac{\phi}{Q_0\Lambda^2}$, we obtain

$$f(\theta^t - \eta\tilde{g}^t) \leqslant f(\theta^t) - 2\eta\phi f(\theta^t) + \eta\phi f(\theta^t) = (1 - \eta\phi)f(\theta^t).$$

**(Step 2.d)** We will show that:

$$f(\theta^{t+1}) \leqslant \alpha f(\theta^t).$$

We have the following relation:

$$f(\theta^{t+1}) = f(\theta^{t+1}) - f(\theta^t - \eta\tilde{g}^t) + f(\theta^t - \eta\tilde{g}^t)$$

$$\overset{(i)}{\leqslant} \frac{1}{2}\|G(\theta^{t+1}) - G(\theta^t - \eta\tilde{g}^t)\|^2 + \langle G(\theta^{t+1}) - G(\theta^t - \eta\tilde{g}^t), G(\theta^t - \eta\tilde{g}^t) - y\rangle$$

$$\quad + f(\theta^t + \eta\tilde{q}^t - \eta\tilde{g}^t)$$

$$\overset{(ii)}{\leqslant} \frac{1}{2}\|G(\theta^{t+1}) - G(\theta^t - \eta\tilde{g}^t)\|^2 + \frac{1}{\beta}\|G(\theta^{t+1}) - G(\theta^t - \eta\tilde{g}^t)\|^2 + \frac{\beta}{2}\|G(\theta^t t - \eta\tilde{g}^t) - y\|^2$$

$$\quad + f(\theta^t t - \eta\tilde{g}^t)$$

$$\overset{(iii)}{\leqslant} (\frac{1}{2} + \frac{1}{\beta})\|G(\theta^{t+1}) - G(\theta^t - \eta\tilde{g}^t)\|^2 + (1 + \beta)(1 - \eta\phi)f(\theta^t)$$

$$\overset{(iv)}{=} (\frac{1}{2} + \frac{1}{\beta})\langle \mathrm{vec}(\nabla G(\hat{\theta}^t)), -\eta\tilde{q}^t + \eta\tilde{g}^t\rangle^2 + (1 + \beta)(1 - \eta\phi)f(\theta^t)$$

$$\overset{(v)}{\leqslant} \eta^2(\frac{1}{2} + \frac{1}{\beta})\|\mathrm{vec}(\nabla G(\hat{\theta}^t))\|^2\|\tilde{q}^t - \tilde{g}^t\|^2 + (1 + \beta)(1 - \eta\phi)f(\theta^t),$$

where (i) expands and reorganizes the loss function; (ii) uses Young's inequality with constant $\beta$; (iii) uses the induction assumption $f(\theta^{t+1}) \leqslant (1 - \eta\phi)f(\theta^t)$; (iv) uses the mean value Theorem where $\hat{\theta}^t$ will be discussed in the next paragraph, the update rule in (37); (v) uses Cauchy-Schwartz inequality.

Now we discuss $\hat{\theta}^t$ and derive a bound for $\|\text{vec}(\nabla G(\hat{\theta}^t))\|$. By the Mean Value Theorem we have

$$\hat{\theta}^t = \theta^{t+1} + \delta(\eta\tilde{g}^t - \eta\tilde{q}^t),$$

for some $\delta \in [0, 1]$. By (61) and (62), we know that

$$s_{\max}(W_l(\theta^{t+1})) \leqslant \frac{5}{4}\bar{\lambda}_l, \; s_{\max}(W_l(\theta^t - \eta\tilde{g}^t)) \leqslant \frac{3}{2}\bar{\lambda}_l.$$

So it is easy to conclude that for $l \geqslant 3$,

$$s_{\max}(W_l(\hat{\theta}^t)) \overset{(i)}{\leqslant} s_{\max}(W_l(\theta^{t+1})) + \delta\eta\|z_l^t - u_l^t\|$$

$$\overset{(ii)}{\leqslant} s_{\max}(W_l^{t+1}) + \eta\|z_l^t - u_l^t\| \overset{(iii)}{\leqslant} \frac{3}{2}\bar{\lambda}_l,$$

where (i) uses Wely's inequality on $W_l^{t+1}$ and $\delta\eta\big(\tilde{\Delta}W_l^t - \Delta W_l^t\big)$; (ii) is because $\delta \in [0, 1]$; (iii) uses (62). We can derive the similar result for $l = 2$. By (42) without $\|O_L^t - y\|$, it is clear that

$$\|\text{vec}(\nabla G(\hat{\theta}^t))\| \leqslant \Lambda.$$

So we have

$$\eta^2(\frac{1}{2} + \frac{1}{\beta})\|\text{vec}(\nabla G(\hat{\theta}^t))\|^2\|\tilde{q}^t - \tilde{g}^t\|^2 + (1 + \beta)(1 - \eta\phi)f(\theta^t)$$

$$\overset{(i)}{\leqslant} \eta^2(\frac{1}{2} + \frac{1}{\beta})\Lambda^2(\|\tilde{u}_2^t - \tilde{v}_2^t\|^2 + \sum_{l=3}^{L}\|u_l^t - z_l^t\|^2) + (1 + \beta)(1 - \eta\phi)f(\theta^t)$$

$$\overset{(ii)}{\leqslant} \eta^2(\frac{1}{2} + \frac{1}{\beta})\Lambda^2(n_1^2 n_2^2\|\tilde{u}_2^t - \tilde{z}_2^t\|_\infty^2 + \sum_{l=3}^{L}n_{l-1}^2 n_l^2\|u_l^t - z_l^t\|_\infty^2) + (1 + \beta)(1 - \eta\phi)f(\theta^t)$$

$$\overset{(iii)}{\leqslant} \eta^2(\frac{1}{2} + \frac{1}{\beta})\Lambda^2(H^2 n_1^4 n_2^2\|\tilde{B}^\top\|_\infty^2\|Bu_2^t - z_2^t\|_\infty^2 + \sum_{l=3}^{L}n_{l-1}^2 n_l^2\|u_l^t - z_l^t\|_\infty^2) + (1 + \beta)(1 - \eta\phi)f(\theta^t)$$

$$\overset{(iv)}{\leqslant} \eta^2(\frac{1}{2} + \frac{1}{\beta})\Lambda^2\left(H^2 n_1^4 n_2^2\|\tilde{B}^\top\|_\infty^2\tau^2\lambda^2(\alpha)^t f(\theta^0) + \sum_{l=3}^{L}n_{l-1}^2 n_l^2\tau^2\lambda^2(\alpha)^t f(\theta^0)\right) + (1 + \beta)(1 - \eta\phi)f(\theta^t)$$

$$= \left((1 + \beta)(1 - \eta\phi) + \eta^2(\frac{1}{2} + \frac{1}{\beta})(H^2 n_1^4 n_2^2\|\tilde{B}^\top\|_\infty^2 + \sum_{l=3}^{L}n_{l-1}^2 n_l^2)\Lambda^2\tau^2\lambda^2\right)(\alpha)^t f(\theta^0),$$

(i) uses (42) with the fact that $s_{\max}(W_l(\hat{\theta}^t)) \leqslant \frac{3}{2}\bar{\lambda}_l$, and expands $\|\tilde{q}^t - \tilde{g}^t\|^2$; (ii) uses the relationship between $l_2$ and $l_\infty$ norm; (iii) uses the fact that $\|B^\top(Bu_2^r - z_2^r)\|_\infty \leqslant Hn_2\|\tilde{B}\|_\infty\|Bu_2^r - z_2^r\|_\infty$; (iv) uses the induction assumption $\|Bu_2^t - z_2^t\| \leqslant \tau\gamma^{t-1}$, $f(\theta^t) \leqslant (\alpha)^t f(\theta^0)$ and the definition of $\gamma^{t-1}$.

Let $\beta = \frac{\frac{1}{2}\eta\phi}{1-\eta\phi}$, we have

$$\left((1 + \beta)(1 - \eta\phi) + \eta^2(\frac{1}{2} + \frac{1}{\beta})(Hn_1 n_2^2\|\tilde{B}^\top\|_\infty^2 + \sum_{l=3}^{L}n_{l-1}^2 n_l^2)\Lambda^2\tau^2\lambda^2\right)(\alpha)^t f(\theta^0)$$

$$\overset{(i)}{\leqslant} \left(1 - \frac{1}{2}\eta\phi + \eta^2(\frac{1}{2} + \frac{2(1 - \eta\phi)}{\eta\phi})(Hn_1 n_2^2\|\tilde{B}^\top\|_\infty^2 + \sum_{l=3}^{L}n_{l-1}^2 n_l^2)\Lambda^2\tau^2\lambda^2\right)(\alpha)^t f(\theta^0)$$

$$\overset{(ii)}{\leqslant} (1 - \frac{1}{4}\eta\phi)(\alpha)^t f(\theta^0) = (\alpha)^{t+1} f(\theta^0),$$

where (i) plugs in the choice of $\beta$; (ii) is because the choice of $\tau$ and $\lambda$ in (50) and (47).

**(Step 2.e)** We will show that:

$$\begin{cases} \|Bu_l^{t+1} - z_l^{t+1}\|_\infty \leqslant \tau\gamma^t, \; l = 2 \\ \|u_l^{t+1} - z_l^{t+1}\|_\infty \leqslant \tau\gamma^t, \; l = \{3, \cdots, L\}. \end{cases}$$

For $l = 2$, we have

$$\|Bu_2^{t+1} - z_2^t\|_\infty \overset{(i)}{\leqslant} \|Bu_2^{t+1} - Bu_2^t\| + \|Bu_2^t - z_2^t\|_\infty \overset{(ii)}{\leqslant} s_{\max}(\tilde{B})\|u_2^{t+1} - u_2^t\| + \|Bu_2^t - z_2^t\|_\infty$$

$$\overset{(iii)}{\leqslant} s_{\max}(\tilde{B})\|u^{t+1} - u^t\| + \|Bu_2^t - z_2^t\|_\infty$$

$$\overset{(iv)}{\leqslant} s_{\max}(\tilde{B})\|\text{vec}(\nabla f(\theta^{t+1})) - \text{vec}(\nabla f(\theta^t))\| + \|Bu_2^t - z_2^t\|_\infty$$

$$\overset{(v)}{\leqslant} s_{\max}(\tilde{B})Q_0\|\theta^{t+1} - \theta^t\| + \|Bu_2^t - z_2^t\|_\infty$$

$$\overset{(vi)}{=} \eta s_{\max}(\tilde{B})Q_0\|\tilde{q}^t\| + \|Bu_2^t - z_2^t\|_\infty$$

$$\overset{(vii)}{\leqslant} \eta s_{\max}(\tilde{B})Q_0(\|\tilde{g}^t\| + \|\tilde{g}^t - \tilde{q}^t\|) + \|Bu_2^t - z_2^t\|_\infty$$

$$\overset{(viii)}{\leqslant} \eta s_{\max}(\tilde{B})Q_0\Lambda\|O_L^t - y\| + \eta s_{\max}^2(\tilde{B})Q_0\|g^t - q^t\| + \|Bu_2^t - z_2^t\|_\infty$$

$$\overset{(ix)}{\leqslant} \eta s_{\max}(\tilde{B})Q_0\Lambda\|O_L^t - y\| + \eta s_{\max}^2(\tilde{B})Q_0\sum_{l=3}^{L}\|u_l^t - z_l^t\| + \eta s_{\max}^2(\tilde{B})Q_0\|Bu_2^t - z_2^t\| + \|Bu_2^t - z_2^t\|_\infty$$

$$\overset{(x)}{\leqslant} \eta s_{\max}(\tilde{B})Q_0\Lambda\|O_L^t - y\| + \eta s_{\max}^2(\tilde{B})Q_0 Hn_2\|u_2^t - z_2^t\|_\infty + \eta s_{\max}^2(\tilde{B})Q_0\sum_{l=3}^{L}n_{l-1}n_l\|u_l^t - z_l^t\|_\infty + \|Bu_2^t - z_2^t\|_\infty$$

$$\overset{(xi)}{\leqslant} \left(1 + \eta s_{\max}^2(\tilde{B})Q_0\big(Hn_2 + \sum_{l=3}^{L}n_{l-1}n_l\big)\right)\tau\gamma^{t-1} + \eta s_{\max}(\tilde{B})Q_0\Lambda\|O_L^t - y\|$$

$$\overset{(xii)}{\leqslant} \lambda\sqrt{(\alpha)^t f(\theta^0)}\left(\tau + \tau\eta s_{\max}^2(\tilde{B})Q_0\big(Hn_2 + \sum_{l=3}^{L}n_{l-1}n_l\big) + \frac{\sqrt{2}\eta s_{\max}(\tilde{B})Q_0\Lambda}{\lambda}\right)$$

$$\overset{(xiii)}{\leqslant} \lambda\sqrt{\alpha}\sqrt{(\alpha)^t f(\theta^0)} = \lambda\gamma^t,$$

where (i) uses triangle inequality; (ii) extracts the largest singular value of $B$, (iii) uses the fact $\|u^{t+1} - u^t\| \geqslant \|u_2^{t+1} - u_2^t\|$; (iv) uses the definition of $u^t$; (v) upper bounds the differences of the gradients by using (45) (vi) uses the update rule in (37); (vii) uses triangle inequality; (viii) uses the upper bound of gradient norm in (42), and it extracts the largest singular value of $B$ and definition of $\Lambda$ in (48); (ix) uses the stacked structure of $g^t$ and $q^t$ to expand $\|g^t - q^t\|$; (x) uses the relationship between $l_2$ and $l_\infty$ norm; (xi) reorganizes the terms and uses induction assumption; (xii) uses the definition of $\gamma^{t-1}$; (xiii) is from the choice of $\tau$ and $\lambda$ in (50) and (47).

Then by Lemma 4.1, with the correspondence that $c = Bu_2^{t+1}, p = z_2^t, r = \gamma^t$, we can obtain

$$\|z_2^{t+1} - Bu_2^{t+1}\| \overset{(i)}{=} \|\text{quant}(Bu_2^{t+1}, z_2^t, \gamma^t, b) - Bu_2^{t+1}\|_\infty \overset{(ii)}{\leqslant} \tau\gamma^t.$$

where $(i)$ comes from the 'Quantize' step in Algorithm 1; $(ii)$ is from the first relation in Lemma 4.1.

For $l \geqslant 3$, we have

$$\|u_l^{t+1} - z_l^t\|_\infty \overset{(i)}{\leqslant} \|u_l^{t+1} - u_l^t\| + \|u_l^t - z_l^t\|_\infty \overset{(ii)}{\leqslant} \|u^{t+1} - u^t\| + \|u_l^t - z_l^t\|_\infty$$

$$\overset{(iii)}{=} \|\text{vec}(\nabla f(\theta^{t+1})) - \text{vec}(\nabla f(\theta^t))\| + \|u_l^t - z_l^t\|_\infty$$

$$\overset{(iv)}{\leqslant} \eta Q_0\|\tilde{q}^t\| + \|u_l^t - z_l^t\|_\infty$$

$$\overset{(v)}{\leqslant} \eta Q_0(\|\tilde{g}^t\| + \|\tilde{g}^t - \tilde{q}^t\|) + \|u_l^t - z_l^t\|_\infty$$

$$\overset{(vi)}{\leqslant} \eta Q_0\Lambda\|O_L^t - y\| + \eta s_{\max}(\tilde{B})Q_0\|g^t - q^t\| + \|u_l^t - z_l^t\|_\infty$$

$$\overset{(vii)}{\leqslant} \eta Q_0\Lambda\|O_L^t - y\| + \eta s_{\max}(\tilde{B})Q_0\sum_{l=3}^{L}\|u_l^t - z_l^t\| + \eta s_{\max}(\tilde{B})Q_0\|Bu_2^t - z_2^t\| + \|u_2^t - z_2^t\|_\infty$$

$$
\overset{(viii)}{\leqslant} \; \eta Q_0\Lambda\|O_L^t - y\| + \eta s_{\max}(\tilde{B})Q_0 H n_2\|Bu_2^t - z_2^t\|_\infty + \eta s_{\max}(\tilde{B})Q_0\sum_{l=3}^{L}n_{l-1}n_l\|u_l^t - z_l^t\|_\infty + \|u_2^t - z_2^t\|_\infty
$$

$$
\overset{(ix)}{\leqslant} \; \left(1 + \eta s_{\max}(\tilde{B})Q_0\big(Hn_2 + \sum_{l=3}^{L}n_{l-1}n_l\big)\right)\tau\gamma^{t-1} + \eta Q_0\Lambda\|O_L^t - y\|
$$

$$
\overset{(x)}{\leqslant} \; \lambda\sqrt{f(\theta^t)}\left(\tau + \tau\eta s_{\max}(\tilde{B})Q_0\big(Hn_2 + \sum_{l=3}^{L}n_{l-1}n_l\big) + \frac{\sqrt{2}\eta Q_0\Lambda}{\lambda}\right)
$$

$$
\overset{(xi)}{\leqslant} \; \lambda\sqrt{\alpha}\sqrt{f(\theta^t)} = \lambda\gamma^t,
$$

where (i) uses triangle inequality; (ii) uses the fact $\|u^{t+1} - u^t\| \geqslant \|u_2^{t+1} - u_2^t\|$; (iii) uses the definition of $u^t$;(iv) uses the update rule in (37); (v) uses triangle inequality; (vi) uses the upper bound of gradient norm in (42), extracts the largest singular value of $B$ and uses the definition of $\Lambda$ in (48); (vii) uses the stacked structure of $g^t$ and $q^t$ to expand $\|g^t - q^t\|$; (viii) uses the relationship between $l_2$ and $l_\infty$ norm; (ix) uses the induction assumption; (x) uses the definition of $\gamma^{t-1}$; (xi) is because the choice of $\tau$ and $\lambda$ in (50) and (47).

Then by Lemma 4.1, with the correspondence that $c = Bu_2^{t+1}, p = z_2^t, r = \gamma^t$, we can obtain

$$
\|z_l^{t+1} - u_l^{t+1}\| \overset{(i)}{=} \|\text{quant}(u_l^{t+1}, z_l^t, \gamma^t, b) - Bu_l^{t+1}\|_\infty \overset{(ii)}{\leqslant} \tau\gamma^t,
$$

where $(i)$ comes from the 'Quantize' step in Algorithm 1; $(ii)$ is from the first relation in Lemma 4.1.

Now we have proved that (52) holds. So for $t > 0$, there is

$$
f(\theta^t) \leqslant (\alpha)^t f(\theta^0), \quad \text{where} \quad \alpha = 1 - \frac{1}{4}\eta\phi = 1 - \frac{\phi^2}{4Q_0\Lambda^2}.
$$

Thus, if we want the objective function to compute an $\epsilon$-optimal solution, the total number of iterations is $\log(f(\theta^0)/\epsilon)/\log(1/(1 - \frac{\phi^2}{4Q_0\Lambda^2}))$. Since in each iteration, each agent $k$ transmits a length-$H$ vector $q_k^t$, so we conclude that the total number of bits each node needs to communicate is $\log(f(\theta^0)/\epsilon)/\log(1/(1 - \frac{\phi^2}{4Q_0\Lambda^2}))$ bits. Notice that $\log(1/(1 - \frac{\phi^2}{4Q_0\Lambda^2})) = -\log(1 - \frac{\phi^2}{4Q_0\Lambda^2}) \sim \frac{4Q_0\Lambda^2}{\phi^2}$, so we can derive the simplified total number of bits as $b \cdot \frac{4Q_0\Lambda^2}{\phi^2} \cdot \log(f(\theta^0)/\epsilon)$. $\qquad\square$