# OpenReview forum: "Distributed Optimization for Overparameterized Problems: Achieving Optimal Dimension Independent Communication Complexity"
_NeurIPS.cc/2022/Conference — NeurIPS 2022 Accept_

### Official Review · Reviewer_zULq · 2022-07-10

**Rating:** 4
**Confidence:** 4
**Soundness:** 3 good
**Presentation:** 2 fair
**Contribution:** 2 fair

**Summary:**

This paper considers distributed (centralized and decentralized) smooth nonconvex (PL) minimization problems in the general case and in the special (quadratic) overparameterized case. The authors give lower bounds on the amount of information transferred in terms of dimension, number of agents, amount of data, and solution accuracy. Then the authors give an algorithm by which these lower bounds can be achieved. The algorithm uses an unusual but not new quantization. In general case (Section 4.1), the algorithm is no different from those already encountered in the community. In the quadratic-overparameterized case, the key is the use of an additional transformation $F$ and an inverse $\tilde F$. The authors also give a decentralized version of the algorithm using a gossip protocol with compression.

**Questions:**

Please, see section "Strengths And Weaknesses"

**Limitations:**

No limitations and potential negative societal impact

**Strengths And Weaknesses:**

I was generally interested in dealing with this work. I was left with a mixed impression of the work. Below I will outline my thoughts, questions, and comments in the order of the paper.

0) The paper is fairly easy to read, in some cases the presentation could have been improved (see below), but overall I had no problem getting into the results.

1) Literature review: it seemed to me that a literature review could be more complete. In particular, one of the most active authors in the field of compressed optimization is P Richtarik. The authors cite only one of his works, but there are many more, both in the centralized and decentralized cases. This is important not only to note Richtarik's contributions, but also to compare your results with those available in the literature (see below, point 3).

2) Lower bounds: I think it is important here to clarify which factors are important to us when deriving lower bounds: $N$, $K$, $D$, $e$, $L$, $\mu$ and $B$, where $B$ is the number of bits used in the current data type (e.g. 32 or 64 bits). If I understand correctly, the bounds take into account the presence of $B$, but do not depend on it. I think it is important to reflect this.

3) For me, the compressor from Definition 4 turned out to be new. It is not an invention of the authors, but I read the original paper about it. Thanks for this experience!

4) Upper bounds in the $C_{pl}$ case: my feeling is that these are not good bounds. The authors' estimate (in terms of $L$, $M$, $\mu$) is equal to $\mathcal{O}\left( \frac{L^2 M^2}{\mu^4}\right)$, this is a very large number. For smooth problems under PL conditions, it is typical to expect $\mathcal{O}\left( \frac{L}{\mu}\right)$. For example, the compression method from [1] has such results. If we remember about communications, then the estimate (in bits) from the work [1]  looks like
$$
\mathcal{O}\left( \frac{BD}{\sqrt{K}} \cdot \frac{L}{\mu}\right),
$$
where $K$ is a number of agents, $B$ is the number of bits used in the current data type (e.g. 32 or 64 bits). This is the square root of $K$ times better than for the uncompressed method. The authors' estimate is
$$
\mathcal{O}\left( D \log_2 D \cdot \frac{L^2 M^2}{\mu^4}\right)
$$
Even if we forget about $L$,$\mu$ and $M$, it is not obvious to me which is smaller $\log_2 D$ or $\frac{B}{\sqrt{K}}$. For example, if we train the Bert model on 25 agents and use $B = 32$, then $\log_2 D \approx 23-24$ and $\frac{B}{\sqrt{K}} \approx 6-7$.That's why I stress that comparison with other results is very important! There are many works, not only [1].

5) The overparameterized case: this is the first time I met such an overparameterized assumption (Definition 2), although I saw many papers where simpler things are introduced as overparameterization. Therefore, Definition 2 is very specific and in the literature it occurred only 1 time. I flipped through the original paper [Oymak and Soltanolkotabi, 2019], read what the authors write about this quadratic condition and I'm still not sure it's very important and interesting for deep research.  My concerns are related to the fact that despite the improvements in convergence in terms of $N$ and $D$, in terms of $L$ and $\mu$ we are far from good convergence estimates, namely $\mathcal{O}\left( \frac{L^4}{\mu^4}\right)$. And this effect is noticeable both in the original paper and in the one under review.
At the moment the most important point of this paper is to understand how interesting this condition and the results around it: how wide a class of problems satisfies it (perhaps the authors will be able to add something here besides what is written in the work), what order of gain can be obtained for certain problems (due to the fact that we change $D$ to $N$, but lose significantly in $L/\mu$) and so on. Now it seems to me that the game is not worth the candle.

6) Decentralized results seemed to me bonus and not particularly interesting, it uses a multistep gossip protocol with compression, and the authors uses the classic sign compression, not special from Definition 4. The essence of this method is that it is a centralized method, but with a slight inexactness in averaging. It is obvious to me that this can be done.

**Conclusion:**

I am looking forward to the discussion process with the authors and fellow reviewers to discuss this work. This is important to me because so far I have mixed feelings, many questions, and have not taken a clear view of the paper. At this point, I will give a borderline reject to motivate  the authors to take an active part in the discussion.

[1] Gorbunov E. et al. MARINA: Faster non-convex distributed learning with compression //International Conference on Machine Learning. – PMLR, 2021. – С. 3788-3798.

---

> ### Author Response · Authors · 2022-08-02
> **Response to Reviewer zULq(Part1)**
>
>  Thank you for your comments and concerns.
>
> 1. Literature review: it seemed to me that a literature review could be more complete. In particular, one of the most active authors in the field of compressed optimization is P Richtarik. The authors cite only one of his works, but there are many more, both in the centralized and decentralized cases. This is important not only to note Richtarik's contributions, but also to compare your results with those available in the literature (see below, point 3).
>
> **[Response]** We have added some related works in the revised version. Please see our updated References. Thanks!
>
>
> 2. Lower bounds: I think it is important here to clarify which factors are important to us when deriving lower bounds: N, K, D, e, L, μ and B, where B is the number of bits used in the current data type (e.g. 32 or 64 bits). If I understand correctly, the bounds take into account the presence of B, but do not depend on it. I think it is important to reflect this.
>
> **[Response]** The factors that are involved in the derivation of the lower bounds are the number of nodes $K$, the solution accuracy $\epsilon$, and the problem dimension $D$. In the PL case the parameters $L, \mu$ are constant, specifically $\mu_k=1/8$ and $L=3K$, where $K$ is the number of nodes. Note that these constants are independent of the dimension $D$ and the number of data samples $N$. Deriving lower bounds for problems with arbitrary $L$ and $\mu$ parameters requires a more detailed analysis, and that is something we might consider in our future works.
>
> Also, our lower bounds express the communication complexity as a number of bits. So, we are not relying on any specific data structure representation at all. We are simply characterizing what is the smallest number of bits needed to be communicated before reaching an $\epsilon$ approximate solution. How to achieve this (i.e., using which data type), is out of the scope of our lower bound characterization.
>
>
> 3. For me, the compressor from Definition 4 turned out to be new. It is not an invention of the authors, but I read the original paper about it. Thanks for this experience!
>
> **[Response]** Good to know. Thanks!

---

> > ### Comment · Reviewer_zULq · 2022-08-05
> > **Reply (Part 1)**
> >
> > Thanks to the authors for their response!
> >
> > 1) Good! Thanks! But what about comparison? I don't see it, for example with [1] (Gorbunov et al.).
> >
> > 2) Let me try to explain again, the upper bound for distributed GD without compression under the PL assumption is
> > $$
> > O\left(BDK\frac{L}{\mu}\right),
> > $$
> > where  $B$ is the number of bits used in the current data type. If one ignore factors $L,\mu, B$, the upper bound is
> > $$
> > O\left(DK\right).
> > $$
> > It meets (and outperforms!!!) authors' lower bound. It looks strange. And we don't need any compression and so on. I understand that authors ignore $L, \mu$. But do they ignore also $B$?

---

> > > ### Author Response · Authors · 2022-08-07
> > > **Response to Reply (Part 1)**
> > >
> > > ## Question (Part 1):
> > > Thanks to the authors for their response!
> > >
> > > Good! Thanks! But what about comparison? I don't see it, for example with [1] (Gorbunov et al.).
> > >
> > > Let me try to explain again, the upper bound for distributed GD without compression under the PL assumption is
> > >
> > > $$O(BDK\frac{L}{\mu})$$
> > >
> > > where $B$ is the number of bits used in the current data type. If one ignore factors $L, \mu, B$, the upper bound is
> > > $$O(DK)$$
> > > It meets (and outperforms!!!) authors' lower bound. It looks strange. And we don't need any compression and so on. I understand that authors ignore $L$, $\mu$. But do they ignore also $B$?
> > >
> > > ## Reply(Part 1):
> > > We thank the reviewer for the prompt reply.
> > >
> > > **[About References].** We have added to Appendix A a few more references.
> > >
> > > We have made additional comparisons in the revised manuscript with [1]; please see Appendix A. Below is the newly added text:
> > >
> > >
> > > "Recent work [1] has analyzed the communication efficient algorithm to solve functions that satisfy PL conditions. Howevver, it has been focused on analyzing the number of communication rounds needed to achieve certain $\epsilon$ optimal solution, while the current paper is focused on finding the minimum bits to be communicated. Therefore, although the two works are both about developing communication efficient algorithms for PL functions, the bounds obtained in these works represent different physical quantities, thus cannot be directly compared."
> > >
> > >
> > > **[Comparison to [1]].** We respectfully disagree with the claim that the algorithm in [1] can "achieve the lower bound without compression". It should be emphasized again that our paper focuses on **bit complexity**, while [1] focuses on the **round complexity**. Thus, our work and [1] discuss two different aspects of communication efficient distributed optimization. It can be unfair to **directly** compare the two bounds. To be more specific, our aim is to explicitly compute the number of **bits** that are needed to be communicated to obtain a sufficiently accurate results; while in [1] no such bit complexity result has been mentioned (see, e.g Theorem 3.2). In fact, it is technically **wrong** to multiply a round complexity bound (such as $D K \frac{L}{\mu}$) by an arbitrary number of bits $B$ (the current data type, as referred to by the reviewer), to obtain bit complexity.
> > >
> > > A simple example: consider a distributed algorithm that requires $C_0/\epsilon$ rounds of communication (of real numbers) to achieve $\epsilon$ optimality (with $C_0$ being a constant). Let us implement the algorithm proposed in [1], but use the binary data type, that is, B=1 (that is, we are only allowed to transmit the sign for each entry). Can we say that the proposed algorithm still requires $C_0 D/\epsilon$ bits to achieve $\epsilon$ optimality, where $D$ is the problem's dimension? Obviously not, since if the transmitted vectors are binary, it is not clear if the algorithm will even converge. See for example, the line of works about signSGD,e.g [Karimireddy, Sai Praneeth, et al. "Error feedback fixes signsgd and other gradient compression schemes." International Conference on Machine Learning. PMLR, 2019.], where Counterexample 1,2,3 show signSGD will not converge in some standard settings.
> > >
> > >
> > > From the above discussion, hopefull it is clear that one **should not** directly multiply the  bits $B$ representing data types to directly get the bits needed for optimization. In contrary, what we have proposed is different. Based on Algorithm 1, at each iteration $t$, each client $k$ is **assigned with $b$ bits** to quantize the local gradient $g_k^t$ as $q_k^t$, where $b=O(\log(D))$ is derived in our analysis. We utilize an **adaptive quantization method** (See Algorithm 1 and Definition 4) to quantize the gradient, where the centers of the grids, as well as the size of the grids, are adapting to the current gradient vectors.
> > >
> > > Summarizing the above, one canont  directly compare the bounds in [1] (by multipling an arbitrary number $B$), with the bound obtained in our paper. It is like comparing apple with orange.

---

> ### Author Response · Authors · 2022-08-02
> **Response to Reviewer zULq(Part2)**
>
>
>
> 4.  Quality of the bound in Theorem 4.1 and comparison with the bound in [1].
>
> **[Response]** Thank you for raising the question about the condition number. In the revised manuscript, we have slightly changed our proof, and derived a new bound $\mathcal{O}(\frac{L^{\frac{3}{2}}}{\mu})$; please see Theorem 4.1 for the statement, as well as Appendix C.2 for the proof.
>
> Further, let us emphasize that, the focus of our work has been given to the case where D is large, while $L,\mu$ are moderate and are treated as constants. Therefore, we have not attempted to design new algorithms and/or sharpen the dependency on the condition number.
>
> Next, let us address the question regarding to the bound $\mathcal{O}\left(\frac{B D}{\sqrt{K}} \cdot \frac{L}{\mu}\right)$. This bound comes from reference [1], which designs a novel communication compression strategy and provides upper bound of communication complexity. We have the following points:
> a) Upon careful inspection of [1], we find that it only characterized **round complexity** without providing **bit complexity**. First of all, to be rigorous, when characterizing convergence and communication complexity, one cannot directly multiply the bit length $B$ with the total number of rounds to obtain the bit-communication complexity, since after quantization the algorithm has changed, and the previous convergence analysis may no longer hold.
>
> b) For the sake of argument, let’s assume that this approach is OK, let us further inspect the result in [1]. Our conclusion is that the bound in [1] has **no actual dependency on the number of clients K**(or in [1], it is n), so the bound $\mathcal{O}\left(\frac{B D}{\sqrt{K}} \cdot \frac{L}{\mu}\right)$ you mentioned does not accurately describe the bound in [1].
> To be more specific,the comparable result of our Theorem 4.1 in [1] is Theorem 2.2, saying that we need the following rounds to achieve $\epsilon$ optimal solution
> $\mathcal{O}\left(\max \left\\{\frac{1}{p}, \frac{L}{\mu}\left(1+\sqrt{\frac{(1-p) \omega}{p K}}\right)\right\\} \log \frac{\Delta_{0}}{\varepsilon}\right)$
> where $p$ is the probability that the gradient is compressed and $K$ is the number of clients (in [1] it is $n$, here for consistency we use $K$ to denote the # of clients). This result means the complexity is controlled by both $p$ and $K$  given $L/\mu$ is constant. We notice two points: 1. The first term in the max function, $1/p$, is important and $1/p$ needs to be $\mathcal{O}(1)$. So even if K can be chosen large, there is still a $1/p$ factor; 2. The second term in the max contains a additive constant $1$, meaning even $K$ is chosen to be large, the second term $\frac{L}{\mu}\left(1+\sqrt{\frac{(1-p) \omega}{p K}}\right)\sim\mathcal{O}(\frac{L}{\mu})$, which cannot be scaled by $1/\sqrt{K}$. Thus, we can conclude that the bound in [1] cannot be scaled by $\frac{1}{\sqrt{K}}$ anyway.
> To conclude, it is worth mentioning that:1. Our analysis **directly** considers the quantized algorithms, which is different than what [1] does; 2. The bound in [1] actually cannot be scaled by $\frac{1}{\sqrt{K}}$. Further, we are not aware of any other work that computes bit-communication complexity for overparameterized problems, or problems with PL loss functions, so it is difficult to compare fairly.

---

> > ### Comment · Reviewer_zULq · 2022-08-05
> > **Reply (Part 2)**
> >
> > 1) New bound looks strange, I expect to see $L^{3/2}/\mu^{3/2}$ or  $L/\mu$. Something goes
> > wrong with the physical dimensions. I mean, let $f(x)$ be measured in meters, $f'(x)$ is meters divided by seconds, then $L$ and $\mu$ are $1$ divided by seconds. Then in the final number of iterations appears $1$ divided by seconds$^{1/2}$.
> >
> > 2) It seems that the authors have not sufficiently checked the article [1]. Look at Definition 1.1. The authors define $\zeta$, which is the quality of compressor in bits (authors of [1] write in coordinates, but in fact it means bits). The estimate that the authors (the article under review) wrote out in their response to me is the iterative complexity of the algorithm. To get the complexity of the algorithm in bits one need to divide the iterative complexity by $\zeta$. If the authors take $\zeta > K$, they get the estimate I wrote out in my review.

---

> > > ### Author Response · Authors · 2022-08-07
> > > **Response to Reply(Part 2)Question 1**
> > >
> > > ## Reply(Part 2):
> > > ## Question
> > > >>New bound looks strange, I expect to see $L^{3/2} / \mu^{3/2}$ or $L/\mu$. Something goes wrong with the physical dimensions. I mean, let $f(x)$ be measured in meters, $f'(x)$ is meters divided by seconds, then $L$ and $\mu$ are 1 divided by seconds. Then in the final number of iterations appears $1$ divided by seconds$^{1/2}$.
> > >
> > > **[Bounds relating to $L, \mu$].**  We agree that the additional factor of $\sqrt{L}$ in the ratio $\frac{L^{\frac{3}{2}}}{\mu}$ is not usual. We typically do not want to see such an additional factor, since this means that one can arbitrarily scale up or down the problem (so that L will be scaled up and down accordingly), to obtain very different complexity bounds.
> > >
> > > However, this is not the case in our analysis. Please note that in our analysis, we have explicitly imposed the conditions that $\mu<1$ and $L>1$ throughout the paper (see the paragraph under Lemma 4.2). This means that the entire problem has already been somewhat "scaled". In particular, since $\mu$ cannot be greater than 1, this means that $L$ will **not** be arbitrarily large. Therefore the bound $\frac{L^{3/2}}{\mu}$ can be viewed as $\kappa\times L^{1/2}$, where  the $\kappa = L/\mu$ is some condition number which is *scale-invariant*, while there is an additional constant $L^{1/2}$ that can change with problem scale. How big this constant can be? It is easy to see that under the constraint $\mu<1$, $L>1$, in the worst case one can scale the problem by $O(1/\mu)$, so this $L^{1/2}$ factor is upper bounded by $\kappa^{1/2}$. Therefore we can conclude that with the constraint $\mu<1$, $L>1$, we should simply upper bound the ratio $\frac{L^{3/2}}{\mu}$ by $\frac{L^{3/2}}{\mu^{3/2}}$.
> > >
> > > Finally, we agree that our analysis on $L,\mu$ may not be optimal since our focus is on the dependency of $D$ regarding the upper/lower bound.

---

> > > ### Author Response · Authors · 2022-08-07
> > > **Response to Reply(Part 2)Question 2**
> > >
> > > ## Reply(Part 2)
> > >
> > > Question
> > >
> > > >>It seems that the authors have not sufficiently checked the article [1]. Look at Definition 1.1. The authors define $\zeta$, which is the quality of compressor in bits (authors of [1] write in coordinates, but in fact it means bits). The estimate that the authors (the article under review) wrote out in their response to me is the iterative complexity of the algorithm. To get the complexity of the algorithm in bits one need to divide the iterative complexity by $\zeta$. If the authors take $\zeta > K$, they get the estimate I wrote out in my review.
> > >
> > > **[Comparion with [1]]**  We have re-examined the results in [1]. While we agree with the reviewer that the communication complexity bounds in [1] could decrease with $K$, we would like to argue that it may not be fair to compare these bounds.
> > >
> > > Let us directly investigate the communication cost complexiy bounds given in the appendix of [1] again. Our conclusion is that, upon careful investigation, the per-worker complexity could be decreasing w.r.t. $K$ (say a scaling of $O(1/K^{1/3})$). This means that the total complexity scales with $O(K^{2/3} D)$, which may seem to contradict our lower bound of $\Omega(KD)$. However, since the algorithm and the upper bounds discussed in [1] are **randomized** algorithms, while our lower and upper bounds are for **deterministic** algorithms, they again **cannot be compared directly**.
> > >
> > >
> > > To be more specific, we believe that the reviewer has been referring to to Corollary C.2 of [1], which directly studies the communication cost (something proportional to the non-zero entry transmitted, defined in [1]) given by
> > > $$\mathcal{O}\left(D+\max \left\\{{D}, \frac{L}{\mu}\left(\zeta_\cal{Q}+\sqrt{\frac{\omega\zeta\_{\mathcal{Q}}}{ K}\left(D-\zeta\_{\mathcal{Q}}\right)}\right)\right\\} \log \frac{\Delta\_{0}}{\varepsilon}\right).$$
> > >
> > > Let $\zeta_\mathcal{Q} = \mathbb{E}[ || {\cal Q}(x) ||_0 ] = \alpha D$ with $0 \leq \alpha \leq 1$ and note that it can be achieved while satisfying Def. 1.1 with a random sparsifier of the form ${\cal Q}(x) = (1/\alpha) {\tt randsparse}(x)$ such that ${\tt randsparse}(x)$ picks $\alpha D$ coordinates from $x$ uniformly at random. Note that the factor $(1/\alpha)$ is necessary for $\mathbb{E}[{\cal Q}(x)]=x$. For this sparsifier, we note that $\omega$ in Def. 1.1. may grow as large as $O(1/\alpha^2)$. Under the above premises, the second term in the max becomes:
> > > $$\frac{L}{\mu} \left(\alpha D + D \sqrt{(1-\alpha)/(\alpha K)} \right).$$ So the best choice is $\alpha = 1/K^{1/3}$, yielding the bound $O(\frac{L}{\mu} \frac{D}{K^{1/3}})$.
> > >
> > > With that being said, we need to emphasize tha the algorithm discussed in [1] is a **randomized** algorithm, where the compression operator $\mathcal{Q}(\cdot)$, as well as the algorithm itself, are random. Therefore, the optimality measure provided in the paper is $\mathbb{E}[f(x^r)-f^*]\le \epsilon$. On the contrary, **both** the lower bound and the algorithm provided in this work is about **deterministic** protocols; see also see  Theorem 3.5 of [Vempala et al 2020], upon which our lower bound result is built. Therefore, even though the randomized algorithm might outperform the lower bound, there is no contradiction. In fact, there are numerous cases in the literature, in which randomized algorithms can (on average) outperform deterministic algorithms. As a well-known example in optimization, a deterministic alternating direction method of multipliers (ADMM) *diverge* for certain special 3-block algorithms, while a randomized ADMM always converges in expectation; see ["On the Expected Convergence of Randomly Permuted ADMM", Sun et al, 2015.]. Of course, it will be an interesting direction to consider the lower bounds for randomized distributed algorithms.
> > >
> > >
> > >
> > > As a summary, we thank the reviewer for bringing up the excellent work [1], and for the detailed discussion about the scaling, which helps us better understand many technical details of [1]. However, as we have discussed above, the complexity bounds dervied in [1] do not contradict with our lower bound. If the reviewer is OK with it, we can include in the final version of the paper, the above discussions about the scaling, and the comparison between randomized and deterministic alogirhtm.

---

> ### Author Response · Authors · 2022-08-02
> **Response to Reviewer zULq(Part3)**
>
>
> 5. The soundness of class function in Definition 2; the quality of our upper bound in overparameterized problem.
>
> **[Response]** Thanks for your comment. For your concern about the Definition 2, this is actually a very **common** problem: **minimizing the mean square loss of an overparameterization model**. There are many works considering this function class, though they may differ in the specific function $G_k$. For example, in [Huang, Baihe, et al. "Fl-ntk: A neural tangent kernel-based framework for federated learning analysis." International Conference on Machine Learning. PMLR, 2021.], authors consider the distributed NTK and minimize the MSE loss (see the formula of loss function after Definition 3.1 in the aforementioned work); in [Deng, Yuyang, Mohammad Mahdi Kamani, and Mehrdad Mahdavi. "Local SGD Optimizes Overparameterized Neural Networks in Polynomial Time." International Conference on Artificial Intelligence and Statistics. PMLR, 2022.], authors consider the same function (See Sec 3 of the aforementioned work). In centralized setting, there are more works considering this function class, e.g. [Du, Simon S., et al. "Gradient descent provably optimizes over-parameterized neural networks." arXiv preprint arXiv:1810.02054 (2018).](See equation (2)), [Oymak, Samet, and Mahdi Soltanolkotabi. "Toward moderate overparameterization: Global convergence guarantees for training shallow neural networks." IEEE Journal on Selected Areas in Information Theory 1.1 (2020): 84-105.](See equation(2)),etc…
> For your concern about the bound, as we have illustrated in the reply to comment 4, we generally consider a problem with large size and problem **dimension $D$** is the main issue regarding the complexity. For parameters like $L,\mu$, they are treated as constants. This corresponds to the case when high dimension data or extremely large network is involved, which is very common in literature, especially when overparameterized network is considered. For example, in the discussion at the beginning of [Li, Yuanzhi, and Yingyu Liang. "Learning overparameterized neural networks via stochastic gradient descent on structured data." NeurIPS (2018)], it is concluded that overparameterization can make the networks smoother, which implies $L$ tends to be small when the model size increases(See Lemma 5.2); Meanwhile, from the equation (6) in Theorem 3.3 in [Nguyen, Quynh N., and Marco Mondelli. "Global convergence of deep networks with one wide layer followed by pyramidal topology." NeurIPS 2020.], it is shown that the smallest eigenvalue of the gram matrix has a lower bound in some standard setting. This implies that $\mu$ could not be too small in overparameterized problems. Thus, we can conclude that $L/\mu$ can be viewed as some constants in overparameterized problems and the dimension $D$ can be the dominant term.
>
>
>
>
> 6. Decentralized results seemed to me bonus and not particularly interesting, it uses a multistep gossip protocol with compression, and the authors uses the classic sign compression, not special from Definition 4. The essence of this method is that it is a centralized method, but with a slight inexactness in averaging. It is obvious to me that this can be done.
>
> **[Response]** We agree with your assessment. However, since the fully decentralized setting is also of interest to many applications, for completeness, we think it is useful to include this setting, so that the readers have an idea of how to extend to that setting.

---

> > ### Comment · Reviewer_zULq · 2022-08-05
> > **Reply (Part 3)**
> >
> > I still have one question here.
> >
> > We have the original problem with the parameters $L$, $\mu$. For this problem, the assumption of overparametrization (Definition 2) is satisfied. For this problem I can ignore overparametrization and run MARINA from [1] (Gorbunov et). Or I can take into account the overparametrization and run new algorithm. Which estimate on the number of bits is better?
> > $$
> > O\left(\frac{BD}{\sqrt{K}} \cdot \frac{L}{\mu}\right) ~~~~~ \text{or} ~~~~~~ O\left( \log N \cdot \kappa^4(Z) \right)
> > $$
> >
> >
> > When answering this question, consider my Reply 2 (I think the authors did not understand the results of [1]).

---

> > > ### Author Response · Authors · 2022-08-07
> > > **Response to Reply 3**
> > >
> > > ## Reply 3:
> > > **Our bound is better** than the bound mentioned in the overparameterized problem setting we consider. As we have illustrated in paper and empasized in our response, we generally consider the problem when $D$ is very large and much larger than $N$ (e.g. NTK[“Huang, Baihe, et al. "Fl-ntk: A neural tangent kernel-based framework for federated learning analysis." ICML, 2021.”,“Deng, Yuyang, Mohammad Mahdi Kamani, and Mehrdad Mahdavi. "Local SGD Optimizes Overparameterized Neural Networks in Polynomial Time." ICML, 2022.”]), and thus $\frac{L}{\mu}$ (or $\kappa(Z)$) can be viewed as constants (see response 5 to your question in the previous round). In this case, our bound is  **better** than the bound reviewer has mentioned since the large dimension **$D$ is the main issue** which can not be ignored.

---

> ### Comment · Reviewer_zULq · 2022-08-09
> **Summary of the discussion**
>
> Thanks to the authors for the response, but I'm still not satisfied with the rebuttal.
>
> 1) The estimates of the paper offers still seem bad to me. The authors suggest turning a blind eye to $L$, $\mu$, and $\kappa$ and treating them as constants, but to me this is strange. It seems like we are playing a game: optimal in one place, but highly suboptimal in another.
>
> 2) The authors avoid direct comparison with other works (not necessarily only with Gorbunov et al). There are works that use biased compressors, compressors with fixed number of bits (natural compressors, sign compressors etc).

---

### Official Review · Reviewer_NYne · 2022-07-10

**Rating:** 7
**Confidence:** 4
**Soundness:** 3 good
**Presentation:** 3 good
**Contribution:** 3 good

**Summary:**

This paper studies an important problem in the distributed optimization:  To train an overparameterized model over a set of distributed nodes, what is the minimum number of bits required to reach zero loss. The paper gives lower bounds in two cases: PL and overparameterized case. Then it further gives multiple algorithm examples that can achieve the lower bound under certain conditions: the model is in general close to the structure of linear regression, and the quantization algorithm should be carefully designed.

**Questions:**

1. How will the current bounds adapt if we include the Lipschitz constants $L_k$ and $L$, will it lose tightness if the global function is  the average over all the local functions?

2. Is it immediately clear if the tightness will still hold when the proposed algorithms are solving general convex/non-convex functions?

**Ethics Review Area:**

["I don’t know"]

**Limitations:**

Please refer to the previous reviews.

**Strengths And Weaknesses:**

This paper addresses the gap of bits complexity in distributed optimization, as many of the related papers focus on lower bounds on the communication rounds, especially in the decentralized case. The theoretical results in the paper look promising, since the paper finds the *(near) optimal* rates in two cases and two settings (with a central coordinator and fully decentralized). In terms of theory, this paper gives strong and new (to my knowledge) results. I did not check the proof line-by-line, but the asymptotic rates seem reasonable.

One concern about the theory is that I believe a discussion in terms of the Lipschitz constants should be included. In this paper, the main objective is the sum over all the local functions (not average). This implies the Lipschitz constants between local and global function might be implicitly related by a $N$ factor. I'm not sure if this would make the final bound hide an extra $N$, this should be clarified in the main paper.

My other concerns are on the settings and experiments. More specifically, I'd be interested in seeing how the bounds and tightness would adapt if we are dealing with general convex or non-convex functions (not necessarily PL or least square). On the other hand, this paper is clearly theory-driven, but more experimental results on different applications (models, datasets) would definitely be beneficial, as in small-scale applications, we wouldn't need communication compression or very large models in general.

---

> ### Author Response · Authors · 2022-08-02
> **Response to Reviewer NYne(Part 1)**
>
> Thank you for your comments and suggestions.
>
> Weakness 1: One concern about the theory is that I believe a discussion in terms of the Lipschitz constants should be included. In this paper, the main objective is the sum over all the local functions (not average). This implies the Lipschitz constants between local and global function might be implicitly related by a N factor. I'm not sure if this would make the final bound hide an extra N, this should be clarified in the main paper.
>
> **[Response]** **We have uploaded an updated version, please refer to the new version in the following discussion.** Thank you for pointing out the issue. We have investigated the issue, and we believe that compared with the current lower bound, in the worst case, our upper bound could have an additional multiplicative factor of $K^{\frac{3}{2}}$ (recall that $K$ is the number of users in the system),  but there will be no additional factor of $N$ (the number of data samples).
>
> To facilitate discussion, for simplicity let us assume that $L_k = \underline{L}$ for all k. Then it is easy to verify that, it is generally the case that $L = \mathcal{O}(K \underline{L})$.
>
> Next, let us examine the upper bound in Theorem 4.1 (the revised version). It says that the *individual* communication bits scales with $\frac{L^{\frac{3}{2}}}{\mu} D \log(1/\epsilon)$. Replacing L with the above estimates, we see that this expression is proportional to $K^{\frac{3}{2}}$.  Comparing with our lower bound, as expressed in Theorem 3.1, which says that the *total* communication bits scale with $K D \log(D/\epsilon)$, this means that the individual communication bits scale with $D \log(D/\epsilon)$. So the upper bound has an additional factors of $K^{\frac{3}{2}}$.
>
> However, we should point out that, the focus of this work is to understand the scaling of overparameterized, or PL distributed optimization problems with respect to $D$ and $N$, so we generally treat constants such as $K$ as fixed, and assume that they are not large. We believe that in the future, we can still sharpen our lower bound (i.e., possibly find even harder cases whose complexity has bad scaling on $K$), or develop extensions of our algorithm/analysis to reduce the dependency on $K$.
>
>
>
>
> Weakness 2: On the other hand, this paper is clearly theory-driven, but more experimental results on different applications (models, datasets) would definitely be beneficial, as in small-scale applications, we wouldn't need communication compression or very large models in general.
>
> **[Response]** We apologize for the preliminary experimental results due to limited resources. The purpose of our experiment is to empirically verify the $O(H \log H)$ dependence. The choice of model dimension often depends on the scale of dataset. For example, ImageNet + ResNet50 is a popular NN setup and has a overparameterization of 18x. (1.28 million training samples + 23 million model parameters) (He, Kaiming, et al. "Deep residual learning for image recognition." Proceedings of the IEEE conference on computer vision and pattern recognition. 2016.) Experiment in the first submission shows overparameterization from 1x ~ 5x. In Figure 3 of the revised paper, we extend our experiment up to 19x to complete the picture for deep learning scale application. The result indicates the increase of communication bits are 1.4x and 2.8x respectively when increasing parameterization by 4x, which verifies that our communication complexity is not ${\cal O}(D)$ dependent.

---

> ### Author Response · Authors · 2022-08-02
> **Response to Reviewer NYne(Part 2)**
>
> 1. How will the current bounds adapt if we include the Lipschitz constants  $L_k$ and $L$, will it lose tightness if the global function is the average over all the local functions?
>
> **[Response]** Your first question has already been addressed when we comment on Weakness 1. That is, when the effect of Lipschitz constants are considered, we will have an additional factor of $K^{\frac{3}{2}}$. So our bound is still tight in $D, \epsilon$, but may not be tight in $K$.
> Now let us consider the case where the average function is used.  In this case, L is in the same order as $\frac{1}{K}\sum\limits_k L_k$, so if we still assume $L_k = \underline{L}$ for all k, then we have $L=\underline{L}$. However, in this case, we have checked that the lower bound will also shrink by K, so there is still some gap between the upper and lower bound, in terms of K. the tightness of other constants remains to be the same.
> As we have mentioned above, the focus of this work is to understand the scaling of overparameterized, or PL distributed optimization problems with respect to D and N, so we generally treat constants such as K as fixed, and assume that they are not large. In the future, we plan to sharpen our lower bound about its dependency on K, L, and to develop extensions of our algorithm/analysis to reduce the dependency on K.
>
>
> 2. Is it immediately clear if the tightness will still hold when the proposed algorithms are solving general convex/non-convex functions?
>
> **[Response]** First, let us note that to our knowledge, there has been no bit-lower bound available in literature for general convex/non-convex functions, so one cannot talk about the *tightness* for these cases. However, there indeed have some recent works discussing the bit-quantization of convex/non-convex function, e.g. Alistarh, Dan, et al. "QSGD: Communication-efficient SGD via gradient quantization and encoding." NeurIPS (2017). Since the lower bound discussion is not available in literature, we do not know whether these results are tight or not.  It would be a very interesting and challenging research question to investigate in the future.

---

### Official Review · Reviewer_NYA5 · 2022-07-14

**Rating:** 7
**Confidence:** 4
**Soundness:** 3 good
**Presentation:** 3 good
**Contribution:** 2 fair

**Summary:**

This paper studies the communication complexity (bits communicated) of distributed optimization problems where the client objectives:
1. satisfy the PL condition, or
2. are overparameterized generalized quadratic functions.

It provides lower bounds for these problems and first-order algorithms with appropriate compression/decompression. The proposed algorithm for PL objectives has optimal communication complexity up to logarithmic terms in the dimension of the problem. Similarly, for quadratic overparameterized objectives, it attains optimal communication complexity (up to logarithmic terms) with high probability. It can also be "decentralized" by using the CHOCO estimator proposed by Kolsokova et al. Finally, the paper also studies a stylized feed-forward neural network and shows that in a particular regime, their algorithm also attains optimal communication complexity for it.

**Questions:**

1. I am nitpicking, but what leads to the extra $\log(D)$ factor? Is it coming from initialization and unavoidable?

2. Can the authors show a better upper bound for the neural networks? In practice, one doesn't use such a dramatic pyramidal structure. If not, do the authors expect a better lower bound for this specific class? Or a more specialized quantization is required? It would be good to discuss this aspect, even if the authors can't provide better results.

3. Can the overparameterized class be extended to overparameterized generalized linear models?

4. Can the authors comment on why they need the second inequality in (2)? My intuition is that this is not necessary for smooth functions. What's unique to the overparameterized setting?

5. What are the significant differences in the quadratic setting from the analysis of Korhonen and Alistarh, 2021?

6. I agree that understanding the bit communication complexity is very important but in practice, reducing communication rounds is as much or often more important than lowering the bit complexity. It is because of the overheads involved in each communication between the (edge) device and the server. The communication model authors use in this paper can't capture this because each round transmits a single bit. What are the challenges to extending the current lower bound results to the [intermittent communication](https://arxiv.org/abs/1805.10222) setting to reconcile with existing results such by Arjevani and Shamir, etc.? Are there other communication models that make more sense and can better capture these aspects of communication? It is worth discussing in the paper in more detail.

7. On the lines of the previous point, I'd like to see how the quantization schemes presented here work with local update algorithms commonly used to reduce communication rounds? The moonshot goal is to come up with an algorithm that can attain optimal bit and round complexity at the same time.

8. I would like to see more large-scale experiments. It would be even better if the authors could identify a regime where bit complexity is more critical than communication rounds and capture that in their experiments.

**Strengths And Weaknesses:**

The paper is well-written and clearly describes the problem and results. The results for PL functions and overparameterized quadratics are tight, which is excellent. The pyramidal neural model seemed a bit stylized, and for the usual widths used in practice, the upper bound shows no savings in communication. Overall, I like the paper but have some questions/concerns, due to which I have currently weakly accepted the paper.

---

> ### Author Response · Authors · 2022-08-02
> **Response to Reviewer NYA5(Part1)**
>
> Thank you for your comments. We are glad to discuss the problem you raised.
> 1. I am nitpicking, but what leads to the extra $\log⁡(D)$ factor? Is it coming from initialization and unavoidable?
>
> **[Response]** This $\log(D)$ factor is actually common in literature. For example, see Corollary 2 in [Magnússon, Sindri, Hossein Shokri-Ghadikolaei, and Na Li. "On maintaining linear convergence of distributed learning and optimization under limited communication." IEEE Transactions on Signal Processing 68 (2020): 6101-6116.] ; also see Proposition 5.2 in [Tsitsiklis, John N., and Zhi-Quan Luo. "Communication complexity of convex optimization." Journal of Complexity 3.3 (1987): 231-243.]  Both of the above two works have a $\log(D)$ factor. In our work, this factor is related to the number of  bits that are used to quantize each entry of the gradient at the first communication iteration. The distance between the original gradient $g^0$ and quantized gradient $q^0$ will be enlarged by D times compared to the distance between each entry. Thus, if we want to let $\\|g^0-q^0\\|_{\infty}$ be small, we have to consider the dimension D and the number of bits can not be arbitrarily small.
>
> 2. Can the authors show a better upper bound for the neural networks? In practice, one doesn't use such a dramatic pyramidal structure. If not, do the authors expect a better lower bound for this specific class? Or a more specialized quantization is required? It would be good to discuss this aspect, even if the authors can't provide better results.
>
> **[Response]** Although the Pyramidal Network structure is complicated, and may not be practical, we use such a network to illustrate that our analysis can be extended to (at least some kind of) neural networks.
> Next we address the questions related to upper/lower bounds. To this point, we are not expecting a better lower bound, since it will be very challenging to construct a bad instance for this specific type of network. In the meantime, we are not expecting a better upper bound, at least not better in terms of dependency on N. This is because overparameterized networks usually have width at least $\mathcal{O}(N)$. To achieve the global convergence with small empirical MSE loss, the literature uses network size with higher order of N. For example, in distributed setting, “Huang, Baihe, et al. "Fl-ntk: A neural tangent kernel-based framework for federated learning analysis." ICML, 2021.” requires $\mathcal{O}(N^4)$ for network width; “Deng, Yuyang, Mohammad Mahdi Kamani, and Mehrdad Mahdavi. "Local SGD Optimizes Overparameterized Neural Networks in Polynomial Time." ICML, 2022.” requires $\mathcal{O}(N^{18})$ for network width. Thus, the upper and lower bounds we derived have $\mathcal{O}(N)$ dependency regarding the model size (network width), and they are much smaller.
>
>   3. Can the overparameterized class be extended to overparameterized generalized linear models?
>
>   **[Response]** We think it is possible to extend the overparameterized class to the overparameterized GLM model. To be more specific, let us consider Part 4.1 in [Oymak, Samet, and Mahdi Soltanolkotabi. "Overparameterized nonlinear learning: Gradient descent takes the shortest path?. ICML, 2019], in which the general formula of overparameterized GLM is provided. This can be viewed as a special case of our Definition 2. Due to the space limit in the current paper we will not discuss this in detail, but we will provide a remark if there is space in the final version.
>
> 4. Can the authors comment on why they need the second inequality in (2)? My intuition is that this is not necessary for smooth functions. What's unique to the overparameterized setting?
>
> **[Response]** Your intuition is correct. The second inequality in (2) is not necessary. In our revised version which has been uploaded, we have removed the inequality. Instead, we use Lemma 4.2 to illustrate how the second inequality in (2) is derived. The inequality $\|\nabla f(\theta)\|^{2} \leqslant 2 L \cdot f(\theta)$ is a consequence from Assumption 2 with Assumption 3. We have also provided the proof in Appendix C.1. This Lemma is an extension to Theorem 2.1.15 in  [Y. Nesterov, Introductory lectures on convex optimization: A basic course, Springer, 2004.], with the difference that we do not consider convex functions.

---

> ### Author Response · Authors · 2022-08-02
> **Response to Reviewer NYA5(Part2)**
>
>
>
>
>
>
> 5. What are the significant differences in the quadratic setting from the analysis of Korhonen and Alistarh, 2021?
>
> **[Response]** In Korhonen and Alistarh, 2021, the authors utilize special quadratic problems to represent the family of strongly convex distributed problems, and to construct the communication lower bounds. In our paper, the lower bound is no longer derived over simple quadratic problems, instead we analyzed a class of *overparameterized* quadratic functions and derived the communication complexity upper bound. Therefore the  major difference lies in the type of problem these two works consider. In addition, in Korhonen and Alistarh, 2021 the focus is on the class of *strongly convex* and *smooth* functions, therefore, the quadratic function they used to construct lower bounds is *strongly convex*. On the other hand, in our work our focus has been given to *overparameterized* problems, so the class of quadratic problems we consider is *overparameterized*, and it is more general than the strongly convex problems considered in Korhonen and Alistarh, 2021. Note that *overparameterized* quadratic problems may not be strongly-convex.
> In summary, the quadratic problems considered in these two works are different, and  so the results are not comparable.
>
> 6. I agree that understanding the bit communication complexity is very important but in practice, reducing communication rounds is as much or often more important than lowering the bit complexity. It is because of the overheads involved in each communication between the (edge) device and the server. The communication model authors use in this paper can't capture this because each round transmits a single bit. What are the challenges to extending the current lower bound results to the intermittent communication setting to reconcile with existing results such by Arjevani and Shamir, etc.? Are there other communication models that make more sense and can better capture these aspects of communication? It is worth discussing in the paper in more detail.
>
> **[Response]** Thanks for the question. Let us address your question from the perspective of both lower and upper bounds.
>
> **(Lower Bounds)** The reviewer is correct, that when constructing the communication lower bound, we adopted a protocol that each time only a bit is transmited. The point there was indeed to understand the total bit complexity, rather than the round complexity. It would be interesting to see if it is possible to combine the lower bound construction of bit complexity and round complexity (such as those derived in  Arjevani and Shamir) to yield a better communication model for the lower bound. At this point it is not clear to us how to do it.
>
> **(Upper Bounds)** **We have uploaded an updated version, we will refer to the bound in the new version now**. On the other hand, we believe that the upper bound we have derived are tight (with respect to $\epsilon$), even when measured by communication rounds (measured by iteration $t$). To be specific, we achieve linear convergence, and the total rounds of communication required scales with $\mathcal{O}(\log(1/\epsilon))$ and $\frac{L^{\frac{3}{2}}}{\mu}$, where the communication round is *optimal* for distributed PL function, while the $L,\mu$ dependency is a factor of $\sqrt{L}$ from optimal. However, the $\sqrt{L}$ factor is due to the quantization scheme, which is not considered in classical PL fucntions. To see the optimal bounds, consider solving the PL problem  in centralized setting.  The best iteration complexity scales as $O(\log(1/\epsilon))$ and $\frac{L}{\mu}$; see Theorem 1 in [Karimi, Hamed, Julie Nutini, and Mark Schmidt. "Linear convergence of gradient and proximal-gradient methods under the Polyak-Lojasiewicz condition. Joint European conference on machine learning and knowledge discovery in databases,  2016.].
>
> **(Compared to Arjevani and Shamir)** We notice that they consider the strongly convex function, and used acceleration techiniques to reduce the the complexity to
> $\mathcal{O}(\sqrt{L/\mu}\log(1/\epsilon))$. To our knowledge, there is no accelarationg algorithm regarding PL functions. We may consider the line of accelaration algorithm in the future work, to apply on PL functions.
> Thus we are not expecting to improve the communication efficiency (in terms of total rounds) at this time.

---

> ### Author Response · Authors · 2022-08-02
> **Response to Reviewer NYA5(Part3)**
>
> 7. On the lines of the previous point, I'd like to see how the quantization schemes presented here work with local update algorithms commonly used to reduce communication rounds? The moonshot goal is to come up with an algorithm that can attain optimal bit and round complexity at the same time.
>
> **[Response]** Thanks for making a very valid point. Since our lower bound has already been met, we do not expect that, at least in theory such an algorithm suggested by the reviewer will provide any significant reduction in communication rounds. However, it is possible that there can be a *constant* reduction of communication rounds, for which our upper bound analysis cannot capture. To investigate this issue, in Figure 3 of the revision we have provided some numerical results combining Algorithm 2 with local updates. The result shows that local updates can slightly accelerate convergence while maintaining the same communication cost.
>
>
> 8. I would like to see more large-scale experiments. It would be even better if the authors could identify a regime where bit complexity is more critical than communication rounds and capture that in their experiments.
>
> **[Response]**
> We apologize for the preliminary experimental results due to limited resources. The purpose of our experiment is to empirically verify the $O(H \log H)$ dependence. The choice of model dimension often depends on the scale of dataset. For example, ImageNet + ResNet50 is a popular NN setup and has a overparameterization of 18x. (1.28 million training samples + 23 million model parameters) (He, Kaiming, et al. "Deep residual learning for image recognition." Proceedings of the IEEE conference on computer vision and pattern recognition. 2016.) Experiment in the first submission shows overparameterization from 1x ~ 5x. In Figure 3 of the revised version, we extend our experiment up to 19x to complete the picture for deep learning scale application. The result indicates the increase of communication bits is only 1.4x when the problem dimension increases by 4x, which verifies that our communication complexity is almost independent of $D$.
> The trade-off linking compression quality (bit complexity) with communication rounds is not directly addressed in our theory since our current setting considers total bits complexity instead of bits complexity per round. In practice, the critical point of bit complexity depends on the design of compressor, whereas the critical point of communication rounds depends on the network topology. Our theory suggests that per round bit complexity $(H)$ controls $\kappa(Z)$ and thus the convergence rate in Theorem 4.3. The critical bit complexity is observed for requiring $H \ge N$ where we enters the regime of $\kappa(Z)$ being independent of $D$ in Proposition 1. For communication rounds, it can be seen through the proof of Theorem 4.3 where the number of communcation rounds depends on the desired error and the network topology.

---

> ### Comment · Reviewer_NYA5 · 2022-08-08
> **Response to the rebuttal**
>
> 1. That makes sense, thanks for pointing the references. It would be good to discuss the log factor, somewhere in the appendix maybe.
> 2. Similarly, should add this discussion.
> 3. Thanks, if the authirs can generalize it to GLMs, one way is to include that result in the main body.
> 4. I am glad the authors made the change.
> 5. That makes sense.
> 6. I think this is an important discussion, and the authors must make it a point to **include it in the revised version of their paper**. Another reviewer has made comments about this issue, and I think the clarification provided to them is very useful. In particular, when and when not can one translate from one set of results to another.
> 7. Thanks for the experiment. I would recommend using more number of local steps in the experiment.
> 8. Thanks for the experiment. I would encourage adding this discussion while talking about 6.
>
> I'd increase my score by a point, and wish the authors best of luck.

---

### Meta-Review · Area_Chair_Ufss · 2022-08-27

**Recommendation:** Accept
**Confidence:** Less certain

**Metareview:**

This paper considers the following problem in distributed optimization: To train an overparameterized model over a set of distributed nodes, what is the minimum number of bits required to reach zero loss. The paper gives lower bounds on the bit complexity for two settings: non-convex functions satisfying a PL condition and overparameterized quadratics. The authors then give an algorithm that (1) for PL objectives, has optimal communication complexity (up to logarithmic terms in the dimension of the problem) and (2) for quadratic overparameterized objectives, attains optimal communication complexity (up to logarithmic terms) with high probability.

This paper generated significant discussion in the initial author-reviewer discussion period. Most of the reviewers were quite positive on the paper, and found the results to be interesting and relevant to the community, and found the paper to be well-written. They found the results, which provide near-optimal sample complexity for two settings (PL functions and overparameterized quadratics) to be technically strong, and were impressed by the tightness of the results. One reviewer took issue with certain limitations of the paper, including:
- A limitation of the lower bounds in the paper is that they concern only deterministic methods.
- The results are only tight with respect to the parameters $D$, $N$, and $\epsilon$, and are not necessarily tight with respect to $L$ and $\mu$.
- Some related work can be discussed in more detail.

These limitations do not seem to take away from the novelty, and neither I nor the other reviewers were convinced by the other issues raised in the discussion. As a result, I believe the paper is worth accepting as a starting point for future research in this direction. Nonetheless, the authors are encouraged to expand the discussion around these issues and limitations in the final version of the paper, as well as expand the comparison to related work.

**Award:**

No

---

### Decision · Program_Chairs · 2022-09-14

Accept